# Methods with Local Steps and Random Reshuffling for Generally Smooth Non-Convex Federated Optimization

**Yury Demidovich**[*]
KAUST

**Petr Ostroukhov**[*]
MBZUAI, MIPT

**Grigory Malinovsky**
KAUST

**Samuel Horváth**
MBZUAI

**Martin Takáč**
MBZUAI

**Peter Richtárik**
KAUST

**Eduard Gorbunov**
MBZUAI

## Abstract

Non-convex Machine Learning problems typically do not adhere to the standard smoothness assumption. Based on empirical findings, Zhang et al. (2020b) proposed a more realistic generalized $(L_0, L_1)$-smoothness assumption, though it remains largely unexplored. Many existing algorithms designed for standard smooth problems need to be revised. However, in the context of Federated Learning, only a few works address this problem but rely on additional limiting assumptions. In this paper, we address this gap in the literature: we propose and analyze new methods with local steps, partial participation of clients, and Random Reshuffling without extra restrictive assumptions beyond generalized smoothness. The proposed methods are based on the proper interplay between clients' and server's stepsizes and gradient clipping. Furthermore, we perform the first analysis of these methods under the Polyak-Łojasiewicz condition. Our theory is consistent with the known results for standard smooth problems, and our experimental results support the theoretical insights.

## 1 Introduction

Distributed optimization problems and distributed algorithms have gained a lot of attention in recent years in the Machine Learning (ML) community. In particular, modern problems often lead to the training of deep neural networks with billions of parameters on large datasets (Brown et al., 2020; Kolesnikov et al., 2019). To make the training time feasible (Li, 2020), it is natural to parallelize computations (e.g., stochastic gradients computations), i.e., apply *distributed training* algorithms (Goyal et al., 2017; You et al., 2019; Le Scao et al., 2023). Another motivation for the usage of distributed methods is dictated by the fact that data can be naturally distributed across multiple devices/clients and be private, which is a typical scenario in *Federated Learning* (FL) (Konecný et al., 2016; McMahan et al., 2016; Kairouz et al., 2019).

Typically, such problems are not $L$-smooth as indicated by Defazio & Bottou (2019) that motivated the optimization researchers to study so-called *generalized smoothness assumptions*. In particular, Zhang et al. (2020b) propose $(L_0, L_1)$-smoothness assumption, which allows the norm of the Hessian to grow linearly with the norm of the gradient, and empirically validate it for several problems involving the training of neural networks. In addition, Ahn et al. (2023); Crawshaw et al. (2024b); Wang et al. (2024) demonstrate that linear transformers with few layers satisfy this assumption, highlighting the practical importance of $(L_0, L_1)$-smoothness. Moreover, the theoretical convergence of different methods is studied under $(L_0, L_1)$-smoothness in the literature (Zhang et al., 2020b;a; Koloskova et al., 2023a; Chen et al., 2023; Li et al., 2024a;b; Crawshaw et al., 2024b). Noticeably, most of these methods utilize *gradient clipping* (Pascanu et al., 2013).

However, in the context of Distributed/Federated Learning, the theoretical convergence of methods is weakly explored under $(L_0, L_1)$-smoothness. In particular, only a couple of papers analyze

---

[*]Equal contribution. Contacts: `yury.demidovich@kaust.edu.sa`.

methods with *local steps* and *Random Reshuffling* – two highly important techniques in FL – under $(L_0, L_1)$-smoothness but only with additional restrictive assumptions such as data homogeneity (Liu et al., 2022), bounded variance (Wang et al., 2024) or cosine relatedness (Qian et al., 2021). Also, to the best of our knowledge, there is only a single result for the methods with *partial participation* of clients under $(L_0, L_1)$-smoothness with local steps but without data shuffling Crawshaw et al. (2024a). This leads us to the question: *is it possible to design methods with local steps, Random Reshuffling, and partial participation of clients with provable convergence guarantees under $(L_0, L_1)$-smoothness without additional restrictive assumptions?* In this paper, we give a positive answer to this question.

## 1.1 OUR CONTRIBUTIONS

- **New method with local steps.** We propose a new method with local steps called Clip-LocalGDJ (Algorithm 1). This method can be seen as a version of LocalGD (Mangasarian, 1995; McMahan et al., 2016) with different clients and server stepsizes and (smoothed) gradient clipping (Pascanu et al., 2013) on a server side. We also prove the convergence of Clip-LocalGDJ for distributed non-convex $(L_0, L_1)$-smooth problems without additional assumptions such as data homogeneity used in the previous works (Liu et al., 2022).

- **New method with local steps and Random Reshuffling.** The second method we propose – CLERR (Algorithm 2) – utilizes local steps and Random Reshuffling and clipping once-in-a-epoch. For the new method, we derive rigorous convergence bounds for distributed non-convex $(L_0, L_1)$-smooth problems without additional assumptions such as bounded variance (Wang et al., 2024) or cosine relatedness (Qian et al., 2021).

- **New method with local steps, Random Reshuffling, and partial participation.** We extend RR-CLI (Malinovsky et al., 2023a), utilizing Random Reshuffling of clients (as an alternative to clients' sampling) and clients' data at each meta-epoch, and adjust it to the case of $(L_0, L_1)$-smooth objectives through the usage of (smoothed) gradient clipping at the end of each meta-epoch. For the resulting method called Clipped RR-CLI (Algorithm 3), we derive a convergence rate for distributed non-convex $(L_0, L_1)$-smooth problems without additional restrictive assumptions. To the best of our knowledge, this is the first result for an FL method with partial participation of clients under $(L_0, L_1)$-smoothness assumption.

- **Results for the PŁ-functions.** For all three new methods, we derive new results under Polyak-Łojasiewicz condition (Polyak, 1963; Lojasiewicz, 1963) that, to the best of our knowledge, are the first results for FL methods under $(L_0, L_1)$-smoothness and Polyak-Łojasiewicz condition. The analysis is based on the careful consideration of two possible cases (the gradient is either "small" or "big") and induction proof of the boundedness of certain metrics.

- **Tightness of the results.** The derived results are tight: in the special case of $L$-smooth functions, our results recover the known ones for the non-clipped version of the algorithms.

- **Numerical experiments.** Our numerical experiments illustrate the superiority of the proposed methods over the existing baselines.

## 1.2 PRELIMINARIES

In this paper, we consider a standard distributed optimization problem

$$\min_{x \in \mathbb{R}^d} \left\{ f(x) \stackrel{\text{def}}{=} \frac{1}{M} \sum_{m=1}^{M} f_m(x) \right\}, \tag{1}$$

where $[M] \stackrel{\text{def}}{=} \{1, 2, \dots, M\}$ represents the set of all workers participating in the training, and each $f_m : \mathbb{R}^d \to \mathbb{R}$ is a non-convex function corresponding to the loss computed on the data available on client $m$ for the current model parameterized by $x \in \mathbb{R}^d$. Throughout the paper, we consider two setups: either workers can compute the full gradient $\nabla f_m(x)$ of their loss functions or they can compute only a stochastic gradient at each step. In the latter case, we will assume that functions $\{f_m\}_{m=1}^{M}$ have the finite-sum form

$$f_m(x) = \frac{1}{N} \sum_{j=1}^{N} f_{mj}(x), \quad \forall m \in [M],$$

where $f_{mj}(x)$ corresponds to the local loss of the current model parameterized by $x \in \mathbb{R}^d$, evaluated for the $j$-th data point on the dataset belonging to the $m$-th client.

## 1.3 RELATED WORK

**Local training.** Local Training (LT), where clients perform multiple optimization steps on their local data before engaging in the resource-intensive process of parameter synchronization, stands out as one of the most effective and practical techniques for training FL models. LT was proposed by Mangasarian (1995); Povey et al. (2014); Moritz et al. (2015) and later promoted by McMahan et al. (2016). Early theoretical analyses of LT methods relied on restrictive data homogeneity assumptions, which are often unrealistic in real-world federated learning (FL) settings (Stich, 2018; Li et al., 2019; Haddadpour & Mahdavi, 2019). Later, Khaled et al. (2019a;b) removed limiting data homogeneity assumptions for LocalGD (Gradient Descent (GD) with LT). Then, Woodworth et al. (2020); Glasgow et al. (2022) derived lower bounds for GD with LT and data sampling, showing that its communication complexity is no better than minibatch Stochastic Gradient Descent (SGD) in settings with heterogeneous data. Another line of works focused on the mitigating so-called client drift phenomenon, which naturally occurs in LocalGD applied to distributed problems with heterogeneous local functions (Karimireddy et al., 2020; Tran-Dinh et al., 2021; Gorbunov et al., 2021b; Thapa et al., 2022; Mishchenko et al., 2022; Malinovsky et al., 2023b).

**Random reshuffling.** Although standard Stochastic Gradient Descent (SGD) (Robbins & Monro, 1951) is well-understood from a theoretical perspective (Rakhlin et al., 2012; Bottou et al., 2018; Nguyen et al., 2018; Gower et al., 2019; Drori & Shamir, 2020; Khaled & Richtárik, 2020; Demidovich et al., 2024), most widely-used ML frameworks rely on *sampling without replacement*, as it works better in the training neural networks (Bottou, 2009; Recht & Ré, 2013; Bengio, 2012; Sun, 2020). It leverages the finite-sum structure by ensuring each function is used once per epoch. However, this introduces bias: individual steps may not reflect full gradient descent steps on average. Thus, proving convergence requires more advanced techniques. Three popular variants of sampling without replacement are commonly used. *Random Reshuffling (RR)*, where the training data is randomly reshuffled before the start of every epoch, is an extremely popular and well-studied approach. The aim of RR is to disrupt any potentially untoward default data sequencing that could hinder training efficiency. RR works very well in practice. *Shuffle Once (SO)* is analogous to RR, however, the training data is permuted randomly only once prior to the training process. The empirical performance is similar to RR. *Incremental Gradient (IG)* is identical to SO with the difference that the initial permutation is deterministic. This approach is the simplest, however, ineffective. IG has been extensively studied over a long period (Luo, 1991; Grippo, 1994; Li et al., 2022; Ying et al., 2019; Gürbüzbalaban et al., 2019; Nguyen et al., 2021). A major challenge with IG lies in selecting a particular permutation for cycling through the iterations, a task that Nedic & Bertsekas (2001) highlight as being quite difficult. (Bertsekas, 2015) provides an example that underscores the vulnerability of IG to poor orderings, especially when contrasted with RR. Meaningful theoretical analyses of the SO method have only emerged recently (Safran & Shamir, 2020; Rajput et al., 2020). RR has been shown to outperform both SGD and IG for objectives that are twice-smooth (Gürbüzbalaban et al., 2015; Haochen & Sra, 2019). Jain et al. (2019) examine the convergence of RR for smooth objectives. Safran & Shamir (2020); Rajput et al. (2020) provide lower bounds for RR. Mishchenko et al. (2020) recently conducted a thorough analysis of IG, SO and RR using innovative and simplified proof techniques, resulting in better convergence rates. Recent advances on RR can be found in (Cha et al., 2023; Cai et al., 2023; Koloskova et al., 2023b).

**Generalized smoothness.** Let us remind that the function $f$ is said to be $L$-*smooth* if there exist $L \geq 0$ such that $\|\nabla f(x) - \nabla f(y)\| \leq L \|x - y\|$ for all $x, y \in \mathbb{R}^d$. For twice-differentiable functions, it is equivalent to $\|\nabla^2 f(x)\| \leq L$, for all $x \in \mathbb{R}^d$. This assumption is very standard in the optimization field. Recently, based on extensive experiments, Zhang et al. (2020b) introduced a generalization of this condition called $(L_0, L_1)$-*smoothness*. Namely, twice-differentiable function $f$ is said to be $(L_0, L_1)$-smooth if $\|\nabla^2 f(x)\| \leq L_0 + L_1 \|\nabla f(x)\|$, for all $x \in \mathbb{R}^d$. Compared to the standard smoothness, this condition is its strict relaxation, and it is applied to a broader range of functions. Zhang et al. (2020b) demonstrated empirically that generalized smoothness provides a more accurate representation of real-world task objectives, especially in the context of training deep neural networks. During LSTM training, it was noted that the local Lipschitz constant $L_0$

---

**Algorithm 1** Clip-LocalGDJ: Clipped Local Gradient Descent with Jumping

---

1: **Input:** Synchronization/communication times $0 = t_0 < t_1 < t_2 < \ldots < t_{P-1}$, initial vector $x_0 \in \mathbb{R}^d$, number of epochs $P \geq 1$, constants $c_0, c_1 > 0$.
2: Initialize $x_0^m = \hat{x}_0 = x_0$ for all $m \in [M] \stackrel{\text{def}}{=} \{1, 2, \ldots, M\}$.
3: **for** $p = 0, 1, \ldots, P - 1$ **do**
4:     Choose the server stepsize $\gamma_p = \frac{1}{c_0 + c_1 \|\nabla f(\hat{x}_{t_p})\|}$.
5:     Choose small inner stepsize $\alpha_p > 0$.
6:     **for** $m = 1, \ldots, M$ **do**
7:         $x_{t_p}^m = \hat{x}_{t_p}$
8:         **for** $t \in \{t_p, \ldots t_{p+1} - 2\}$ **do**
9:             $x_{t+1}^m = x_t^m - \alpha_p \nabla f_m(x_t^m)$
10:         **end for**
11:     **end for**
12:     $g_p = \frac{1}{\alpha_p(t_{p+1} - 1 - t_p)} \left( \hat{x}_{t_p} - \frac{1}{M} \sum_{m=1}^{M} x_{t_{p+1}-1}^m \right)$
13:     $\hat{x}_{t_{p+1}} = \hat{x}_{t_p} - \gamma_p g_p$
14: **end for**

---

near the stationary point is thousands of times smaller than the global Lipschitz constant $L$. Under this condition, Zhang et al. (2020b) provided a theoretical justification for the gradient clipping technique (Pascanu et al., 2013), which is considered effective in mitigating the issue of exploding gradients. Their results were improved by (Zhang et al., 2020a; Koloskova et al., 2023a). (Chen et al., 2023) establish various useful properties of generalized-smooth functions, propose generalizations of $(L_0, L_1)$-smoothness and optimal first-order algorithms for solving generalized-smooth non-convex problems. Li et al. (2024a;b) extend the $(L_0, L_1)$-smoothness condition, introduce a novel analysis technique that bounds gradients along the trajectory, analyze GD, SGD, Nesterov's accelerated gradient method and Adam. (Crawshaw et al., 2024b) consider a coordinate-wise version of generalized smoothness. (Ahn et al., 2023; Crawshaw et al., 2024b; Wang et al., 2024) demonstrate that linear transformers with few layers satisfy generalized smoothness empirically. There are few papers on distributed algorithms that combine local steps or reshuffling with generalized smoothness. Qian et al. (2021) examined clipped IG; Wang et al. (2024) investigated Adam with RR; (Liu et al., 2022) studied LocalGD, however, all of the papers contain additional restrictive assumptions. This is a significant gap in the literature and we close it in our paper. Finally, Crawshaw et al. (2024a) use local steps and partial participation for $(L_0, L_1)$-smooth objectives, but they do not use reshuffling and add heterogeneity assumptions.

## 2 NEW METHODS

In this section, we introduce the new methods – Clip-LocalGDJ (Algorithm 1), CLERR (Algorithm 2), and Clipped RR-CLI (Algorithm 3).

**Clip-LocalGDJ.** As standard LocalGD, the first method (Clip-LocalGDJ, Algorithm 1) alternates between local GD steps on each worker and synchronization/averaging steps. However, there are two noticeable differences between Clip-LocalGDJ and LocalGD. The first one is the usage of different clients' and server's stepsizes. In our method, clients' stepsizes are typically smaller than the server's ones, which allows us to handle the client drift. Then, on the server, the pseudogradient $g_p$ is computed, and the server performs a Clip-GD-type step, which is a second important difference compared to LocalGD. Since the server's stepsize is typically larger than the clients' stepsizes, the local steps can be seen as steps determining the update direction, and the server step can be seen as a larger "jump" in the averaged update direction.

**CLERR.** In CLERR (Algorithm 2), each client does a full epoch of RR before between synchronization steps (similarly to (Malinovsky et al., 2023b)), and similarly to Clip-LocalGDJ, (smoothed) clipping is applied only to the averaged pseudogradient $g_t$ once in an epoch. In contrast, a naïve combination of clipping with RR uses clipping at each step, which can amplify the bias of RR and lead to poor performance (as we illustrate in our experiments).

---

**Algorithm 2** CLERR: Clipped once in an Epoch Random Reshuffling

---

1: **Input:** Starting point $x_0 \in \mathbb{R}^d$, number of epochs $T$, constants $c_0, c_1 > 0$.
2: **for** $t = 0, \ldots, T-1$ **do**
3:     Choose global stepsize $\gamma_t = \frac{1}{c_0 + c_1 \|\nabla f(x_t)\|}$.
4:     Choose small inner stepsize $\alpha_t > 0$.
5:     Sample a permutation $\pi_t = \{\pi_t(1), \ldots, \pi_t(N)\}$.
6:     **for** $m = 1, \ldots, M$ **do**
7:        $x_{t,0}^m = x_t$
8:        **for** $j = 0, \ldots, N-1$ **do**
9:           $x_{t,j+1}^m = x_{t,j}^m - \alpha_t \nabla f_{m,\pi_t(j)}(x_{t,j}^m)$.
10:        **end for**
11:        $g_t^m = \frac{1}{\alpha_t N}(x_t - x_{t,N}^m)$
12:     **end for**
13:     $g_t = \frac{1}{M} \sum_{m=1}^M g_t^m$.
14:     $x_{t+1} = x_t - \gamma_t g_t$.
15: **end for**

---

**Clipped RR-CLI.** Clipped RR-CLI (Algorithm 3) is the first FL algorithm that combines clipping, local steps, local dataset reshuffling, server and client step sizes and regularized client partial participation (sampling of clients without replacement). It is based on RR-CLI proposed by Malinovsky et al. (2023a) and leverages the core techniques proposed in FedAvg (McMahan et al., 2016). The key idea is similar to CLERR, but in addition to the reshuffling of clients' data, Clipped RR-CLI performs a reshuffling of the groups of clients as an alternative to the standard i.i.d. sampling of clients at each communication round. At the end of each meta-epoch, the server performs a smoothed Clip-GD-type step similar to the one used in CLERR, which allows the method to make a larger step with an accumulated pseudogradient.

When the number of workers is large, partial participation is preferable. In this case, Clipped RR-CLI (Algorithm 3) is the best option as it utilizes partial participation. Otherwise, if we have access to full gradients on the workers, then Clip-LocalGDJ(Algorithm 1) is preferable. In case when the workers can compute only a stochastic gradient, then CLERR (Algorithm 2) is recommended.

## 3 ASSUMPTIONS

In this section, we list assumptions adopted in the paper.

**Assumption 1.** *There exists $f^\star, f_m^\star, f_{mj}^\star \in \mathbb{R}$ such that $f(x) \geq f^\star$, $f_m(x) \geq f_m^\star$, $f_{mj}(x) \geq f_{mj}^\star$, $m \in [M], j \in [N]$, for all $x \in \mathbb{R}^d$.*

The next assumption is a strict relaxation of the standard smoothness.

**Assumption 2** (Asymmetric $(L_0, L_1)$-smoothness (Zhang et al., 2020b; Chen et al., 2023))**.** *The functions $f(x)$, $\{f_m(x)\}_{m=1}^M$ and $\{f_{mj}(x)\}_{m=1,j=1}^{M,N}$ are asymmetrically $(L_0, L_1)$-smooth:*

$$\|\nabla f(x) - \nabla f(y)\| \leq (L_0 + L_1\|\nabla f(x)\|)\|x - y\|, \quad \forall x, y \in \mathbb{R}^d,$$

$$\|\nabla f_m(x) - \nabla f_m(y)\| \leq (L_0 + L_1\|\nabla f_m(x)\|)\|x - y\|, \quad \forall m \in [M], x, y \in \mathbb{R}^d,$$

$$\|\nabla f_{mj}(x) - \nabla f_{mj}(y)\| \leq (L_0 + L_1\|\nabla f_{mj}(x)\|)\|x - y\|, \quad \forall m \in [M], j \in [N], x, y \in \mathbb{R}^d.$$

Empirical findings of Zhang et al. (2020b) revealed that generalized smoothness characterizes real-world task objectives in a more precise way, particularly when applied to the training of DNNs. Moreover, the above assumption is satisfied in Distributionally Robust Optimization for some problems (Jin et al., 2021).

The assumption below generalizes the smoothness condition even further.

**Assumption 3** (Symmetric $(L_0, L_1)$-smoothness (Chen et al., 2023))**.** *The functions $f(x)$, $\{f_m(x)\}_{m=1}^M$ and $\{f_{mj}(x)\}_{m=1,j=1}^{M,N}$ are symmetrically $(L_0, L_1)$-smooth:*

$$\|\nabla f(x) - \nabla f(y)\| \leq (L_0 + L_1 \sup_{u \in [x,y]} \|\nabla f(u)\|)\|x - y\|, \quad \forall x, y \in \mathbb{R}^d,$$

---

**Algorithm 3** Clipped RR-CLI: Federated optimization with server and global steps, clipping, random shuffling and partial participation with shuffling

---

1: **Input:** cohort size $C \in \{1, 2, \ldots, M\}$; number of rounds $R = M/C$; initial iterate/model $x_0 \in \mathbb{R}^d$; number of meta-epochs $T \geq 1$, constants $c_0, c_1 > 0$.
2: **for** meta-epoch $t = 0, 1, \ldots, T - 1$ **do**
3:     Choose global stepsize $\theta_t = \frac{1}{c_0 + c_1 \|\nabla f(x_t)\|}$.
4:     Choose small server stepsize $\eta_t > 0$.
5:     Choose small client stepsize $\gamma_t > 0$.
6:     $x_t^0 = x_t$
7:     **Client-Reshuffling:** sample a permutation $\lambda = (\lambda_0, \lambda_1, \ldots, \lambda_{R-1})$ of $[R]$
8:     **for** communication rounds $r = 0, \ldots, R - 1$ **do**
9:         Send model $x_t^r$ to participating clients $m \in S_t^{\lambda_r}$    (server broadcasts $x_t^r$ to clients $m \in S_t^{\lambda_r}$)
10:         **for** all clients $m \in S_t^{\lambda_r}$, locally in parallel **do**
11:             $x_{m,t}^{r,0} = x_t^r$       (client $m$ initializes local training using the latest global model $x_t^r$)
12:             **Data-Random-Reshuffling:** sample a permutation $\pi_m = (\pi_m^0, \pi_m^1, \ldots, \pi_m^{N-1})$ of $[N]$
13:             **for** all local training data points $j = 0, 1, \ldots, N - 1$ **do**
14:                 $x_{m,t}^{r,j+1} = x_{m,t}^{r,j} - \gamma_t \nabla f_m^{\pi_m^j}(x_{m,t}^{r,j})$    (client $m$ passes once its local data in $\pi_m$ order)
15:             **end for**
16:             $g_{m,t}^r = \frac{1}{\gamma_t N}(x_t^r - x_{m,t}^{r,N})$       (client $m$ computes local update direction $g_{m,t}$)
17:         **end for**
18:         $g_t^r = \frac{1}{C} \sum_{m \in S_t^{\lambda_r}} g_{m,t}^r$       (server aggregates local directions $g_{m,t}$ of the clients cohort $S_t$)
19:         $x_t^{r+1} = x_t^r - \eta_t g_t^r$    (server updates the model in aggregated direction $g_t$ with server stepsize $\eta_t$)
20:     **end for**
21:     $g_t = \frac{1}{R} \sum_{i=0}^{R-1} g_t^r$
22:     $x_{t+1} = x_t - \theta_t g_t$       (global step after all communication rounds during meta-epoch)
23: **end for**

---

$$\|\nabla f_m(x) - \nabla f_m(y)\| \leq (L_0 + L_1 \sup_{u \in [x,y]} \|\nabla f_m(u)\|)\|x - y\|, \quad \forall m \in [M], x, y \in \mathbb{R}^d,$$

$$\|\nabla f_{mj}(x) - \nabla f_{mj}(y)\| \leq (L_0 + L_1 \sup_{u \in [x,y]} \|\nabla f_{mj}(u)\|)\|x - y\|, \quad \forall m \in [M], j \in [N], x, y \in \mathbb{R}^d.$$

A common generalization of strong convexity in the literature is the Polyak–Łojasiewicz condition.

**Assumption 4** (Polyak–Łojasiewicz condition (Polyak, 1963; Lojasiewicz, 1963)). *Suppose Assumption 1 holds for the function $f$. There exists $\mu > 0$, such that $\|\nabla f(x)\|^2 \geq 2\mu (f(x) - f^\star)$.*

## 4 THEORETICAL CONVERGENCE RATES

In this section, we describe our convergence results. Let us first introduce the notation. Put $\Delta^\star \stackrel{\text{def}}{=} f^\star - \frac{1}{M} \sum_{m=1}^M f_m^\star$, $\overline{\Delta}^\star \stackrel{\text{def}}{=} f^\star - \frac{1}{M} \sum_{m=1}^M \frac{1}{N} \sum_{j=0}^{N-1} f_{mj}^\star$. Define $\delta_0 \stackrel{\text{def}}{=} f(x_0) - f^\star$. Let $\zeta$ be a constant such that $0 < \zeta \leq \frac{1}{4}$. Fix accuracy $\varepsilon > 0$. Let $P \geq 1$ be the number of epochs. For all $0 \leq p \leq P - 1$, denote

$$\hat{a}_p = L_0 + L_1 \|\nabla f(\hat{x}_{t_p})\|, \quad a_p = L_0 + L_1 \max_m \|\nabla f_m(\hat{x}_{t_p})\|, \quad 1 \leq t_{p+1} - t_p \leq H.$$

We start by formulating the convergence result for Clip-LocalGDJ (Algorithm 1) in non-convex asymmetric generalized-smooth case. More details can be found in Appendix B.1.

**Theorem 1.** *Let Assumptions 1 and 2 hold. Choose any $P \geq 1$. Choose small local stepsizes $\alpha_p$, server stepsizes $\gamma_p$ so that $\frac{\zeta}{\hat{a}_p} \leq \gamma_p \leq \frac{1}{4\hat{a}_p}$. Then, the iterates $\left\{\hat{x}_{t_p}\right\}_{p=0}^{P-1}$ of Algorithm 1 satisfy*

$$\min_{0 \leq p \leq P-1} \left\{ \frac{\zeta}{8} \min \left\{ \frac{\|\nabla f(\hat{x}_{t_p})\|^2}{L_0}, \frac{\|\nabla f(\hat{x}_{t_p})\|}{L_1} \right\} \right\} \leq \frac{\left(1 + \frac{3(H-1)^2 \alpha_p^2 a_p^3}{2\hat{a}_p}\right)^P}{P} \delta_0$$
$$+ \frac{3(H-1)^2 \alpha_p^2 a_p^3}{2\hat{a}_p} \Delta^\star.$$

**Corollary 1.** *If* $P \geq \frac{32\delta_0}{\zeta\varepsilon}$ *and* $\alpha_p$ *is small enough, then* $\min_{0 \leq p \leq P-1} \left\{ \min\left\{ \frac{\|\nabla f(\hat{x}_{t_p})\|^2}{L_0}, \frac{\|\nabla f(\hat{x}_{t_p})\|}{L_1} \right\} \right\} \leq \varepsilon.$

The rates we obtain in Corollary 1 are consistent with the previously established rates of LocalGD and GD in the standard smooth case, i.e., when $L_1 = 0$. Indeed, we recover the rate $\mathcal{O}\left(\frac{L_0\delta_0}{\varepsilon}\right)$ for LocalGD (Koloskova et al., 2020). Notice, that if $H = 1$, the Algorithm 1 reduces to vanilla GD, and we recover its rate $\mathcal{O}\left(\frac{L_0\delta_0}{\varepsilon}\right)$ (Khaled & Richtárik, 2020). In the $(L_0, L_1)$-smooth case, setting $H = 1$, we recover the rate $\mathcal{O}\left(\frac{L_0\delta_0}{\varepsilon}\right)$ of clipped GD from (Zhang et al., 2020b).

Below we state the convergence result for Clip-LocalGDJ (Algorithm 1) in non-convex asymmetric generalized-smooth case under the PŁ-condition. For more details, see Appendix B.2.

**Theorem 2.** *Let Assumptions 1 and 2 hold. Let Assumption 4 hold. Choose any integer $P > \frac{64\delta_0 L_1^2}{\mu\zeta}$. Choose small local stepsizes $\alpha_p$, server stepsizes $\gamma_p$ so that $\frac{\zeta}{\hat{a}_p} \leq \gamma_p \leq \frac{1}{4\hat{a}_p}$. Let $\tilde{P}$ be an integer such that $0 \leq \tilde{P} \leq \frac{64\delta_0 L_1^2}{\mu\zeta}$, $A > 0$ be a constant, $\alpha \leq \sqrt{\frac{\delta_0}{AP}}$. Put $\delta_P \overset{def}{=} f\left(\hat{x}_{t_P}\right) - f^\star$. Then, the iterates $\left\{\hat{x}_{t_p}\right\}_{p=0}^P$ of Algorithm 1 satisfy*

$$\delta_P \leq \left(1 - \frac{\mu\zeta}{4L_0}\right)^{P-\tilde{P}} \delta_0 + \frac{4L_0 A\alpha^2}{\mu\zeta}.$$

**Corollary 2.** *Choose $\alpha \leq \min\left\{ \sqrt{\frac{\delta_0}{AP}}, L_1\sqrt{\frac{8\delta_0\varepsilon}{L_0 AP}} \right\}$. If $P \geq \frac{64\delta_0 L_1^2}{\mu\zeta} + \frac{4L_0}{\mu\zeta}\ln\frac{2\delta_0}{\varepsilon}$, then $\delta_P \leq \varepsilon$.*

In the standard smooth case, when $L_1 = 0$, we guarantee the iteration complexity $\mathcal{O}\left(\frac{L_0}{\mu}\ln\frac{2\delta_0}{\varepsilon}\right)$, which matches the LocalGD (Koloskova et al., 2020) and GD (Khaled & Richtárik, 2020) rates.

The above results can be generalized to the symmetric $(L_0, L_1)$-smooth case, see Theorem 5 in Appendix B.3 for details.

Let $T \geq 1$ be the number of epochs. For all $0 \leq t \leq T-1$, denote

$$\hat{a}_t = L_0 + L_1\|\nabla f(x_t)\|, \quad a_t = L_0 + L_1\max_m\|\nabla f_m(x_t)\|, \quad \tilde{a}_t = L_0 + L_1\max_{m,j}\|\nabla f_{mj}(x_t)\|.$$

Further, we outline the convergence result for CLERR (Algorithm 2) in non-convex asymmetric generalized-smooth case. For more details, see Appendix C.1.

**Theorem 3.** *Let Assumptions 1 and 2 hold. Choose any $T \geq 1$. Choose small client stepsizes $\alpha_t$, global stepsizes $\gamma_t$ so that $\frac{\zeta}{\hat{a}_t} \leq \gamma_t \leq \frac{1}{4\hat{a}_t}$. Then, the iterates $\{x_t\}_{t=0}^{T-1}$ of Algorithm 2 satisfy*

$$\mathbb{E}\left[\min_{t=0,\dots,T-1}\left\{ \frac{\zeta}{8}\min\left\{ \frac{\|\nabla f(x_t)\|^2}{L_0}, \frac{\|\nabla f(x_t)\|}{L_1} \right\} \right\}\right]$$
$$\leq \frac{8\left(1 + \frac{3\alpha_t^2\tilde{a}_t^3}{8\hat{a}_t}((N-1)(2N-1) + 2(N+1))\right)^T}{T}\delta_0 + \frac{6\alpha_t^2\tilde{a}_t^3}{\hat{a}_t}(N+1)\Delta^\star. \quad (2)$$

**Corollary 3.** *If $T \geq \frac{256\delta_0}{\zeta\varepsilon}$ and $\alpha_t$ is small enough, we have* $\mathbb{E}\left[\min_{t=0,\dots,T-1}\left\{\min\left\{ \frac{\|\nabla f(x_t)\|^2}{L_0}, \frac{\|\nabla f(x_t)\|}{L_1} \right\}\right\}\right] \leq \varepsilon.$

In the standard smooth case, we recover the rate $\mathcal{O}\left(\frac{L_0\delta_0}{\varepsilon}\right)$ of RR (Mishchenko et al., 2020).

We relegate the convergence result for CLERR (Algorithm 2) in non-convex asymmetric generalized-smooth case under the PŁ-condition to Appendix C.2. In the standard smooth case we recover the rate $\mathcal{O}\left(\frac{L_0}{\mu}\ln\frac{2\delta_0}{\varepsilon}\right)$ of RR (Mishchenko et al., 2020).

Further, we formulate the convergence result for Clipped RR-CLI (Algorithm 3) in non-convex asymmetric generalized-smooth case. For more details, see Appendix D.1.

**Theorem 4.** *Let Assumptions 1 and 2 hold for functions $f$, $\{f_m\}_{m=1}^M$ and $\{f_{mj}\}_{m=1,j=1}^{M,N}$. Choose any $T \geq 1$. Choose small local stepsizes $\gamma_t$, small server stepsizes $\eta_t$, global stepsizes $\theta_t$ so that $\frac{\zeta}{\hat{a}_t} \leq \theta_t \leq \frac{1}{4\hat{a}_t}$. Then, the iterates $\{x_t\}_{t=0}^{T-1}$ of Algorithm 3 satisfy*

$$
\mathbb{E}\left[\min_{0 \leq t \leq T-1}\left\{\frac{\zeta}{8}\min\left\{\frac{\|\nabla f(x_t)\|^2}{L_0}, \frac{\|\nabla f(x_t)\|}{L_1}\right\}\right\}\right]
$$

$$
\leq \frac{\left(1 + \frac{2\hat{a}_t\tilde{a}_t^2 + \hat{a}_t^3}{4\hat{a}_t^2}\left(\eta_t^2 a_t + \eta_t^2 R^2 \hat{a}_t + \gamma_t^2 N\tilde{a}_t + \eta_t^2 R a_t\right)\right)^T}{T}\delta_0
$$

$$
+ \frac{2\hat{a}_t\tilde{a}_t^2 + \hat{a}_t^3}{4\hat{a}_t^2}\left(\eta_t^2 a_t \Delta^\star + \gamma_t^2 N\tilde{a}_t \overline{\Delta}^\star + \eta_t^2 R a_t \Delta^\star\right).
$$

**Corollary 4.** *If $T \geq \frac{72\delta_0}{\zeta\varepsilon}$ and $\gamma_t, \eta_t$ are small enough, then $\mathbb{E}\left[\min_{t=0,\dots,T-1}\left\{\min\left\{\frac{\|\nabla f(x_t)\|^2}{L_0}, \frac{\|\nabla f(x_t)\|}{L_1}\right\}\right\}\right] \leq \varepsilon$.*

Finally, we provide the convergence result for Clipped RR-CLI (Algorithm 3) in non-convex asymmetric generalized-smooth case under the PŁ-condition in Appendix D.2.

## 5 EXPERIMENTS

We split our experimental results into 3 parts. In Section 5.1, we provide results for the Algorithm 2 with random reshuffling and jumping in the end of each epoch. In Section 5.2, we consider Algorithm 1 with local steps and jumping in the end of every communication round. In Section 5.3 we consider Algorithm 3, which has local steps, uses random reshuffling of clients and client data and performs jumping in the end of every epoch. Moreover, in Section F we provide additional technical details on the experiments. Finally, in Section G we provide additional experiments, that did not fit in the main text. In Section G.1 we investigate the influence of inner step size on the convergence of Algorithm 2, and in Section G.2 we provide additional logistic regression experiments.

All the mentioned methods have a parameterized stepsize $\gamma_t = \frac{1}{c_0 + c_1\|g_t\|}$. If we denote

$$
\beta = \frac{1}{2c_0}, \quad \lambda = \frac{c_0}{c_1}, \tag{3}
$$

we can estimate $\gamma_t$ as stepsize multiplied by clipping coefficient: $\frac{\beta}{2}\min\left\{1, \frac{\lambda}{\|g_t\|}\right\} \leq \gamma_t \leq \beta\min\left\{1, \frac{\lambda}{\|g_t\|}\right\}$. We use this connection in the process of tuning constants $c_0$ and $c_1$.

In our experiments, we consider the synthetic problem, a sum of shifted fourth-order functions:

$$
f(x) = \frac{1}{N}\sum_{i=1}^N \|x - x_i\|^4, \ x_i \in [-10, 10]^d. \tag{4}
$$

The main reason to consider this problem is that it is $(L_0, L_1)$-smooth, but not $L$-smooth Zhang et al. (2020b). Additionally, in Section 5.1.1 we consider the problem of image classification of ResNet-20 He et al. (2016) on CIFAR-10 dataset Krizhevsky et al. (2009). All the methods and baselines were tuned with grid-search over logarithmic grid.

### 5.1 METHODS WITH RANDOM RESHUFFLING

We conduct this experiment on problem (4), where $d = 1$, $N = 1000$. We consider the Shuffle Once methods, which shuffle data once at the beginning of training. As baselines, we consider the following methods: regular SO method, which is just SGD with shuffling at the start of training, Nastya from Malinovsky et al. (2022) with one worker, Clipped SO (CSO), which clips stochastic gradients at every step of the method. The results are presented in Figure 1. As one can see from Figure 1, methods with clipping significantly outperform the rest. This empirical result justifies the theoretical fact of the importance of clipping for optimization of $(L_0, L_1)$-smooth objectives. Additionally, we see that among methods with clipping, CLERR shows better results than CSO. From this, we can conclude that clipping the final (pseudo)gradient approximation at the end of an epoch gives better results than clipping on every step.

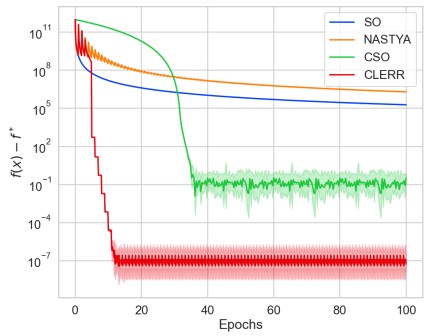

Figure 1: Function residual for (4), $\alpha_t = 10^{-7}$.

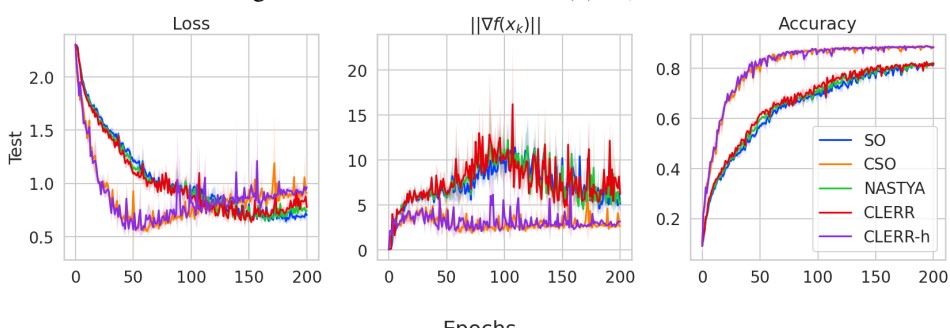

Figure 2: Loss, gradient norm and accuracy on test dataset for ResNet-18 on Cifar-10, $\alpha_t = 0.01$

### 5.1.1 RESNET-18 ON CIFAR-10

In Zhang et al. (2020b) the authors obtained results on a positive correlation between gradient norm and local smoothness for the problem of training neural networks in language modeling and image classification tasks. To check, whether our findings in synthetic experiments also take place for neural networks, we decided to test Algorithm 2 in the same image classification problem: train ResNet-18 He et al. (2016) on the CIFAR-10 dataset Krizhevsky et al. (2009). Additionally, we consider heuristical modification of Algorithm 2, which we call CLERR-h. The details of it we provide next. The overall results of the experiment on test data are shown in Figure 2. Additionally, we provide results on train data along with technical details in Appendix **??**.

From this Figure we can see, that both jumping (Nastya and CLERR) and clipping on outer step (CLERR) does not have any impact on this problem. On the other hand, CSO shows the best results. Since in this problem regular clipping already works very well, we decided to heuristically modify our Algorithm 2: take the best clipping level and inner stepsize of CSO and use it on inner iterations, and tune $c_0$ with $c_1$ for outer stepsize. We call this method CLERR-h and also provide its results in Figure 2. CLERR-h chooses a rather big outer stepsize, while the outer clipping level is very tiny. For big clipping levels method diverges. These results show that jumping does not give performance gains when the method clips on every inner step.

### 5.2 METHODS WITH LOCAL STEPS

In this experiment, we aim to show the effect of the jumping technique on federated learning methods. We consider problem (4) with $d = 100, N = 1000$. To make the distributions of data on each client more distinct between each other, we sort the whole dataset at the beginning of the experiment by $\|x_i\|$. Here we consider a high-dimensional setup so that the starting point has less impact on the algorithm performance. Indeed, in one-dimensional case, if we started from $x_0 \notin [-10, 10]$, the anti-gradient of every $f_i(x) = (x - x_i)^4$ would point towards minimum. Therefore, we could find such stepsize, that method converges in one iteration. On the other hand, if we consider a high-dimensional setup, then regardless of the starting point, the gradient of each $f_i(x)$ has a different direction. In this experiment we compare Algorithm 1 (C-LGDJ) with Communication Efficient Local Gradient Clipping (CELGC) (Liu et al., 2022) and Clipping-Enabled-FedAvg (CE-FedAvg) Zhang et al. (2022). The results are shown in Figure 3.

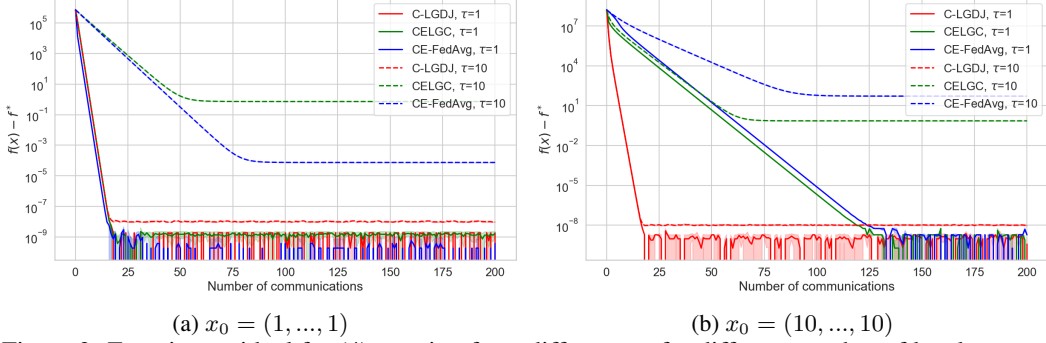

(a) $x_0 = (1, ..., 1)$            (b) $x_0 = (10, ..., 10)$

Figure 3: Function residual for (4), starting from different $x_0$ for different number of local steps on the client device $\tau$.

Overall, we arrive at two conclusions. Firstly, local steps do not have any positive effect on this problem. The plots with the increased number of client steps $\tau$ only strengthen this point. Secondly, since local steps are pointless, the method works better if the server gets a better gradient approximation, which is true if the method clips gradients on the server, not on the client. This is exactly the reason why C-LGDJ has better performance in Figure 3b.

## 5.3 METHODS WITH LOCAL STEPS, RANDOM RESHUFFLING AND PARTIAL PARTICIPATION

In the final experiment, we consider methods with partial participation. The goal of this experiment is to investigate how clipping, local steps, partial participation and random reshuffling of both clients and client data works together. We compare Algorithm 3 with CE-FedAvg Zhang et al. (2022) with partial participation (CE-FedAvg-PP) on problem (4) with $d = 100, N = 1000$. Again, to make the distributions of data on each client more distinct between each other, we sort the whole dataset at the beginning of the experiment by $\|x_i\|$. The results are presented in Figure 4.

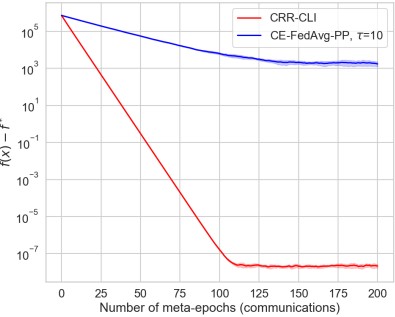

Since CRR-CLI uses random reshuffling of the data instead of sampling with replacement, and clips only in the end of meta-epoch, it has better gradient approximation on the global step, which results in better performance, than CE-FedAvg-PP.

Figure 4: Function residual for (4), starting from $x_0 = (1, ..., 1)$ with batch size 16.

## 6 DISCUSSION

In this paper, we consider a more general smoothness assumption and propose three new distributed methods for Federated Learning with local steps under this setting. Specifically, we analyze local gradient descent (GD) steps, local steps with Random Reshuffling, and a method that combines local steps with Random Reshuffling and Partial Participation. We provide a tight analysis for general non-convex and Polyak-Łojasiewicz settings, recovering previous results as special cases. Furthermore, we present numerical results to support our theoretical findings.

For future work, it would be valuable to explore local methods with communication compression under the generalized smoothness assumption, as well as methods incorporating incomplete local epochs. Additionally, investigating local methods with client drift reduction mechanisms to address the effects of heterogeneity, along with potentially parameter-free approaches, represents a promising direction.

ACKNOWLEDGEMENTS

The authors thank M. Crawshaw for drawing our attention to the related literature, Éric Moulines for spotting a mistake in the proof (which affected the choice of the inner stepsizes, but the main results remained the same), and anonymous reviewers for valuable feedback.

The work of Yury Demidovich, Grigory Malinovsky and Peter Richtárik was supported by funding from King Abdullah University of Science and Technology (KAUST): i) KAUST Baseline Research Scheme, ii) Center of Excellence for Generative AI, under award number 5940, iii) SDAIA-KAUST Center of Excellence in Artificial Intelligence and Data Science.

The work of Petr Ostroukhov was supported by the Ministry of Science and Higher Education of the Russian Federation (Goszadaniye), project No. FSMG-2024-0011.

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

CONTENTS

## A  IMPLICATIONS OF GENERALIZED SMOOTHNESS

**Lemma 1.** *Let $f$ satisfy Assumption 2. Then, for any $x, y \in \mathbb{R}^d$ we have*

$$f(y) \leq f(x) + \langle \nabla f(x), y - x \rangle + \frac{L_0 + L_1 \|\nabla f(x)\|}{2} \|x - y\|^2.$$

*Moreover, if $f^\star := \inf_{x \in \mathbb{R}^d} f(x) > -\infty$, then, for all $x \in \mathbb{R}^d$, we obtain*

$$\frac{\|\nabla f(x)\|^2}{2(L_0 + L_1 \|\nabla f(x)\|)} \leq f(x) - f^\star.$$

*Proof of Lemma 1.* The first statement of the lemma is proven in (Zhang et al., 2020b, Appendix A.1). The second statement is proven in (Gorbunov et al., 2024): it follows from the first statement, if one substitutes $y$ for $x - \frac{\|\nabla f(x)\|}{L_0 + L_1 \|\nabla f(x)\|} \nabla f(x)$ and uses the fact that $f^\star \leq f(y)$. $\qquad\square$

**Lemma 2.** *Let $f$ satisfy Assumption 3. Then, for any $x, y \in \mathbb{R}^d$ we have*

$$f(y) - f(x) \leq \langle \nabla f(x), y - x \rangle + \frac{L_0 + L_1 \|\nabla f(x)\|}{2} \exp(L_1 \|x - y\|) \|x - y\|^2.$$

*Moreover, if $f^\star := \inf_{x \in \mathbb{R}^d} f(x) > -\infty$, then, for $\eta > 0$, such that $\eta \exp \eta \leq 1$, for all $x \in \mathbb{R}^d$, we obtain*

$$\frac{\eta \|\nabla f(x)\|^2}{2(L_0 + L_1 \|\nabla f(x)\|)} \leq f(x) - f^\star.$$

*Proof of Lemma 2.* The first part of this lemma is one of the results of (Chen et al., 2023, Proposition 3.2). The second statement is proven in (Gorbunov et al., 2024). To deal with it, let us substitute $y$ in the first statement with $x - \frac{\eta \|\nabla f(x)\|}{L_0 + L_1 \|\nabla f(x)\|} \nabla f(x)$ :

$$f^\star \leq f(y) \leq f(x) + \langle \nabla f(x), y - x \rangle + \frac{L_0 + L_1 \|\nabla f(x)\|}{2} \exp(L_1 \|x - y\|) \|x - y\|^2$$

$$= f(x) - \frac{\eta \|\nabla f(x)\|^2}{L_0 + L_1 \|\nabla f(x)\|}$$

$$+ \frac{L_0 + L_1 \|\nabla f(x)\|}{2} \cdot \exp\left(\frac{L_1 \eta \|\nabla f(x)\|}{L_0 + L_1 \|\nabla f(x)\|}\right) \cdot \frac{\eta^2 \|\nabla f(x)\|^2}{(L_0 + L_1 \|\nabla f(x)\|)^2}$$

$$\leq f(x) - \frac{\eta \|\nabla f(x)\|^2}{L_0 + L_1 \|\nabla f(x)\|} + \frac{\eta \|\nabla f(x)\|^2}{2(L_0 + L_1 \|\nabla f(x)\|)} \cdot \eta \exp(\eta)$$

$$\leq f(x) - \frac{\eta \|\nabla f(x)\|^2}{2(L_0 + L_1 \|\nabla f(x)\|)}.$$

Rearranging the terms, we get the second statement of the lemma. $\qquad\square$

**Lemma 3.** *Assumption 3 holds for the function $f$ if and only if, for any $x, y \in \mathbb{R}^d$,*

$$\|\nabla f(x) - \nabla f(y)\| \leq (L_0 + L_1 \|\nabla f(y)\|) \exp(L_1 \|x - y\|) \|x - y\|.$$

*Proof of Lemma 3.* This lemma is one of the results of (Chen et al., 2023, Proposition 3.2) $\qquad\square$

# B   LOCAL GRADIENT DESCENT

## B.1   ASYMMETRIC GENERALIZED-SMOOTH NON-CONVEX FUNCTIONS

**Theorem 1** (non-convex asymmetric generalized-smooth convergence analysis of Algorithm 1). *Let Assumptions 1 and 2 hold for functions $f$ and $\{f_m\}_{m=1}^M$. Choose any $P \geq 1$. For all $0 \leq p \leq P-1$, denote*

$$\hat{a}_p = L_0 + L_1 \|\nabla f(\hat{x}_{t_p})\|, \quad a_p = L_0 + L_1 \max_m \|\nabla f_m(\hat{x}_{t_p})\|, \quad 1 \leq t_{p+1} - t_p \leq H.$$

*Put $\Delta^\star = f^\star - \frac{1}{M} \sum_{m=1}^M f_m^\star$. Impose the following conditions on the local stepsizes $\alpha_p$ and server stepsizes $\gamma_p$ :*

$$\alpha_p \leq \min\left\{ \frac{1}{2Ha_p}, \frac{1}{ca_p}\sqrt{\frac{\hat{a}_p}{a_p}} \right\}, \quad \frac{\zeta}{\hat{a}_p} \leq \gamma_p \leq \frac{1}{4\hat{a}_p}, \quad 0 \leq p \leq P-1,$$

*where $0 < \zeta \leq \frac{1}{4}$, $c \geq \sqrt{P}$. Let $\delta_0 \overset{def}{=} f(x_0) - f^\star$. Then, the iterates $\{\hat{x}_{t_p}\}_{p=0}^{P-1}$ of Algorithm 1 satisfy*

$$\min_{0 \leq p \leq P-1}\left\{ \frac{\zeta}{8} \min\left\{ \frac{\|\nabla f(\hat{x}_{t_p})\|^2}{L_0}, \frac{\|\nabla f(\hat{x}_{t_p})\|}{L_1} \right\} \right\} \leq \frac{\left(1 + \frac{3(H-1)^2\alpha_p^2 a_p^3}{2\hat{a}_p}\right)^P}{P}\delta_0$$
$$+ \frac{3(H-1)^2\alpha_p^2 a_p^3}{2\hat{a}_p}\Delta^\star.$$

Put $v_p \overset{def}{=} t_{p+1} - 1$.

**Lemma 4.** *Assume that $f$ and each $f_m$ satisfy Assumptions 1 and 2. Then we have the following bound:*

$$\frac{1}{M} \sum_{m=1}^M \sum_{t=t_p+1}^v \left\| x_t^m - \hat{x}_{t_p} \right\|^2 \leq 8 \left(v_p - t_p\right)^3 a_p \alpha_p^2 \left( f(\hat{x}_{t_p}) - f^\star + \Delta^\star \right).$$

*Proof of Lemma 4.*  We have

$$\left\| x_t^m - \hat{x}_{t_p} \right\|^2 = \left\| \sum_{j=t_p}^{t-1} \alpha_p \nabla f_m\left(x_j^m\right) \right\|^2$$

$$\leq 2 \left\| \sum_{j=t_p}^{t-1} \alpha_p \left(\nabla f_m(x_j^m) - \nabla f_m(\hat{x}_{t_p})\right) \right\|^2 + 2 \left\| \sum_{j=t_p}^{t-1} \alpha_p \nabla f_m(\hat{x}_{t_p}) \right\|^2$$

$$\leq 2(t - t_p) \sum_{j=t_p}^{t-1} (\alpha_p)^2 \left(L_0 + L_1 \left\|\nabla f_m(\hat{x}_{t_p})\right\|\right)^2 \left\|x_j^m - \hat{x}_{t_p}\right\|^2$$

$$+ 2 \left\| \sum_{j=t_p}^{t-1} \alpha_p \nabla f_m(\hat{x}_{t_p}) \right\|^2.$$

Averaging, we get

$$
\frac{1}{M} \sum_{m=1}^{M} \left\| x_t^m - \hat{x}_{t_p} \right\|^2 \leq \frac{2(t - t_p)}{M} \sum_{m=1}^{M} \sum_{j=t_p}^{t-1} \left( \alpha_p \right)^2 \left( L_0 + L_1 \left\| \nabla f_m(\hat{x}_{t_p}) \right\| \right)^2 \left\| x_j^m - \hat{x}_{t_p} \right\|^2
$$

$$
+ \frac{2}{M} \sum_{m=1}^{M} \left\| \sum_{j=t_p}^{t-1} \alpha_p \nabla f_m(\hat{x}_{t_p}) \right\|^2
$$

$$
\leq \frac{2(t - t_p)}{M} \left( a_p \right)^2 \sum_{m=1}^{M} \sum_{j=t_p}^{t-1} \left( \alpha_p \right)^2 \left\| x_j^m - \hat{x}_{t_p} \right\|^2
$$

$$
+ \frac{2}{M} \sum_{m=1}^{M} \left\| \sum_{j=t_p}^{t-1} \alpha_p \nabla f_m(\hat{x}_{t_p}) \right\|^2 .
$$

Recall that $\alpha_p \leq \frac{1}{2H \left( L_0 + L_1 \max_m \left\| \nabla f_m(\hat{x}_{t_p}) \right\| \right)}$. Then we have

$$
\frac{1}{M} \sum_{m=1}^{M} \left\| x_t^m - \hat{x}_{t_p} \right\|^2 \leq \frac{t - t_p}{2H^2 M} \sum_{m=1}^{M} \sum_{j=t_p}^{t-1} \left\| x_j^m - \hat{x}_{t_p} \right\|^2
$$

$$
+ \frac{2}{M} \sum_{m=1}^{M} \left\| \sum_{j=t_p}^{t-1} \alpha_p \nabla f_m(\hat{x}_{t_p}) \right\|^2 . \tag{5}
$$

Let us bound the last term:

$$
\frac{2}{M} \sum_{m=1}^{M} \left\| \sum_{j=t_p}^{t-1} \alpha_p \nabla f_m(\hat{x}_{t_p}) \right\|^2 \leq \frac{2}{M} \sum_{m=1}^{M} \left\| \nabla f_m(\hat{x}_{t_p}) \right\|^2 (t - t_p)^2 \alpha_p^2
$$

$$
\leq \frac{4}{M} \sum_{m=1}^{M} \left( L_0 + L_1 \left\| \nabla f_m(\hat{x}_{t_p}) \right\| \right) \left( f_m \left( \hat{x}_{t_p} \right) - f_m^\star \right)
$$

$$
\times (t - t_p)^2 \alpha_p^2
$$

$$
\leq \frac{4 (t - t_p)^2 a_p \alpha_p^2}{M} \sum_{m=1}^{M} \left( f_m \left( \hat{x}_{t_p} \right) - f_m^\star \right)
$$

$$
= 4 (t - t_p)^2 a_p \alpha_p^2 \left( f(\hat{x}_{t_p}) - f^\star + \left( f^\star - \frac{1}{M} \sum_{m=1}^{M} f_m^\star \right) \right) p
$$

$$
= 4 (t - t_p)^2 a_p \alpha_p^2 \left( f(\hat{x}_{t_p}) - f^\star + \Delta^\star \right) .
$$

Further, summing (7) with respect to $t$, we obtain

$$
\begin{aligned}
\frac{1}{M} \sum_{m=1}^{M} \sum_{t=t_p+1}^{v} \left\| x_t^m - \hat{x}_{t_p} \right\|^2 &\leq \frac{1}{2H^2 M} \sum_{m=1}^{M} \sum_{t=t_p+1}^{v} (t - t_p) \sum_{j=t_p}^{t-1} \left\| x_j^m - \hat{x}_{t_p} \right\|^2 \\
&\quad + \sum_{t=t_p+1}^{v} 4 (t - t_p)^2 a_p \alpha_p^2 \left( f(\hat{x}_{t_p}) - f^\star + \Delta^\star \right) \\
&\leq \frac{v - t_p}{2H^2 M} \sum_{m=1}^{M} \sum_{t=t_p+1}^{v} \sum_{j=t_p}^{v} \left\| x_j^m - \hat{x}_{t_p} \right\|^2 \\
&\quad + 4 \sum_{t=t_p+1}^{v} (v - t_p)^2 a_p \alpha_p^2 \left( f(\hat{x}_{t_p}) - f^\star + \Delta^\star \right) \\
&\leq \frac{(v - t_p)^2}{2H^2 M} \sum_{m=1}^{M} \sum_{j=t_p}^{v} \left\| x_j^m - \hat{x}_{t_p} \right\|^2 \\
&\quad + 4 (v - t_p)^3 a_p \alpha_p^2 \left( f(\hat{x}_{t_p}) - f^\star + \Delta^\star \right).
\end{aligned}
$$

Using the fact that $v - t_p \leq H - 1 < H$, we obtain that

$$
\frac{1}{M} \sum_{m=1}^{M} \sum_{t=t_p+1}^{v} \left\| x_t^m - \hat{x}_{t_p} \right\|^2 \leq 8 (v - t_p)^3 a_p \alpha_p^2 \left( f(\hat{x}_{t_p}) - f^\star + \Delta^\star \right).
$$

$\square$

*Proof of Theorem 1.* Applying Lemma 1, we obtain that

$$
f(\hat{x}_{t_{p+1}}) \leq f(\hat{x}_{t_p}) - \gamma_p \langle \nabla f\left(\hat{x}_{t_p}\right), g_p \rangle + \left( L_0 + L_1 \left\| \nabla f\left(\hat{x}_{t_p}\right) \right\| \right) \frac{\gamma_p^2 \left\| g_p \right\|^2}{2}.
$$

Additionally, from the fact that $2\langle a, b \rangle = -\|a - b\|^2 + \|a\|^2 + \|b\|^2$

$$
\begin{aligned}
f(\hat{x}_{t_{p+1}}) &\leq f(\hat{x}_{t_p}) - \gamma_p \langle \nabla f\left(\hat{x}_{t_p}\right), g_p \rangle + \left( L_0 + L_1 \left\| \nabla f\left(\hat{x}_{t_p}\right) \right\| \right) \frac{\gamma_p^2 \left\| g_p \right\|^2}{2} \\
&\leq f(\hat{x}_{t_p}) - \frac{\gamma_p}{2} (-\|\nabla f(\hat{x}_{t_p}) - g_p\|^2 + \|\nabla f(x_p)\|^2 + \|g_p\|^2) \\
&\quad + (L_0 + L_1 \|\nabla f(\hat{x}_{t_p})\|) \frac{\gamma_p^2 \|g_p\|^2}{2} \\
&\leq f(\hat{x}_{t_p}) - \frac{\gamma_p}{2} \|\nabla f(\hat{x}_{t_p})\|^2 + \frac{\gamma_p}{2} \|\nabla f(\hat{x}_{t_p}) - g_p\|^2 + (L_0 + L_1 \|\nabla f(\hat{x}_{t_p})\|) \frac{\gamma_p^2 \|g_p\|^2}{2}.
\end{aligned}
$$

Consider $\frac{\gamma_p}{2} \|\nabla f(\hat{x}_{t_p}) - g_p\|^2$. We have

$$
\begin{aligned}
\frac{\gamma_p}{2} \|\nabla f(\hat{x}_{t_p}) - g_p\|^2 &= \frac{\gamma_p}{2} \left\| \frac{1}{M} \sum_{m=1}^{M} \left( \nabla f_m(\hat{x}_{t_p}) - \frac{1}{v - t_p} \sum_{j=t_p}^{v} \nabla f_m\left(x_j^m\right) \right) \right\|^2 \\
&\leq \frac{\gamma_p}{2} \frac{1}{M(v - t_p)} \sum_{m=1}^{M} (L_0 + L_1 \|\nabla f_m(\hat{x}_{t_p})\|)^2 \sum_{j=t_p}^{v} \|x_j^m - \hat{x}_{t_p}\|^2 \\
&\leq \frac{\gamma_p}{2(v - t_p)} (L_0 + L_1 \max_m \|\nabla f_m(\hat{x}_{t_p})\|)^2 \frac{1}{M} \sum_{m=1}^{M} \sum_{j=t_p}^{v} \|x_j^m - \hat{x}_{t_p}\|^2 \\
&= \frac{\gamma_p a_p^2}{2(v - t_p)} \frac{1}{M} \sum_{m=1}^{M} \sum_{j=t_p}^{v} \|x_j^m - \hat{x}_{t_p}\|^2.
\end{aligned}
$$

Notice that

$$
\begin{aligned}
\frac{\gamma_p^2 \|g_p\|^2}{2} &= \frac{\gamma_p^2}{2} \left\| \frac{1}{M(v - t_p)} \sum_{m=1}^{M} \sum_{j=t_p+1}^{v} \nabla f_m \left( x_j^m \right) \right\|^2 \\
&\leq \frac{\gamma_p^2}{(v - t_p)^2} \left\| \frac{1}{M} \sum_{m=1}^{M} \sum_{j=t_p+1}^{v} \left( \nabla f_m \left( x_j^m \right) - \nabla f_m \left( \hat{x}_{t_p} \right) \right) \right\|^2 \\
&\quad + \frac{\gamma_p^2}{(v - t_p)^2} \left\| \frac{1}{M} \sum_{m=1}^{M} \sum_{j=t_p+1}^{v} \nabla f_m \left( \hat{x}_{t_p} \right) \right\|^2 \\
&\leq \frac{\gamma_p^2}{(v - t_p)} \left( L_0 + L_1 \max_m \left\| \nabla f_m(\hat{x}_{t_p}) \right\| \right)^2 \frac{1}{M} \sum_{m=1}^{M} \sum_{j=t_p}^{v} \left\| x_j^m - \hat{x}_{t_p} \right\|^2 \\
&\quad + \gamma_p^2 \left\| \nabla f \left( \hat{x}_{t_p} \right) \right\|^2 \\
&= \frac{\gamma_p^2 a_p^2}{M(v - t_p)} \sum_{m=1}^{M} \sum_{j=t_p}^{v} \left\| x_j^m - \hat{x}_{t_p} \right\|^2 + \gamma_p^2 \left\| \nabla f \left( \hat{x}_{t_p} \right) \right\|^2 .
\end{aligned}
$$

Therefore, we obtain

$$
\begin{aligned}
f \left( \hat{x}_{t_{p+1}} \right) &\leq f(\hat{x}_{t_p}) - \frac{\gamma_p}{2} \|\nabla f(\hat{x}_{t_p})\|^2 + \frac{\gamma_p a_p^2}{2M(v - t_p)} \sum_{m=1}^{M} \sum_{j=t_p}^{v} \left\| x_j^m - \hat{x}_{t_p} \right\|^2 + \frac{\hat{a}_p \gamma_p^2 \|g_p\|^2}{2} \\
&\leq f(\hat{x}_{t_p}) - \frac{\gamma_p}{2} \|\nabla f(\hat{x}_{t_p})\|^2 + \frac{\gamma_p a_p^2}{2M(v - t_p)} \sum_{m=1}^{M} \sum_{j=t_p}^{v} \left\| x_j^m - \hat{x}_{t_p} \right\|^2 \\
&\quad + \frac{\hat{a}_p a_p^2 \gamma_p^2}{M(v - t_p)} \sum_{m=1}^{M} \sum_{j=t_p}^{v} \left\| x_j^m - \hat{x}_{t_p} \right\|^2 + \hat{a}_p \gamma_p^2 \left\| \nabla f \left( \hat{x}_{t_p} \right) \right\|^2 \\
&= f(\hat{x}_{t_p}) + \left( \hat{a}_p \gamma_p^2 - \frac{\gamma_p}{2} \right) \left\| \nabla f \left( \hat{x}_{t_p} \right) \right\|^2 \\
&\quad + \left( \hat{a}_p a_p^2 \gamma_p^2 + \frac{\gamma_p a_p^2}{2} \right) \frac{1}{M(v - t_p)} \sum_{m=1}^{M} \sum_{j=t_p}^{v} \left\| x_j^m - \hat{x}_{t_p} \right\|^2 .
\end{aligned}
$$

Recall that $\gamma_p \leq \frac{1}{4\hat{a}_p}$. Then, using Lemma 4, we have

$$
\begin{aligned}
f \left( \hat{x}_{t_{p+1}} \right) &\leq f \left( \hat{x}_{t_p} \right) - \frac{\gamma_p}{4} \left\| \nabla f \left( \hat{x}_{t_p} \right) \right\|^2 \\
&\quad + \left( \hat{a}_p a_p^2 \gamma_p^2 + \frac{\gamma_p a_p^2}{2} \right) \frac{1}{M(v - t_p)} \sum_{m=1}^{M} \sum_{j=t_p}^{v} \left\| x_j^m - \hat{x}_{t_p} \right\|^2 \\
&\leq f \left( \hat{x}_{t_p} \right) - \frac{\gamma_p}{4} \left\| \nabla f \left( \hat{x}_{t_p} \right) \right\|^2 \\
&\quad + \left( \hat{a}_p a_p^2 \gamma_p^2 + \frac{\gamma_p a_p^2}{2} \right) 8(v - t_p)^2 a_p \alpha_p^2 \left( f(\hat{x}_{t_p}) - f^\star + \Delta^\star \right) \\
&\leq f \left( \hat{x}_{t_p} \right) - \frac{\gamma_p}{4} \left\| \nabla f \left( \hat{x}_{t_p} \right) \right\|^2 + \frac{3(H - 1)^2 a_p^3 \alpha_p^2 \left( f(\hat{x}_{t_p}) - f^\star + \Delta^\star \right)}{2\hat{a}_p} .
\end{aligned}
$$

Let us rewrite the inequality in the following way:

$$
\frac{\gamma_p}{4} \left\| \nabla f \left( \hat{x}_{t_p} \right) \right\|^2 \leq f \left( \hat{x}_{t_p} \right) - f \left( \hat{x}_{t_{p+1}} \right) + \frac{3(H - 1)^2 a_p^3 \alpha_p^2 \left( f(\hat{x}_{t_p}) - f^\star + \Delta^\star \right)}{2\hat{a}_p} . \tag{6}
$$

Since $\gamma_p \geq \frac{\zeta}{\hat{a}_p}$, we get that

$$\frac{\gamma_p \left\| \nabla f \left( \hat{x}_{t_p} \right) \right\|^2}{4} \geq \frac{\zeta \left\| \nabla f \left( \hat{x}_{t_p} \right) \right\|^2}{4\hat{a}_p}.$$

Therefore,

$$\frac{\gamma_p}{4} \|\nabla f(\hat{x}_{t_p})\|^2 \geq \begin{cases} \frac{\zeta \|\nabla f(\hat{x}_{t_p})\|^2}{8L_0}, & \|\nabla f(\hat{x}_{t_p})\| \leq \frac{L_0}{L_1}, \\ \frac{\zeta \|\nabla f(\hat{x}_{t_p})\|}{8L_1}, & \|\nabla f(\hat{x}_{t_p})\| > \frac{L_0}{L_1}, \end{cases} = \frac{\zeta}{8} \min \left\{ \frac{\|\nabla f(\hat{x}_{t_p})\|^2}{L_0}, \frac{\|\nabla f(\hat{x}_{t_p})\|}{L_1} \right\}.$$

Denote $\delta_p \stackrel{\text{def}}{=} f \left( \hat{x}_{t_p} \right) - f^\star$. Then we have

$$\frac{\zeta}{8} \min \left\{ \frac{\|\nabla f(\hat{x}_{t_p})\|^2}{L_0}, \frac{\|\nabla f(\hat{x}_{t_p})\|}{L_1} \right\} \leq \delta_p - \delta_{p+1} + \frac{3(H-1)^2 \alpha_p^2 a_p^3 (\delta_p + \Delta^\star)}{2\hat{a}_p}.$$

Let $\alpha_p \leq \frac{1}{ca_p} \sqrt{\frac{\hat{a}_p}{a_p}}$, where $c \geq \sqrt{P}$. Applying the result of Mishchenko et al. (2020, Lemma 6), we appear at

$$\min_{0 \leq p \leq P-1} \left\{ \frac{\zeta}{8} \min \left\{ \frac{\|\nabla f(\hat{x}_{t_p})\|^2}{L_0}, \frac{\|\nabla f(\hat{x}_{t_p})\|}{L_1} \right\} \right\} \leq \frac{\left( 1 + \frac{3(H-1)^2 \alpha_p^2 a_p^3}{2\hat{a}_p} \right)^P}{P} \delta_0$$
$$+ \frac{3(H-1)^2 \alpha_p^2 a_p^3}{2\hat{a}_p} \Delta^\star.$$

$\square$

**Corollary 1.** *Fix $\varepsilon > 0$. Choose $c = \sqrt{3(H-1)^2 P}$. Let $\alpha_p \leq 2\sqrt{\frac{\hat{a}_p \delta_0}{3P(H-1)^2 a_p^3 \Delta^\star}}$. Then, if $P \geq \frac{32\delta_0}{\zeta\varepsilon}$, we have $\min_{0 \leq p \leq P-1} \left\{ \min \left\{ \frac{\|\nabla f(\hat{x}_{t_p})\|^2}{L_0}, \frac{\|\nabla f(\hat{x}_{t_p})\|}{L_1} \right\} \right\} \leq \varepsilon$.*

*Proof of Corollary 1.* Since $c = \sqrt{3(H-1)^2 P}$ and $\alpha_p \leq \frac{1}{ca_p} \sqrt{\frac{\hat{a}_p}{a_p}}$, $\alpha_p \leq 2\sqrt{\frac{\hat{a}_p \delta_0}{3P(H-1)^2 a_p^3 \Delta^\star}}$, due to the choice of $P \geq \frac{32\delta_0}{\zeta\varepsilon}$, we obtain that

$$\frac{\left( 1 + \frac{3(H-1)^2 \alpha_p^2 a_p^3}{2\hat{a}_p} \right)^P}{P} \delta_0 \leq \frac{\sqrt{e}\delta_0}{P} \leq \frac{2\delta_0}{P} \leq \frac{\zeta\varepsilon}{16},$$

and that

$$\frac{3(H-1)^2 \alpha_p^2 a_p^3}{2\hat{a}_p} \Delta^\star \leq \frac{\zeta\varepsilon}{16}.$$

Therefore, $\min_{0 \leq p \leq P-1} \left\{ \min \left\{ \frac{\|\nabla f(\hat{x}_{t_p})\|^2}{L_0}, \frac{\|\nabla f(\hat{x}_{t_p})\|}{L_1} \right\} \right\} \leq \varepsilon$. $\square$

## B.2 ASYMMETRIC GENERALIZED-SMOOTH FUNCTIONS UNDER PŁ-CONDITION

**Theorem 2** (Asymmetric generalized-smooth convergence analysis of Algorithm 1 in PŁ-case). *Let Assumptions 1 and 2 hold for functions $f$ and $\{f_m\}_{m=1}^M$. Let Assumption 4 hold. Choose $0 < \zeta \leq \frac{1}{4}$. Let $\delta_0 \stackrel{\text{def}}{=} f(x_0) - f^\star$. Choose any integer $P > \frac{64\delta_0 L_1^2}{\mu\zeta}$. For all $0 \leq p \leq P-1$, denote*

$$\hat{a}_p = L_0 + L_1 \|\nabla f(\hat{x}_{t_p})\|, \quad a_p = L_0 + L_1 \max_m \|\nabla f_m(\hat{x}_{t_p})\|, \quad 1 \leq t_{p+1} - t_p \leq H.$$

*Put $\Delta^\star = f^\star - \frac{1}{M} \sum_{m=1}^M f_m^\star$. Impose the following conditions on the local stepsizes $\alpha_p$ and server stepsizes $\gamma_p$:*

$$\alpha_p \leq \min \left\{ \frac{1}{2Ha_p}, \frac{1}{ca_p} \sqrt{\frac{\hat{a}_p}{a_p}}, \sqrt{\frac{\mu\zeta\hat{a}_p}{48L_1^2(H-1)^2 a_p^3 \left( f(\hat{x}_{t_p}) - f^\star + \Delta^\star \right)}}, \right.$$
$$\left. \sqrt{\frac{2\delta_0 \hat{a}_p}{3P(H-1)^2 a_p^3 \left( f(\hat{x}_{t_p}) - f^\star + \Delta^\star \right)}} \right\},$$

$$\frac{\zeta}{\hat{a}_p} \le \gamma_p \le \frac{1}{4\hat{a}_p}, \quad 0 \le p \le P - 1,$$

where $c \ge \sqrt{P}$. Let $\tilde{P}$ be an integer such that $0 \le \tilde{P} \le \frac{64\delta_0 L_1^2}{\mu\zeta}$, $A > 0$ be a constant, $\alpha \le \sqrt{\frac{\delta_0}{AP}}$. Then, the iterates $\left\{\hat{x}_{t_p}\right\}_{p=0}^{P}$ of Algorithm 1 satisfy

$$\delta_P \le \left(1 - \frac{\mu\zeta}{4L_0}\right)^{P-\tilde{P}} \delta_0 + \frac{4L_0 A\alpha^2}{\mu\zeta},$$

where $\delta_P \overset{def}{=} f\left(\hat{x}_{t_P}\right) - f^\star$.

*Proof of Theorem 2.* Let us follow the first steps of the proof of Theorem 1. Consider (6):

$$\frac{\gamma_p}{4}\left\|\nabla f\left(\hat{x}_{t_p}\right)\right\|^2 \le f\left(\hat{x}_{t_p}\right) - f\left(\hat{x}_{t_{p+1}}\right) + \frac{3(H-1)^2 a_p^3 \alpha_p^2 \left(f(\hat{x}_{t_p}) - f^\star + \Delta^\star\right)}{2\hat{a}_p}.$$

Since $\gamma_p \ge \frac{\zeta}{\hat{a}_p}$, and $f$ satisfies Polyak–Łojasiewicz Assumption 4, we obtain that

$$\frac{\mu\zeta\left(f(\hat{x}_{t_p}) - f^\star\right)}{2\hat{a}_p} \le f\left(\hat{x}_{t_p}\right) - f\left(\hat{x}_{t_{p+1}}\right) + \frac{3(H-1)^2 a_p^3 \alpha_p^2 \left(f(\hat{x}_{t_p}) - f^\star + \Delta^\star\right)}{2\hat{a}_p}.$$

**1.** Let $\tilde{P}$ be the number of steps $p$, so that $\left\|\nabla f\left(\hat{x}_{t_p}\right)\right\| \ge \frac{L_0}{L_1}$. For such $p$, we have $L_0 + L_1\left\|\nabla f\left(\hat{x}_{t_p}\right)\right\| = \hat{a}_p \le 2L_1\left\|\nabla f\left(\hat{x}_{t_p}\right)\right\|$. Therefore, we get

$$\frac{\mu\zeta\left(f(\hat{x}_{t_p}) - f^\star\right)}{4L_1\left\|\nabla f\left(\hat{x}_{t_p}\right)\right\|} \le f\left(\hat{x}_{t_p}\right) - f\left(\hat{x}_{t_{p+1}}\right) + \frac{3(H-1)^2 a_p^3 \alpha_p^2 \left(f(\hat{x}_{t_p}) - f^\star + \Delta^\star\right)}{2\hat{a}_p}.$$

Notice that the relation $\hat{a}_p \le 2L_1\left\|\nabla f\left(\hat{x}_{t_p}\right)\right\|$ and Lemma 1 together imply

$$\frac{\left\|\nabla f\left(\hat{x}_{t_p}\right)\right\|}{4L_1} \le \frac{\left\|\nabla f\left(\hat{x}_{t_p}\right)\right\|^2}{2\hat{a}_p} \le f\left(\hat{x}_{t_p}\right) - f^\star.$$

Hence, we have

$$\frac{\mu\zeta}{16L_1^2} \le f\left(\hat{x}_{t_p}\right) - f\left(\hat{x}_{t_{p+1}}\right) + \frac{3(H-1)^2 a_p^3 \alpha_p^2 \left(f(\hat{x}_{t_p}) - f^\star + \Delta^\star\right)}{2\hat{a}_p}.$$

Subtracting $f^\star$ on both sides and introducing $\delta_p \overset{def}{=} f\left(\hat{x}_{t_p}\right) - f^\star$, we obtain

$$\delta_{p+1} \le \delta_p - \frac{\mu\zeta}{16L_1^2} + \frac{3(H-1)^2 a_p^3 \alpha_p^2 \left(f(\hat{x}_{t_p}) - f^\star + \Delta^\star\right)}{2\hat{a}_p}.$$

As $\alpha_p \le \sqrt{\frac{\mu\zeta\hat{a}_p}{48L_1^2(H-1)^2 a_p^3\left(f(\hat{x}_{t_p}) - f^\star + \Delta^\star\right)}}$, it follows that $\frac{3(H-1)^2 a_p^3 \alpha_p^2\left(f(\hat{x}_{t_p}) - f^\star + \Delta^\star\right)}{2\hat{a}_p} \le \frac{\mu\zeta}{32L_1^2}$. Therefore, we get

$$\delta_{p+1} \le \delta_p - \frac{\mu\zeta}{32L_1^2}.$$

**2.** Suppose now that $\left\|\nabla f\left(\hat{x}_{t_p}\right)\right\| \le \frac{L_0}{L_1}$. For such $p$, we have $L_0 + L_1\left\|\nabla f\left(\hat{x}_{t_p}\right)\right\| = \hat{a}_p \le 2L_0$. Hence,

$$\frac{\mu\zeta\left(f(\hat{x}_{t_p}) - f^\star\right)}{4L_0} \le f\left(\hat{x}_{t_p}\right) - f\left(\hat{x}_{t_{p+1}}\right) + \frac{3(H-1)^2 a_p^3 \alpha_p^2 \left(f(\hat{x}_{t_p}) - f^\star + \Delta^\star\right)}{2\hat{a}_p}.$$

Subtracting $f^\star$ on both sides and introducing $\delta_p \overset{def}{=} f\left(\hat{x}_{t_p}\right) - f^\star$, we obtain

$$\delta_{p+1} \le \delta_p \rho + \frac{3(H-1)^2 a_p^3 \alpha_p^2 \left(f(\hat{x}_{t_p}) - f^\star + \Delta^\star\right)}{2\hat{a}_p}, \quad \text{where } \rho \overset{def}{=} 1 - \frac{\mu\zeta}{4L_0}.$$

Let $\alpha_p \stackrel{\text{def}}{=} \alpha\hat{\alpha}_p$ and $\hat{\alpha}_p \leq \sqrt{\frac{2A\hat{a}_p}{3(H-1)^2 a_p^3\left(f(\hat{x}_{t_p})-f^\star+\Delta^\star\right)}}$ for some constant $A > 0$. Then,

$$\delta_{p+1} \leq \rho\delta_p + A\alpha^2.$$

Unrolling the recursion, we derive

$$\delta_P \leq \rho^{P-\tilde{P}}\delta_0 + A\alpha^2 \sum_{i=0}^{\infty} \rho^i - \frac{\mu\zeta}{32L_1^2} \sum_{i=0}^{N-1} \rho^i$$

$$\leq \rho^{P-\tilde{P}}\delta_0 + \frac{A\alpha^2}{1-\rho} - \frac{1-\rho^{\tilde{P}}}{1-\rho}\frac{\mu\zeta}{32L_1^2}.$$

Notice that $\delta_{p+1} \leq \delta_p + A\alpha^2$, which implies

$$\delta_P \leq \delta_0 + \left(P - \tilde{P}\right) A\alpha^2 - \tilde{P}\frac{\mu\zeta}{32L_1^2}.$$

Since $\alpha \leq \sqrt{\frac{\delta_0}{AP}}$, we conclude that

$$0 \leq \delta_P \leq 2\delta_0 - \tilde{P}\frac{\mu\zeta}{32L_1^2}, \quad \Rightarrow \tilde{P} \leq \frac{64\delta_0 L_1^2}{\mu\zeta}.$$

Therefore, for $P > \frac{64\delta_0 L_1^2}{\mu\zeta}$ we can guarantee that $P - \tilde{P} > 0$ and

$$\delta_P \leq \rho^{P-\tilde{P}}\delta_0 + \frac{A\alpha^2}{1-\rho} - \tilde{P}\rho^{\tilde{P}}\frac{\mu\zeta}{32L_1^2}$$

$$\leq \rho^{P-\tilde{P}}\delta_0 + \frac{A\alpha^2}{1-\rho}.$$

$\square$

**Corollary 2.** *Fix $\varepsilon > 0$. Choose $\alpha \leq \min\left\{\sqrt{\frac{\delta_0}{AP}}, L_1\sqrt{\frac{8\delta_0\varepsilon}{L_0 AP}}\right\}$. Then, if $P \geq \frac{64\delta_0 L_1^2}{\mu\zeta} + \frac{4L_0}{\mu\zeta}\ln\frac{2\delta_0}{\varepsilon}$, we have $\delta_P \leq \varepsilon$.*

*Proof of Corollary 2.* Since $0 \leq \tilde{P} \leq \frac{64\delta_0 L_1^2}{\mu\zeta}$, $A > 0$, $\alpha \leq \sqrt{\frac{\delta_0}{AP}}$, $\alpha \leq L_1\sqrt{\frac{8\delta_0\varepsilon}{L_0 AP}}$, due to the choice of $P \geq \frac{64\delta_0 L_1^2}{\mu\zeta} + \frac{4L_0}{\mu\zeta}\ln\frac{2\delta_0}{\varepsilon}$, we obtain that

$$\left(1 - \frac{\mu\zeta}{4L_0}\right)^{P-\tilde{P}}\delta_0 \leq e^{-\frac{\mu\zeta}{4L_0}(P-\tilde{P})}\delta_0 \leq \frac{\varepsilon}{2},$$

and that

$$\frac{4L_0 A}{\mu\zeta} \cdot \frac{\delta_0}{AP} \leq \frac{\varepsilon}{2}.$$

Therefore, $\delta_P \leq \varepsilon$.

$\square$

### B.3 SYMMETRIC GENERALIZED-SMOOTH NON-CONVEX FUNCTIONS

**Theorem 5.** *Let Assumptions 1 and 2 hold for functions $f$ and $\{f_m\}_{m=1}^M$. Choose any $P \geq 1$. For all $0 \leq p \leq P - 1$, denote*

$$\hat{a}_p = L_0 + L_1\|\nabla f(\hat{x}_{t_p})\|, \quad a_p = L_0 + L_1\max_m\|\nabla f_m(\hat{x}_{t_p})\|, \quad 1 \leq t_{p+1} - t_p \leq H.$$

*Put $\Delta^\star = f^\star - \frac{1}{M}\sum_{m=1}^M f_m^\star$. Impose the following conditions on the local stepsizes $\alpha_p$ and server stepsizes $\gamma_p$:*

$$\alpha_p \leq \min\left\{\frac{1}{2Ha_p}, \frac{1}{ca_p}\sqrt{\frac{\hat{a}_p}{a_p}}, \frac{\hat{a}_p C}{a_p\left(L_0 + L_1 A_{t_p}\right)}\right\}, \quad \frac{\zeta}{\hat{a}_p} \leq \gamma_p \leq \frac{1}{8\hat{a}_p}, \quad 0 \leq p \leq P - 1,$$

where $0 < \zeta \leq \frac{1}{8}$, $c \geq \sqrt{P}$, $C \leq \frac{\ln 1.5}{H}$, $A_{t_p} \stackrel{def}{=} \max\left\{ \sqrt{\frac{2L_0 M(\delta_{t_p} + \Delta^\star)}{\nu}}, \frac{2L_1 M(\delta_{t_p} + \Delta^\star)}{\nu} \right\}$, $\nu$ such

that $\nu \exp \nu = 1$. Let $\delta_0 \stackrel{def}{=} f(x_0) - f^\star$. Then, the iterates $\{\hat{x}_{t_p}\}_{p=0}^{P-1}$ of Algorithm 1 satisfy

$$\min_{0 \leq p \leq P-1} \left\{ \frac{\zeta}{8} \min \left\{ \frac{\|\nabla f(\hat{x}_{t_p})\|^2}{L_0}, \frac{\|\nabla f(\hat{x}_{t_p})\|}{L_1} \right\} \right\} \leq \frac{\left(1 + \frac{7(H-1)^2 \alpha_p^2 a_p^3}{\hat{a}_p}\right)^P}{P} \delta_0$$
$$+ \frac{7(H-1)^2 \alpha_p^2 a_p^3}{\hat{a}_p} \Delta^\star.$$

Let us remind that $\hat{a}_p = L_0 + L_1 \|\nabla f(\hat{x}_{t_p})\|$, $a_p = L_0 + L_1 \max_m \|\nabla f_m(\hat{x}_{t_p})\|$ and $\Delta^\star = f^\star - \frac{1}{M}\sum_{m=1}^{M} f_m^\star$. Put $v_p \stackrel{def}{=} t_{p+1} - 1$.

**Lemma 5.** *Assume that $f$ and each $f_m$ satisfy Assumptions 1 and 3. Then we have the following bound:*

$$\frac{1}{M} \sum_{m=1}^{M} \sum_{t=t_p+1}^{v} \left\| x_t^m - \hat{x}_{t_p} \right\|^2 \leq 32 (v - t_p)^3 a_p \alpha_p^2 \left( f(\hat{x}_{t_p}) - f^\star + \Delta^\star \right).$$

*Proof of Lemma 5.* We have

$$\left\| x_t^m - \hat{x}_{t_p} \right\|^2 = \left\| \sum_{j=t_p}^{t-1} \alpha_p \nabla f_m \left( x_j^m \right) \right\|^2$$

$$\leq 2 \left\| \sum_{j=t_p}^{t-1} \alpha_p \left( \nabla f_m(x_j^m) - \nabla f_m(\hat{x}_{t_p}) \right) \right\|^2 + 2 \left\| \sum_{j=t_p}^{t-1} \alpha_p \nabla f_m(\hat{x}_{t_p}) \right\|^2$$

$$\leq 2(t - t_p) \sum_{j=t_p}^{t-1} (\alpha_p)^2 \left( L_0 + L_1 \left\| \nabla f_m(\hat{x}_{t_p}) \right\| \right)^2$$

$$\times \exp\left\{ L_1 \left\| x_j^m - \hat{x}_{t_p} \right\| \right\} \left\| x_j^m - \hat{x}_{t_p} \right\|^2 + 2 \left\| \sum_{j=t_p}^{t-1} \alpha_p \nabla f_m(\hat{x}_{t_p}) \right\|^2$$

$$\leq 2(t - t_p) \sum_{j=t_p}^{t-1} (\alpha_p)^2 \left( L_0 + L_1 \left\| \nabla f_m(\hat{x}_{t_p}) \right\| \right)^2$$

$$\times \exp\left\{ L_1 \left\| \sum_{\ell=t_p}^{j-1} \alpha_p \nabla f_m \left( x_\ell^m \right) \right\| \right\} \left\| x_j^m - \hat{x}_{t_p} \right\|^2 + 2 \left\| \sum_{j=t_p}^{t-1} \alpha_p \nabla f_m(\hat{x}_{t_p}) \right\|^2.$$

Let us show that if $\alpha_p \leq \frac{1}{2\left(L_0 + L_1 \max_m \max_{t_p \leq \ell \leq t_{p+1}} \|\nabla f_m(x_\ell^m)\|\right)}$, then $f_m(x_\ell^m) \leq f_m(x_{t_p}^m)$ for $t_p \leq \ell \leq t_{p+1} - 1$. Notice that locally we perform the iterations of the gradient descent. It means, that

$$f_m(x_{\ell+1}^m) \leq f_m(x_\ell^m) - \alpha_p \|\nabla f_m(x_\ell^m)\|^2 + \frac{L_0 + L_1 \|\nabla f_m(x_\ell^m)\|}{2} \exp\left\{ L_1 \alpha_p \|\nabla f_m(x_\ell^m)\| \right\} \alpha_p^2 \|\nabla f_m(x_\ell^m)\|^2$$

$$\leq f_m(x_\ell^m) - \alpha_p \|\nabla f_m(x_\ell^m)\|^2 + \frac{\alpha_p}{2} \|\nabla f_m(x_\ell^m)\|^2 \exp\left\{ \alpha_p \left( L_0 + L_1 \|\nabla f_m(x_\ell^m)\| \right) \right\}$$

$$= f_m(x_\ell^m) - \alpha_p \left( 1 - \frac{\sqrt{3}}{2} \right) \|\nabla f_m(x_\ell^m)\|^2.$$

Then $f_m(x_\ell^m) \leq f_m(x_{t_p}^m) = f_m(\hat{x}_{t_p})$ for $t_p \leq \ell \leq t_{p+1} - 1$ follows. Therefore, for such $\alpha_p$ we have that

$$f_m(x_\ell^m) - f_m^\star \leq f_m(x_{t_p}^m) - f_m^\star \leq \sum_{m=1}^{M} \left( f_m(x_{t_p}^m) - f_m^\star \right) = M\delta_{t_p} + M\Delta^\star.$$

From Lemma 2 we have

$$\min\left\{\frac{\nu\|\nabla f_m(x_\ell^m)\|^2}{L_0}, \frac{\nu\|\nabla f_m(x_\ell^m)\|}{L_1}\right\} \le \frac{\nu\|\nabla f_m(x_\ell^m)\|^2}{2(L_0 + L_1\|\nabla f_m(x_\ell^m)\|)} \le f_m(x_\ell^m) - f_m^\star$$

$$\le M\left(\delta_{t_p} + \Delta^\star\right).$$

For every $t_p \le \ell \le t_{p+1} - 1$, for every $m$, we establish

$$\|\nabla f_m(x_\ell^m)\| \le \max\left\{\sqrt{\frac{2L_0 M\left(\delta_{t_p} + \Delta^\star\right)}{\nu}}, \frac{2L_1 M\left(\delta_{t_p} + \Delta^\star\right)}{\nu}\right\} \overset{\text{def}}{=} A_{t_p}.$$

Let us choose $\alpha_p \le \frac{C}{L_0 + L_1 A_{t_p}}$ for some $C \le \frac{\ln 1.5}{H}$ and show by induction that for such local processes $\max_m \|\nabla f_m(x_\ell^m)\| \le A_{t_p}$, for all $t_p \le \ell \le t_{p+1} - 1$. Indeed, for $\ell = t_p$ it holds trivially. Suppose it holds for all $\ell$ such that $t_p \le \ell \le \ell'$ for some $\ell'$. Then, $f_m\left(x_{\ell'+1}^m\right) \le f_m\left(x_{\ell'}^m\right)$ holds for any $\alpha_p \le \frac{C}{L_0 + L_1\|\nabla f_m(x_{\ell'})\|}$, including the chosen stepsize. Hence, $f_m\left(x_{\ell'+1}^m\right) \le f_m(x_{t_p})$. Therefore, $\max_m \|\nabla f_m(x_\ell^m)\| \le A_{t_p}$, for all $t_p \le \ell \le t_{p+1} - 1$. Then, $\frac{C}{L_0 + L_1 A_{t_p}} \le \frac{1}{2\left(L_0 + L_1 \max_m \max_{t_p \le \ell \le t_{p+1}} \|\nabla f_m(x_\ell^m)\|\right)}$.

It means that $\exp\left\{L_1 \left\|\sum_{\ell=t_p}^{j-1} \alpha_p \nabla f_m\left(x_\ell^m\right)\right\|\right\} \le e^{\ln 1.5} = 1.5$.

Averaging, we get

$$\frac{1}{M}\sum_{m=1}^{M}\left\|x_t^m - \hat{x}_{t_p}\right\|^2 \le \frac{3(t - t_p)}{M}\sum_{m=1}^{M}\sum_{j=t_p}^{t-1}(\alpha_p)^2\left(L_0 + L_1\left\|\nabla f_m(\hat{x}_{t_p})\right\|\right)^2\left\|x_j^m - \hat{x}_{t_p}\right\|^2$$

$$+ \frac{2}{M}\sum_{m=1}^{M}\left\|\sum_{j=t_p}^{t-1}\alpha_p\nabla f_m(\hat{x}_{t_p})\right\|^2$$

$$\le \frac{3(t - t_p)}{M}(a_p)^2\sum_{m=1}^{M}\sum_{j=t_p}^{t-1}(\alpha_p)^2\left\|x_j^m - \hat{x}_{t_p}\right\|^2$$

$$+ \frac{2}{M}\sum_{m=1}^{M}\left\|\sum_{j=t_p}^{t-1}\alpha_p\nabla f_m(\hat{x}_{t_p})\right\|^2.$$

Recall that $\alpha_p \le \frac{1}{2Ha_p}$. Then we have

$$\frac{1}{M}\sum_{m=1}^{M}\left\|x_t^m - \hat{x}_{t_p}\right\|^2 \le \frac{1.5(t - t_p)}{2H^2 M}\sum_{m=1}^{M}\sum_{j=t_p}^{t-1}\left\|x_j^m - \hat{x}_{t_p}\right\|^2 + \frac{2}{M}\sum_{m=1}^{M}\left\|\sum_{j=t_p}^{t-1}\alpha_p\nabla f_m(\hat{x}_{t_p})\right\|^2.$$

$$(7)$$

Let us bound the last term, using Lemma 2 and the fact that $1/\eta \le 2$ :

$$\frac{2}{M} \sum_{m=1}^{M} \left\| \sum_{j=t_p}^{t-1} \alpha_p \nabla f_m(\hat{x}_{t_p}) \right\|^2 \le \frac{2}{M} \sum_{m=1}^{M} \left\| \nabla f_m(\hat{x}_{t_p}) \right\|^2 (t-t_p)^2 \alpha_p^2$$

$$\le \frac{8}{M} \sum_{m=1}^{M} \left( L_0 + L_1 \left\| \nabla f_m(\hat{x}_{t_p}) \right\| \right) \left( f_m \left( \hat{x}_{t_p} \right) - f_m^\star \right)$$

$$\times (t-t_p)^2 \alpha_p^2$$

$$\le \frac{8 (t-t_p)^2 a_p \alpha_p^2}{M} \sum_{m=1}^{M} \left( f_m \left( \hat{x}_{t_p} \right) - f_m^\star \right)$$

$$= 8 (t-t_p)^2 a_p \alpha_p^2 \left( f(\hat{x}_{t_p}) - f^\star + \left( f^\star - \frac{1}{M} \sum_{m=1}^{M} f_m^\star \right) \right) p$$

$$= 8 (t-t_p)^2 a_p \alpha_p^2 \left( f(\hat{x}_{t_p}) - f^\star + \Delta^\star \right).$$

Further, summing (7) with respect to $t$, we obtain

$$\frac{1}{M} \sum_{m=1}^{M} \sum_{t=t_p+1}^{v} \left\| x_t^m - \hat{x}_{t_p} \right\|^2 \le \frac{1.5}{2H^2 M} \sum_{m=1}^{M} \sum_{t=t_p+1}^{v} (t-t_p) \sum_{j=t_p}^{t-1} \left\| x_j^m - \hat{x}_{t_p} \right\|^2$$

$$+ \sum_{t=t_p+1}^{v} 8 (t-t_p)^2 a_p \alpha_p^2 \left( f(\hat{x}_{t_p}) - f^\star + \Delta^\star \right)$$

$$\le \frac{1.5 (v-t_p)}{2H^2 M} \sum_{m=1}^{M} \sum_{t=t_p+1}^{v} \sum_{j=t_p}^{v} \left\| x_j^m - \hat{x}_{t_p} \right\|^2$$

$$+ 8 \sum_{t=t_p+1}^{v} (v-t_p)^2 a_p \alpha_p^2 \left( f(\hat{x}_{t_p}) - f^\star + \Delta^\star \right)$$

$$\le \frac{1.5 (v-t_p)^2}{2H^2 M} \sum_{m=1}^{M} \sum_{j=t_p}^{v} \left\| x_j^m - \hat{x}_{t_p} \right\|^2$$

$$+ 8 (v-t_p)^3 a_p \alpha_p^2 \left( f(\hat{x}_{t_p}) - f^\star + \Delta^\star \right).$$

Using the fact that $v - t_p \le H - 1 < H$, we obtain that

$$\frac{1}{M} \sum_{m=1}^{M} \sum_{t=t_p+1}^{v} \left\| x_t^m - \hat{x}_{t_p} \right\|^2 \le 32 (v-t_p)^3 a_p \alpha_p^2 \left( f(\hat{x}_{t_p}) - f^\star + \Delta^\star \right).$$

$\square$

*Proof of Theorem 5.* Applying Lemma 1, we obtain that

$$f(\hat{x}_{t_{p+1}}) \le f(\hat{x}_{t_p}) - \gamma_p \langle \nabla f \left( \hat{x}_{t_p} \right), g_p \rangle + \left( L_0 + L_1 \left\| \nabla f \left( \hat{x}_{t_p} \right) \right\| \right) \exp \left\{ L_1 \gamma_p \left\| g_p \right\| \right\} \frac{\gamma_p^2 \left\| g_p \right\|^2}{2}.$$

Additionally, from the fact that $2 \langle a, b \rangle = - \|a-b\|^2 + \|a\|^2 + \|b\|^2$

$$f(\hat{x}_{t_{p+1}}) \le f(\hat{x}_{t_p}) - \gamma_p \langle \nabla f \left( \hat{x}_{t_p} \right), g_p \rangle + \left( L_0 + L_1 \left\| \nabla f \left( \hat{x}_{t_p} \right) \right\| \right) \exp \left\{ L_1 \gamma_p \left\| g_p \right\| \right\} \frac{\gamma_p^2 \left\| g_p \right\|^2}{2}$$

$$\le f(\hat{x}_{t_p}) - \frac{\gamma_p}{2} (- \| \nabla f(\hat{x}_{t_p}) - g_p \|^2 + \| \nabla f(x_p) \|^2 + \| g_p \|^2)$$

$$+ (L_0 + L_1 \| \nabla f(\hat{x}_{t_p}) \|) \exp \left\{ L_1 \gamma_p \left\| g_p \right\| \right\} \frac{\gamma_p^2 \| g_p \|^2}{2}$$

$$\le f(\hat{x}_{t_p}) - \frac{\gamma_p}{2} \| \nabla f(\hat{x}_{t_p}) \|^2 + \frac{\gamma_p}{2} \| \nabla f(\hat{x}_{t_p}) - g_p \|^2 + (L_0 + L_1 \| \nabla f(\hat{x}_{t_p}) \|) \exp \left\{ L_1 \gamma_p \left\| g_p \right\| \right\} \frac{\gamma_p^2 \| g_p \|^2}{2}.$$

Consider $\frac{\gamma_p}{2}\|\nabla f(\hat{x}_{t_p}) - g_p\|^2$. We have

$$
\begin{aligned}
\frac{\gamma_p}{2}\|\nabla f(\hat{x}_{t_p}) - g_p\|^2 &= \frac{\gamma_p}{2}\left\|\frac{1}{M}\sum_{m=1}^{M}\left(\nabla f_m(\hat{x}_{t_p}) - \frac{1}{v-t_p}\sum_{j=t_p}^{v}\nabla f_m\left(x_j^m\right)\right)\right\|^2 \\
&\leq \frac{\gamma_p}{2}\frac{1}{M(v-t_p)}\sum_{m=1}^{M}(L_0 + L_1\|\nabla f_m(\hat{x}_{t_p})\|)^2\sum_{j=t_p}^{v}\|x_j^m - \hat{x}_{t_p}\|^2\exp\left\{2L_1\left\|x_j^m - \hat{x}_{t_p}\right\|\right\} \\
&\leq \frac{9\gamma_p}{8(v-t_p)}(L_0 + L_1\max_m\|\nabla f_m(\hat{x}_{t_p})\|)^2\frac{1}{M}\sum_{m=1}^{M}\sum_{j=t_p}^{v}\|x_j^m - \hat{x}_{t_p}\|^2 \\
&= \frac{9\gamma_p a_p^2}{8(v-t_p)}\frac{1}{M}\sum_{m=1}^{M}\sum_{j=t_p}^{v}\|x_j^m - \hat{x}_{t_p}\|^2.
\end{aligned}
$$

Notice that

$$
\begin{aligned}
\frac{\gamma_p^2\|g_p\|^2}{2} &= \frac{\gamma_p^2}{2}\left\|\frac{1}{M(v-t_p)}\sum_{m=1}^{M}\sum_{j=t_p+1}^{v}\nabla f_m\left(x_j^m\right)\right\|^2 \\
&\leq \frac{\gamma_p^2}{(v-t_p)^2}\left\|\frac{1}{M}\sum_{m=1}^{M}\sum_{j=t_p+1}^{v}\left(\nabla f_m\left(x_j^m\right) - \nabla f_m\left(\hat{x}_{t_p}\right)\right)\right\|^2 + \frac{\gamma_p^2}{(v-t_p)^2}\left\|\frac{1}{M}\sum_{m=1}^{M}\sum_{j=t_p+1}^{v}\nabla f_m\left(\hat{x}_{t_p}\right)\right\|^2 \\
&\leq \frac{\gamma_p^2}{(v-t_p)}\left(L_0 + L_1\max_m\|\nabla f_m(\hat{x}_{t_p})\|\right)^2\frac{1}{M}\sum_{m=1}^{M}\sum_{j=t_p}^{v}\|x_j^m - \hat{x}_{t_p}\|^2\exp\left\{2L_1\left\|x_j^m - \hat{x}_{t_p}\right\|\right\} \\
&\quad + \gamma_p^2\left\|\nabla f\left(\hat{x}_{t_p}\right)\right\|^2 \\
&= \frac{9\gamma_p^2 a_p^2}{4M(v-t_p)}\sum_{m=1}^{M}\sum_{j=t_p}^{v}\left\|x_j^m - \hat{x}_{t_p}\right\|^2 + \gamma_p^2\left\|\nabla f\left(\hat{x}_{t_p}\right)\right\|^2.
\end{aligned}
$$

Further, recalling that $\gamma_{\leq\frac{1}{8\hat{a}_p}}$ and $\alpha_p \leq \frac{\hat{a}_p C}{a_p\left(L_0+L_1 A_{t_p}\right)} \leq \frac{C}{L_0+L_1 A_{t_p}}$, we have

$$
\begin{aligned}
L_1\gamma_p\|g_p\| &= \gamma_p L_1\left\|\frac{1}{M(v-t_p)}\sum_{m=1}^{M}\sum_{j=t_p+1}^{v}\nabla f_m\left(x_j^m\right)\right\| \\
&\leq \frac{\gamma_p L_1}{M(v-t_p)}\left\|\sum_{m=1}^{M}\sum_{j=t_p+1}^{v}\left(\nabla f_m\left(x_j^m\right) - \nabla f_m\left(\hat{x}_{t_p}\right)\right)\right\| + \frac{\gamma_p L_1}{M(v-t_p)}\left\|\sum_{m=1}^{M}\sum_{j=t_p+1}^{v}\nabla f_m\left(\hat{x}_{t_p}\right)\right\| \\
&\leq \frac{\gamma_p a_p L_1}{M(v-t_p)}\sum_{m=1}^{M}\sum_{j=t_p+1}^{v}\left\|x_j^m - \hat{x}_{t_p}\right\| + \gamma_p L_1\left\|\nabla f\left(\hat{x}_{t_p}\right)\right\| \\
&\leq \frac{a_p L_1}{8\hat{a}_p M(v-t_p)}\sum_{m=1}^{M}\sum_{j=t_p+1}^{v}\left\|\sum_{\ell=t_p}^{j-1}\alpha_p\nabla f_m\left(x_\ell^m\right)\right\| + \gamma_p L_1\left\|\nabla f\left(\hat{x}_{t_p}\right)\right\| \\
&\leq \frac{\ln 1.5}{8} + \frac{1}{8}.
\end{aligned}
$$

Therefore, since $\exp\{L_1\gamma_p\|g_p\|\} \leq 2$, we obtain

$$f\left(\hat{x}_{t_{p+1}}\right) \leq f(\hat{x}_{t_p}) - \frac{\gamma_p}{2}\|\nabla f(\hat{x}_{t_p})\|^2 + \frac{9\gamma_p a_p^2}{8M(v-t_p)}\sum_{m=1}^{M}\sum_{j=t_p}^{v}\|x_j^m - \hat{x}_{t_p}\|^2 + \frac{\hat{a}_p\gamma_p^2\|g_p\|^2}{2}\exp\{L_1\gamma_p\|g_p\|\}$$

$$\leq f(\hat{x}_{t_p}) - \frac{\gamma_p}{2}\|\nabla f(\hat{x}_{t_p})\|^2 + \frac{9\gamma_p a_p^2}{8M(v-t_p)}\sum_{m=1}^{M}\sum_{j=t_p}^{v}\|x_j^m - \hat{x}_{t_p}\|^2$$

$$+ \frac{9\hat{a}_p a_p^2\gamma_p^2}{2M(v-t_p)}\sum_{m=1}^{M}\sum_{j=t_p}^{v}\left\|x_j^m - \hat{x}_{t_p}\right\|^2 + 2\hat{a}_p\gamma_p^2\left\|\nabla f\left(\hat{x}_{t_p}\right)\right\|^2$$

$$= f(\hat{x}_{t_p}) + \left(2\hat{a}_p\gamma_p^2 - \frac{\gamma_p}{2}\right)\left\|\nabla f\left(\hat{x}_{t_p}\right)\right\|^2$$

$$+ \left(\frac{9\hat{a}_p a_p^2\gamma_p^2}{2} + \frac{9\gamma_p a_p^2}{8}\right)\frac{1}{M(v-t_p)}\sum_{m=1}^{M}\sum_{j=t_p}^{v}\left\|x_j^m - \hat{x}_{t_p}\right\|^2.$$

Since $\gamma_p \leq \frac{1}{8\hat{a}_p} \leq \frac{1}{2\sqrt{2}\hat{a}_p}$. Then, using Lemma 5, we have

$$f\left(\hat{x}_{t_{p+1}}\right) \leq f\left(\hat{x}_{t_p}\right) - \frac{\gamma_p}{4}\left\|\nabla f\left(\hat{x}_{t_p}\right)\right\|^2$$

$$+ \left(\frac{9\hat{a}_p a_p^2\gamma_p^2}{2} + \frac{9\gamma_p a_p^2}{8}\right)\frac{1}{M(v-t_p)}\sum_{m=1}^{M}\sum_{j=t_p}^{v}\left\|x_j^m - \hat{x}_{t_p}\right\|^2$$

$$\leq f\left(\hat{x}_{t_p}\right) - \frac{\gamma_p}{4}\left\|\nabla f\left(\hat{x}_{t_p}\right)\right\|^2$$

$$+ \left(\frac{9\hat{a}_p a_p^2\gamma_p^2}{2} + \frac{9\gamma_p a_p^2}{8}\right)32(v-t_p)^2 a_p\alpha_p^2\left(f(\hat{x}_{t_p}) - f^\star + \Delta^\star\right)$$

$$\leq f\left(\hat{x}_{t_p}\right) - \frac{\gamma_p}{4}\left\|\nabla f\left(\hat{x}_{t_p}\right)\right\|^2 + \frac{7(H-1)^2 a_p^3\alpha_p^2\left(f(\hat{x}_{t_p}) - f^\star + \Delta^\star\right)}{\hat{a}_p}.$$

Let us rewrite the inequality in the following way:

$$\frac{\gamma_p}{4}\left\|\nabla f\left(\hat{x}_{t_p}\right)\right\|^2 \leq f\left(\hat{x}_{t_p}\right) - f\left(\hat{x}_{t_{p+1}}\right) + \frac{7(H-1)^2 a_p^3\alpha_p^2\left(f(\hat{x}_{t_p}) - f^\star + \Delta^\star\right)}{\hat{a}_p}. \qquad (8)$$

Since $\gamma_p \geq \frac{\varsigma}{\hat{a}_p}$, we get that

$$\frac{\gamma_p\left\|\nabla f\left(\hat{x}_{t_p}\right)\right\|^2}{4} \geq \frac{\varsigma\left\|\nabla f\left(\hat{x}_{t_p}\right)\right\|^2}{4\hat{a}_p}.$$

Therefore,

$$\frac{\gamma_p}{4}\|\nabla f(\hat{x}_{t_p})\|^2 \geq \begin{cases} \frac{\varsigma\|\nabla f(\hat{x}_{t_p})\|^2}{8L_0}, & \|\nabla f(\hat{x}_{t_p})\| \leq \frac{L_0}{L_1}, \\ \frac{\varsigma\|\nabla f(\hat{x}_{t_p})\|}{8L_1}, & \|\nabla f(\hat{x}_{t_p})\| > \frac{L_0}{L_1}, \end{cases} = \frac{\varsigma}{8}\min\left\{\frac{\|\nabla f(\hat{x}_{t_p})\|^2}{L_0}, \frac{\|\nabla f(\hat{x}_{t_p})\|}{L_1}\right\}.$$

Denote $\delta_p \overset{\text{def}}{=} f\left(\hat{x}_{t_p}\right) - f^\star$. Then we have

$$\frac{\varsigma}{8}\min\left\{\frac{\|\nabla f(\hat{x}_{t_p})\|^2}{L_0}, \frac{\|\nabla f(\hat{x}_{t_p})\|}{L_1}\right\} \leq \delta_p - \delta_{p+1} + \frac{7(H-1)^2\alpha_p^2 a_p^3\left(\delta_p + \Delta^\star\right)}{\hat{a}_p}.$$

Let $\alpha_p \leq \frac{1}{ca_p}\sqrt{\frac{\hat{a}_p}{a_p}}$, where $c \geq \sqrt{P}$. Applying the result of Mishchenko et al. (2020, Lemma 6), we appear at

$$\min_{0 \leq p \leq P-1}\left\{\frac{\varsigma}{8}\min\left\{\frac{\|\nabla f(\hat{x}_{t_p})\|^2}{L_0}, \frac{\|\nabla f(\hat{x}_{t_p})\|}{L_1}\right\}\right\} \leq \frac{\left(1 + \frac{7(H-1)^2\alpha_p^2 a_p^3}{\hat{a}_p}\right)^P}{P}\delta_0 + \frac{7(H-1)^2\alpha_p^2 a_p^3}{\hat{a}_p}\Delta^\star.$$

$\square$

**Corollary 5.** *Fix* $\varepsilon > 0$. *Choose* $c = \sqrt{14(H-1)^2 P}$. *Let* $\alpha_p \le \sqrt{\frac{2\hat{a}_p \delta_0}{7P(H-1)^2 a_p^3 \Delta^\star}}$. *Then, if* $P \ge \frac{32\delta_0}{\zeta\varepsilon}$, *we have* $\min_{0 \le p \le P-1}\left\{\min\left\{\frac{\|\nabla f(\hat{x}_{t_p})\|^2}{L_0}, \frac{\|\nabla f(\hat{x}_{t_p})\|}{L_1}\right\}\right\} \le \varepsilon$.

*Proof of Corollary 5.* Since $c = \sqrt{14(H-1)^2 P}$ and $\alpha_p \le \frac{1}{ca_p}\sqrt{\frac{\hat{a}_p}{a_p}}$, $\alpha_p \le \sqrt{\frac{2\hat{a}_p \delta_0}{7P(H-1)^2 a_p^3 \Delta^\star}}$, due to the choice of $P \ge \frac{32\delta_0}{\zeta\varepsilon}$, we obtain that

$$\frac{\left(1 + \frac{7(H-1)^2 \alpha_p^2 a_p^3}{\hat{a}_p}\right)^P}{P}\delta_0 \le \frac{\sqrt{e}\delta_0}{P} \le \frac{2\delta_0}{P} \le \frac{\zeta\varepsilon}{16},$$

and that

$$\frac{7(H-1)^2 \alpha_p^2 a_p^3}{\hat{a}_p}\Delta^\star \le \frac{\zeta\varepsilon}{16}.$$

Therefore, $\min_{0 \le p \le P-1}\left\{\min\left\{\frac{\|\nabla f(\hat{x}_{t_p})\|^2}{L_0}, \frac{\|\nabla f(\hat{x}_{t_p})\|}{L_1}\right\}\right\} \le \varepsilon$. $\qquad\square$

### B.4 SYMMETRIC GENERALIZED-SMOOTH FUNCTIONS UNDER PŁ-CONDITION

**Theorem 6** (Symmetric generalized-smooth convergence analysis of Algorithm 1 in PŁ-case). *Let Assumptions 1 and 3 hold for functions $f$ and $\{f_m\}_{m=1}^M$. Let Assumption 4 hold. Choose $0 < \zeta \le \frac{1}{4}$. Let $\delta_0 \stackrel{def}{=} f(x_0) - f^\star$. Choose any integer $P > \frac{64\delta_0 L_1^2}{\mu\zeta}$. For all $0 \le p \le P-1$, denote*

$$\hat{a}_p = L_0 + L_1\|\nabla f(\hat{x}_{t_p})\|, \quad a_p = L_0 + L_1 \max_m \|\nabla f_m(\hat{x}_{t_p})\|, \quad 1 \le t_{p+1} - t_p \le H.$$

*Put $\Delta^\star = f^\star - \frac{1}{M}\sum_{m=1}^M f_m^\star$. Impose the following conditions on the local stepsizes $\alpha_p$ and server stepsizes $\gamma_p$ :*

$$\alpha_p \le \min\left\{\frac{1}{2Ha_p}, \frac{1}{ca_p}\sqrt{\frac{\hat{a}_p}{a_p}}, \sqrt{\frac{\mu\zeta\hat{a}_p}{224L_1^2(H-1)^2 a_p^3\left(f(\hat{x}_{t_p}) - f^\star + \Delta^\star\right)}},\right.$$
$$\left.\sqrt{\frac{\delta_0\hat{a}_p}{7P(H-1)^2 a_p^3\left(f(\hat{x}_{t_p}) - f^\star + \Delta^\star\right)}}\right\},$$

$$\frac{\zeta}{\hat{a}_p} \le \gamma_p \le \frac{1}{8\hat{a}_p}, \quad 0 \le p \le P-1,$$

*where $c \ge \sqrt{P}$. Let $\tilde{P}$ be an integer such that $0 \le \tilde{P} \le \frac{64\delta_0 L_1^2}{\mu\zeta}$, $A > 0$ be a constant, $\alpha \le \sqrt{\frac{\delta_0}{AP}}$. Then, the iterates $\left\{\hat{x}_{t_p}\right\}_{p=0}^P$ of Algorithm 1 satisfy*

$$\delta_P \le \left(1 - \frac{\mu\zeta}{4L_0}\right)^{P-\tilde{P}}\delta_0 + \frac{4L_0 A\alpha^2}{\mu\zeta},$$

*where $\delta_P \stackrel{def}{=} f(\hat{x}_{t_P}) - f^\star$.*

*Proof of Theorem 6.* Let us follow the first steps of the proof of Theorem 5. Consider (8):

$$\frac{\gamma_p}{4}\left\|\nabla f(\hat{x}_{t_p})\right\|^2 \le f(\hat{x}_{t_p}) - f(\hat{x}_{t_{p+1}}) + \frac{7(H-1)^2 a_p^3\alpha_p^2\left(f(\hat{x}_{t_p}) - f^\star + \Delta^\star\right)}{\hat{a}_p}.$$

Since $\gamma_p \ge \frac{\zeta}{\hat{a}_p}$, and $f$ satisfies Polyak–Łojasiewicz Assumption 4, we obtain that

$$\frac{\mu\zeta\left(f(\hat{x}_{t_p}) - f^\star\right)}{2\hat{a}_p} \le f(\hat{x}_{t_p}) - f(\hat{x}_{t_{p+1}}) + \frac{7(H-1)^2 a_p^3\alpha_p^2\left(f(\hat{x}_{t_p}) - f^\star + \Delta^\star\right)}{\hat{a}_p}.$$

**1.** Let $\tilde{P}$ be the number of steps $p$, so that $\left\|\nabla f\left(\hat{x}_{t_p}\right)\right\| \geq \frac{L_0}{L_1}$. For such $p$, we have $L_0 + L_1 \left\|\nabla f\left(\hat{x}_{t_p}\right)\right\| = \hat{a}_p \leq 2L_1 \left\|\nabla f\left(\hat{x}_{t_p}\right)\right\|$. Therefore, we get

$$\frac{\mu\zeta\left(f(\hat{x}_{t_p}) - f^\star\right)}{4L_1 \left\|\nabla f\left(\hat{x}_{t_p}\right)\right\|} \leq f\left(\hat{x}_{t_p}\right) - f\left(\hat{x}_{t_{p+1}}\right) + \frac{7(H-1)^2 a_p^3 \alpha_p^2 \left(f(\hat{x}_{t_p}) - f^\star + \Delta^\star\right)}{\hat{a}_p}.$$

Notice that the relation $\hat{a}_p \leq 2L_1 \left\|\nabla f\left(\hat{x}_{t_p}\right)\right\|$ and Lemma 1 together imply

$$\frac{\left\|\nabla f\left(\hat{x}_{t_p}\right)\right\|}{4L_1} \leq \frac{\left\|\nabla f\left(\hat{x}_{t_p}\right)\right\|^2}{2\hat{a}_p} \leq f\left(\hat{x}_{t_p}\right) - f^\star.$$

Hence, we have

$$\frac{\mu\zeta}{16L_1^2} \leq f\left(\hat{x}_{t_p}\right) - f\left(\hat{x}_{t_{p+1}}\right) + \frac{7(H-1)^2 a_p^3 \alpha_p^2 \left(f(\hat{x}_{t_p}) - f^\star + \Delta^\star\right)}{\hat{a}_p}.$$

Subtracting $f^\star$ on both sides and introducing $\delta_p \overset{\text{def}}{=} f\left(\hat{x}_{t_p}\right) - f^\star$, we obtain

$$\delta_{p+1} \leq \delta_p - \frac{\mu\zeta}{16L_1^2} + \frac{7(H-1)^2 a_p^3 \alpha_p^2 \left(f(\hat{x}_{t_p}) - f^\star + \Delta^\star\right)}{\hat{a}_p}.$$

As $\alpha_p \leq \sqrt{\frac{\mu\zeta\hat{a}_p}{224 L_1^2 (H-1)^2 a_p^3 \left(f(\hat{x}_{t_p}) - f^\star + \Delta^\star\right)}}$, it follows that $\frac{7(H-1)^2 a_p^3 \alpha_p^2 \left(f(\hat{x}_{t_p}) - f^\star + \Delta^\star\right)}{\hat{a}_p} \leq \frac{\mu\zeta}{32L_1^2}$.
Therefore, we get

$$\delta_{p+1} \leq \delta_p - \frac{\mu\zeta}{32L_1^2}.$$

**2.** Suppose now that $\left\|\nabla f\left(\hat{x}_{t_p}\right)\right\| \leq \frac{L_0}{L_1}$. For such $p$, we have $L_0 + L_1 \left\|\nabla f\left(\hat{x}_{t_p}\right)\right\| = \hat{a}_p \leq 2L_0$. Hence,

$$\frac{\mu\zeta\left(f(\hat{x}_{t_p}) - f^\star\right)}{4L_0} \leq f\left(\hat{x}_{t_p}\right) - f\left(\hat{x}_{t_{p+1}}\right) + \frac{7(H-1)^2 a_p^3 \alpha_p^2 \left(f(\hat{x}_{t_p}) - f^\star + \Delta^\star\right)}{\hat{a}_p}.$$

Subtracting $f^\star$ on both sides and introducing $\delta_p \overset{\text{def}}{=} f\left(\hat{x}_{t_p}\right) - f^\star$, we obtain

$$\delta_{p+1} \leq \delta_p \rho + \frac{7(H-1)^2 a_p^3 \alpha_p^2 \left(f(\hat{x}_{t_p}) - f^\star + \Delta^\star\right)}{\hat{a}_p}, \quad \text{where } \rho \overset{\text{def}}{=} 1 - \frac{\mu\zeta}{4L_0}.$$

Let $\alpha_p \overset{\text{def}}{=} \alpha\hat{\alpha}_p$ and $\hat{\alpha}_p \leq \sqrt{\frac{A\hat{a}_p}{7(H-1)^2 a_p^3 \left(f(\hat{x}_{t_p}) - f^\star + \Delta^\star\right)}}$ for some constant $A > 0$. Then,

$$\delta_{p+1} \leq \rho\delta_p + A\alpha^2.$$

Unrolling the recursion, we derive

$$\delta_P \leq \rho^{P-\tilde{P}}\delta_0 + A\alpha^2 \sum_{i=0}^{\infty} \rho^i - \frac{\mu\zeta}{32L_1^2} \sum_{i=0}^{N-1} \rho^i$$

$$\leq \rho^{P-\tilde{P}}\delta_0 + \frac{A\alpha^2}{1-\rho} - \frac{1-\rho^{\tilde{P}}}{1-\rho} \frac{\mu\zeta}{32L_1^2}.$$

Notice that $\delta_{p+1} \leq \delta_p + A\alpha^2$, which implies

$$\delta_P \leq \delta_0 + \left(P - \tilde{P}\right) A\alpha^2 - \tilde{P}\frac{\mu\zeta}{32L_1^2}.$$

Since $\alpha \leq \sqrt{\frac{\delta_0}{AP}}$, we conclude that

$$0 \leq \delta_P \leq 2\delta_0 - \tilde{P}\frac{\mu\zeta}{32L_1^2}, \quad \Rightarrow \tilde{P} \leq \frac{64\delta_0 L_1^2}{\mu\zeta}.$$

Therefore, for $P > \frac{64\delta_0 L_1^2}{\mu\zeta}$ we can guarantee that $P - \tilde{P} > 0$ and

$$\delta_P \leq \rho^{P-\tilde{P}}\delta_0 + \frac{A\alpha^2}{1-\rho} - \tilde{P}\rho^{\tilde{P}}\frac{\mu\zeta}{32L_1^2}$$
$$\leq \rho^{P-\tilde{P}}\delta_0 + \frac{A\alpha^2}{1-\rho}.$$

$\square$

**Corollary 6.** *Fix $\varepsilon > 0$. Choose $\alpha \leq \min\left\{\sqrt{\frac{\delta_0}{AP}}, L_1\sqrt{\frac{8\delta_0\varepsilon}{L_0 AP}}\right\}$. Then, if $P \geq \frac{64\delta_0 L_1^2}{\mu\zeta} + \frac{4L_0}{\mu\zeta}\ln\frac{2\delta_0}{\varepsilon}$, we have $\delta_P \leq \varepsilon$.*

*Proof of Corollary 6.* Since $0 \leq \tilde{P} \leq \frac{64\delta_0 L_1^2}{\mu\zeta}$, $A > 0$, $\alpha \leq \sqrt{\frac{\delta_0}{AP}}$, $\alpha \leq L_1\sqrt{\frac{8\delta_0\varepsilon}{L_0 AP}}$, due to the choice of $P \geq \frac{64\delta_0 L_1^2}{\mu\zeta} + \frac{4L_0}{\mu\zeta}\ln\frac{2\delta_0}{\varepsilon}$, we obtain that

$$\left(1 - \frac{\mu\zeta}{4L_0}\right)^{P-\tilde{P}}\delta_0 \leq e^{-\frac{\mu\zeta}{4L_0}(P-\tilde{P})}\delta_0 \leq \frac{\varepsilon}{2},$$

and that

$$\frac{4L_0 A}{\mu\zeta} \cdot \frac{\delta_0}{AP} \leq \frac{\varepsilon}{2}.$$

Therefore, $\delta_P \leq \varepsilon$. $\square$

## C  RANDOM RESHUFFLING

There are several approaches, that fall under the category of permutation methods, and one of the most popular is **Random Reshuffling (RR)**. In each epoch $t$ of the RR algorithm, we sample indices $\pi_t(1), \ldots, \pi_t(N)$ without replacement from the set $\{1, 2, \ldots, N\}$. In other words, $\pi_t(1), \ldots, \pi_t(N)$ forms a random permutation of $\{1, 2, \ldots, N\}$. We then perform $N$ steps in the following manner:

$$x_{t,j}^m = x_{t,j-1}^m - \alpha_t \nabla f_{m,\pi_t(j)}(x_{t,j-1}^m), \tag{9}$$

where $f_{m,\pi_t(j)}$ is the $m$-th function after permutation $\pi_t$ on epoch $t$, and $\alpha_t$ is a stepsize at $t$-th epoch. We can rewrite this step as

$$x_{t,j}^m = x_{t,0}^m - \alpha_t \sum_{k=0}^{j-1} \nabla f_{m,\pi_t(j)}(x_{t,k}^m).$$

After each epoch we perform additional outer step with stepsize $\gamma_t$:

$$x_{t+1} = x_t - \gamma_t g_t, \quad g_t = \frac{1}{MN}\sum_{j=1}^{N}\sum_{m=1}^{M}\nabla f_{m,\pi_t(j)}(x_{t,j-1}^m). \tag{10}$$

### C.1  ASYMMETRIC GENERALIZED-SMOOTH NON-CONVEX FUNCTIONS

**Theorem 3** (non-convex asymmetric generalized-smooth convergence analysis of Algorithm 2). *Let Assumptions 1 and 2 hold for functions $f$ and $\{f_m\}_{m=1}^{M}$. Choose any $T \geq 1$. For all $0 \leq t \leq T-1$, denote*

$$\hat{a}_t = L_0 + L_1\|\nabla f(x_t)\|, \quad \tilde{a}_t = L_0 + L_1 \max_{mj}\|\nabla f_{mj}(x_t)\|.$$

*Put $\overline{\Delta}^\star = f^\star - \frac{1}{MN}\sum_{j=0}^{N-1}\sum_{m=1}^{M}f_m^\star$. Impose the following conditions on the client stepsizes $\alpha_t$ and global stepsizes $\gamma_t$:*

$$\alpha_t \leq \min\left\{\frac{\sqrt{2}}{\sqrt{3N(N-1)}\tilde{a}_t}, \frac{\sqrt{\hat{a}_t}}{c\tilde{a}_t^{3/2}}\right\}, \quad \frac{\zeta}{\hat{a}_t} \leq \gamma_t \leq \frac{1}{4\hat{a}_t}, \quad 0 \leq t \leq T-1,$$

where $0 < \zeta \leq \frac{1}{4}$, $c \geq \sqrt{((N-1)(2N-1) + 2(N+1))T}$. Let $\delta_0 \stackrel{def}{=} f(x_0) - f^\star$. Then, the iterates $\{x_t\}_{t=0}^{T-1}$ of Algorithm 2 satisfy

$$\mathbb{E}\left[\min_{t=0,\ldots,T-1}\left\{\frac{\zeta}{8}\min\left\{\frac{\|\nabla f(x_t)\|^2}{L_0}, \frac{\|\nabla f(x_t)\|}{L_1}\right\}\right\}\right]$$
$$\leq \frac{8\left(1 + \frac{3\alpha_t^2\tilde{a}_t^3}{8\hat{a}_t}((N-1)(2N-1) + 2(N+1))\right)^T}{T}\delta_0 + \frac{6\alpha_t^2\tilde{a}_t^3}{\hat{a}_t}(N+1)\Delta^\star.$$

**Lemma 6.** *Recall that $\tilde{a}_t = L_0 + L_1 \max_{m,j} \|\nabla f_{mj}(x_t)\|$. Then*

$$\frac{\gamma_t^2\|g_t\|^2}{2} \leq \tilde{a}_t\gamma_t^2 \frac{1}{MN}\sum_{j=1}^N\sum_{m=1}^M \|x_{t,j-1}^m - x_t\|^2 + \gamma_t^2\|\nabla f(x_t)\|^2. \tag{11}$$

*Proof.*

$$\frac{\|g_t\|^2}{2} = \frac{1}{2}\left\|\frac{1}{MN}\sum_{j=1}^N\sum_{m=1}^M \nabla f_{m,\pi_t(j)}(x_{t,j-1}^m)\right\|^2$$
$$= \left\|\frac{1}{MN}\sum_{j=1}^N\sum_{m=1}^M \left(\nabla f_{m,\pi_t(j)}(x_{t,j-1}^m) - \nabla f_{m,\pi_t(j)}(x_t)\right)\right\|^2 + \|\nabla f(x_t)\|^2$$
$$\leq \frac{1}{MN}\sum_{j=1}^N\sum_{m=1}^M (L_0 + L_1\|\nabla f_{m,\pi_t(j)}(x_t)\|)^2 \|x_{t,j-1}^m - x_t\|^2 + \|\nabla f(x_t)\|^2$$
$$\leq (\tilde{a}_t)^2\frac{1}{MN}\sum_{j=1}^N\sum_{m=1}^M \|x_{t,j-1}^m - x_t\|^2 + \|\nabla f(x_t)\|^2$$
$$= (\tilde{a}_t)^2\frac{1}{MN}\sum_{j=1}^N\sum_{m=1}^M \|x_{t,j-1}^m - x_t\|^2 + \|\nabla f(x_t)\|^2.$$

$\square$

**Lemma 7.** *Let Assumptions 1 and 2 hold for functions $f$ and $\{f_m\}_{m=1}^M$. Then, if we choose $\alpha_t \leq \frac{\sqrt{2}}{\sqrt{3n(n-1)(\tilde{a}_t)}}$, we get*

$$\mathbb{E}\left[\frac{1}{N}\sum_{j=1}^N \|x_{t,j}^m - x_t\|^2 \Big| x_t\right] \leq 2\alpha_t^2\tilde{a}_t((N-1)(2N-1) + 2(N+1)(f(x_t) - f^\star))$$
$$+ 4\alpha_t^2\tilde{a}_t(N+1)\overline{\Delta}^\star. \tag{12}$$

*Proof.* From (9) we have

$$x_{t,j}^m = x_{t,j-1}^m - \alpha_t\nabla f_{m,\pi_t(j)}(x_{t,j-1}^m) = x_t - \sum_{k=1}^j \alpha_t\nabla f_{m,\pi_t(k)}(x_{t,k-1}^m).$$

Thus,

$$
\begin{aligned}
\left\| x_{t,j}^m - x_t \right\|^2 &= \left\| \sum_{k=1}^j \alpha_t \nabla f_{m,\pi_t(k)}(x_{t,k-1}^m) \right\|^2 \\
&\leq 2 \left\| \sum_{k=1}^j \alpha_t \left( \nabla f_{m,\pi_t(k)}(x_{t,k-1}^m) - \nabla f_{m,\pi_t(k)}(x_t) \right) \right\|^2 \\
&\quad + 2 \left\| \sum_{k=1}^j \alpha_t \nabla f_{m,\pi_t(k)}(x_t) \right\|^2 \\
&\leq 2j \sum_{k=1}^j (\alpha_t)^2 \left( L_0 + L_1 \left\| \nabla f_{m,\pi_t(k)}(x_t) \right\| \right)^2 \left\| x_{t,k-1}^m - x_t \right\|^2 \\
&\quad + 2 \left\| \sum_{k=1}^j \alpha_t \nabla f_{m,\pi_t(k)}(x_t) \right\|^2.
\end{aligned}
$$

Using last inequality, we get

$$
\begin{aligned}
\frac{1}{N} \sum_{j=1}^N \left\| x_{t,j}^m - x_t \right\|^2 &\leq \sum_{j=1}^N \frac{2j}{N} \sum_{k=1}^j (\alpha_t)^2 \left( L_0 + L_1 \left\| \nabla f_{m,\pi_t(k)}(x_t) \right\| \right)^2 \left\| x_{t,k-1}^m - x_t \right\|^2 \\
&\quad + \frac{2}{N} \sum_{j=1}^N \left\| \sum_{k=1}^j \alpha_t \nabla f_{m,\pi_t(k)}(x_t) \right\|^2 \\
&\leq (\alpha_t)^2 (\tilde{a}_t)^2 \sum_{j=1}^N \frac{2j}{N} \sum_{k=1}^j \left\| x_{t,k-1}^m - x_t \right\|^2 + \frac{2\alpha_t^2}{N} \sum_{j=1}^N \left\| \sum_{k=1}^j \nabla f_{m,\pi_t(k)}(x_t) \right\|^2.
\end{aligned}
$$

Let $\alpha_t \leq \frac{\beta}{\tilde{a}_t}$, where $\beta$ is constant. Then, we take a conditional expectation of the last inequality and get the following

$$
\begin{aligned}
\mathbb{E}\left[ \frac{1}{N} \sum_{j=1}^N \left\| x_{t,j}^m - x_t \right\|^2 \Big| x_t \right] &\leq \mathbb{E}\left[ \frac{2\beta^2}{N} \sum_{j=1}^N j \sum_{k=1}^j \left\| x_{t,k-1}^m - x_t \right\|^2 \Big| x_t \right] \\
&\quad + \frac{2\alpha_t^2}{N} \sum_{j=1}^N \mathbb{E}\left[ \left\| \sum_{k=1}^j \nabla f_{m,\pi_t(k)}(x_t) \right\|^2 \Big| x_t \right].
\end{aligned}
$$

Denote $\sigma_t^2 = \frac{1}{N} \sum_{j=0}^{N-1} \left\| \nabla f_{m,\pi_t(j)}(x_t) - f(x_t) \right\|^2$, and consider

$$
\mathbb{E}\left[ \left\| \sum_{k=1}^j \nabla f_{m,\pi_t(k)}(x_t) \right\|^2 \Big| x_t \right].
$$

From Malinovsky et al. (2022, Lemma 1) we get

$$
\begin{aligned}
\mathbb{E}\left[ \left\| \sum_{k=1}^j \nabla f_{m,\pi_t(k)}(x_t) \right\|^2 \Big| x_t \right] &\leq j^2 \left\| \nabla f(x_t) \right\| + j^2 \mathbb{E}\left[ \left\| \frac{1}{j} \sum_{k=1}^j \left( \nabla f_{m,\pi_t(k)}(x_t) - f(x_t) \right) \right\|^2 \Big| x_t \right] \\
&\leq j^2 \left\| \nabla f(x_t) \right\| + \frac{j(N-j)}{N-1} \sigma_t^2.
\end{aligned}
$$

Thus,

$$
\begin{aligned}
\mathbb{E}\left[\frac{1}{N}\sum_{j=1}^{N}\left\|x_{t,j}^{m}-x_t\right\|^2 \Big| x_t\right] &\leq \mathbb{E}\left[\frac{2\beta^2}{N}\sum_{j=1}^{N}j\sum_{k=1}^{j}\left\|x_{t,k}^{m}-x_t\right\|^2 \Big| x_t\right] \\
&\quad + \frac{2\alpha_t^2}{N}\sum_{j=1}^{N}\left(j^2\left\|\nabla f(x_t)\right\| + \frac{j(N-j)}{N-1}\sigma_t^2\right) \\
&\leq \mathbb{E}\left[\frac{2\beta^2}{N}\cdot\frac{N(N-1)}{2}\sum_{j=1}^{N}\left\|x_{t,j}^{m}-x_t\right\|^2 \Big| x_t\right] \\
&\quad + \frac{2\alpha_t^2}{N}\left(\frac{(N(N-1)(2N-1))}{6}\left\|\nabla f(x_t)\right\|^2\right) \\
&\quad + \frac{2\alpha_t^2}{N}\frac{N(N+1)}{3}\sigma_t^2.
\end{aligned}
$$

Further,

$$
\begin{aligned}
3\cdot\mathbb{E}\left[\frac{1}{N}\sum_{j=1}^{N}\left\|x_{t,j}^{m}-x_t\right\|^2 \Big| x_t\right] &\leq 3\beta^2 N(N-1)\mathbb{E}\left[\frac{1}{N}\sum_{j=1}^{N}\left\|x_{t,j}^{m}-x_t\right\|^2 \Big| x_t\right] \\
&\quad + 2\alpha_t^2\left(\frac{(N-1)(2N-1)}{2}\left\|\nabla f(x_t)\right\|^2 + (N+1)\sigma_t^2\right).
\end{aligned}
$$

Thus, if we choose $\beta \leq \sqrt{\frac{2}{3N(N-1)}}$, we get

$$
\begin{aligned}
\mathbb{E}\left[\frac{1}{N}\sum_{j=1}^{N}\left\|x_{t,j}^{m}-x_t\right\|^2 \Big| x_t\right] &\leq (3-3\beta^2 N(N-1))\mathbb{E}\left[\frac{1}{N}\sum_{j=1}^{N}\left\|x_{t,j}^{m}-x_t\right\|^2 \Big| x_t\right] \\
&\leq 2\alpha_t^2\left(\frac{(N-1)(2N-1)}{2}\left\|\nabla f(x_t)\right\|^2 + (N+1)\sigma_t^2\right) \\
&\leq 2\alpha_t^2\Big((N-1)(2N-1)(f(x_t)-f^\star)(L_0+L_1\left\|\nabla f(x_t)\right\|) \\
&\quad + (N+1)\frac{1}{N}\sum_{j=1}^{N}\left\|\nabla f_{m,\pi_t(j)}(x_t)\right\|^2\Big) \\
&\stackrel{\text{Lemma 1}}{\leq} 2\alpha_t^2\Big((N-1)(2N-1)(f(x_t)-f^\star)\tilde{a}_t \\
&\quad + 2(N+1)\left(\tilde{a}_t\right)\frac{1}{N}\sum_{j=1}^{N}(f_{mj}(x_t)-f^\star_{mj})\Big) \\
&\leq 2\alpha_t^2\Big((N-1)(2N-1)(f(x_t)-f^\star)\tilde{a}_t \\
&\quad + 2(N+1)\left(\tilde{a}_t\right)\frac{1}{N}\sum_{j=1}^{N}(f_{mj}(x_t)-f^\star_{mj})\Big)
\end{aligned}
$$

Now, adding and removing $f^\star$ to the sum factor on the right-hand side, we get

$$\mathbb{E}\left[\frac{1}{MN}\sum_{j=1}^{N}\sum_{m=1}^{M}\left\|x_{t,j}^m - x_t\right\|^2 \Big| x_t\right] \leq 2\alpha_t^2\tilde{a}_t(N-1)(2N-1)(f(x_t) - f^\star)$$

$$+ 4\alpha_t^2\tilde{a}_t(N+1)\frac{1}{NM}\sum_{j=1}^{N}\sum_{m=1}^{M}(f_{mj}(x_t) - f_{mj}^\star)$$

$$= 2\alpha_t^2\tilde{a}_t\left((N-1)(2N-1) + 2(N+1)\right)(f(x_t) - f^\star)$$

$$+ 4\alpha_t^2\tilde{a}_t(N+1)\overline{\Delta}^\star.$$

$\square$

*Proof of Theorem 3.* From Lemma 1 and (10) we get

$$f(x_{t+1}) \leq f(x_t) - \gamma_t\left\langle\nabla f(x_t), g_t\right\rangle + (L_0 + L_1\|\nabla f(x_t)\|)\frac{\gamma_t^2\|g_t\|^2}{2}.$$

Additionally, from the fact that $2\left\langle a, b\right\rangle = -\|a - b\|^2 + \|a\|^2 + \|b\|^2$ we can get

$$f(x_{t+1}) \leq f(x_t) - \gamma_t\left\langle\nabla f(x_t), g_t\right\rangle + (L_0 + L_1\|\nabla f(x_t)\|)\frac{\gamma_t^2\|g_t\|^2}{2}$$

$$\leq f(x_t) - \frac{\gamma_t}{2}\left(-\|\nabla f(x_t) - g_t\|^2 + \|\nabla f(x_t)\|^2 + \|g_t\|^2\right)$$

$$+ (L_0 + L_1\|\nabla f(x_t)\|)\frac{\gamma_t^2\|g_t\|^2}{2}$$

$$\leq f(x_t) - \frac{\gamma_t}{2}\|\nabla f(x_t)\|^2 + \frac{\gamma_t}{2}\|\nabla f(x_t) - g_t\|^2 + (L_0 + L_1\|\nabla f(x_t)\|)\frac{\gamma_t^2\|g_t\|^2}{2}.$$

Consider $\frac{\gamma_t}{2}\|\nabla f(x_t) - g_t\|^2$ and denote $\hat{a}_t = (L_0 + L_1\|\nabla f(x_t)\|)$ and $a_t = (L_0 + L_1\max_m\|\nabla f_m(x_t)\|)$, then:

$$\frac{\gamma_t}{2}\|\nabla f(x_t) - g_t\|^2 = \frac{\gamma_t}{2}\left\|\frac{1}{MN}\sum_{j=1}^{N}\sum_{m=1}^{M}\nabla f_{m,\pi_t(j)}(x_t) - \nabla f_{m,\pi_t(j)}(x_{t,j}^m)\right\|$$

$$\leq \frac{\gamma_t}{2}\frac{1}{MN}\sum_{j=1}^{N}\sum_{m=1}^{M}(L_0 + L_1\|\nabla f_{m,\pi_t(j)}(x_t)\|)^2\|x_t - x_{t,j}^m\|^2$$

$$= \frac{\gamma_t}{2}\tilde{a}_t^2\frac{1}{MN}\sum_{j=1}^{N}\sum_{m=1}^{M}\|x_t - x_{t,j}^m\|^2.$$

From the above inequality and Lemma 6 we get

$$f(x_{t+1}) \leq f(x_t) - \frac{\gamma_t}{2}\|\nabla f(x_t)\|^2 + \frac{\gamma_t}{2}\tilde{a}_t^2\frac{1}{MN}\sum_{j=1}^{N}\sum_{m=1}^{M}\|x_t - x_{t,j}^m\|^2 + \hat{a}_t\frac{\gamma_t^2\|g_t\|^2}{2}$$

$$\overset{(11)}{\leq} f(x_t) - \frac{\gamma_t}{2}\|\nabla f(x_t)\|^2 + \frac{\gamma_t}{2}\tilde{a}_t^2\frac{1}{MN}\sum_{j=1}^{N}\sum_{m=1}^{M}\|x_t - x_{t,j}^m\|^2$$

$$+ \hat{a}_t\tilde{a}_t^2\gamma_t^2\frac{1}{MN}\sum_{j=1}^{N}\sum_{m=1}^{M}\left\|x_{t,j}^m - x_t\right\|^2 + \hat{a}_t\gamma_t^2\|\nabla f(x_t)\|^2$$

$$\leq f(x_t) + \left(\hat{a}_t\gamma_t^2 - \frac{\gamma_t}{2}\right)\|\nabla f(x_t)\|^2 + \left(\hat{a}_t\tilde{a}_t^2\gamma_t^2 + \frac{\gamma_t}{2}\tilde{a}_t^2\right)\frac{1}{MN}\sum_{j=1}^{N}\sum_{m=1}^{M}\left\|x_{t,j}^m - x_t\right\|^2.$$

Let $\gamma_t \leq \frac{1}{4\hat{a}_t}$, then

$$f(x_{t+1}) \leq f(x_t) - \frac{\gamma_t}{4}\|\nabla f(x_t)\|^2 + \left(\hat{a}_t \tilde{a}_t^2 \gamma_t^2 + \frac{\gamma_t}{2}\tilde{a}_t^2\right)\frac{1}{MN}\sum_{j=1}^{N}\sum_{m=1}^{M}\left\|x_{t,j}^m - x_t\right\|^2.$$

Now, if we take conditional expectation of this and use Lemma 9, we get

$$\mathbb{E}\left[f(x_{t+1})|x_t\right] \leq f(x_t) - \frac{\gamma_t}{4}\|\nabla f(x_t)\|^2$$
$$+ \left(\hat{a}_t \tilde{a}_t^2 \gamma_t^2 + \frac{\gamma_t}{2}\tilde{a}_t^2\right)\mathbb{E}\left[\frac{1}{MN}\sum_{j=1}^{N}\sum_{m=1}^{M}\left\|x_{t,j}^m - x_t\right\|^2 \Bigg| x_t\right]$$
$$\leq f(x_t) - \frac{\gamma_t}{4}\|\nabla f(x_t)\|^2$$
$$+ 2\alpha_t^2 \tilde{a}_t \left(\hat{a}_t \tilde{a}_t^2 \gamma_t^2 + \frac{\gamma_t}{2}\tilde{a}_t^2\right)$$
$$\times \left(((N-1)(2N-1) + 2(N+1))\left(f(x_t) - f^\star\right) + 2(N+1)\overline{\Delta}^\star\right).$$

Since $\gamma_t \leq \frac{1}{4\hat{a}_t}$, then

$$\frac{\gamma_t}{4}\|\nabla f(x_t)\|^2 \leq f(x_t) - \mathbb{E}\left[f(x_{t+1})|x_t\right] + \frac{3\alpha_t^2 \tilde{a}_t^3}{8\hat{a}_t}\left(((N-1)(2N-1) + 2(N+1))\,\delta_t + 2(N+1)\overline{\Delta}^\star\right). \tag{13}$$

Consider the left-hand side of (16). Due to the bounds $\frac{1}{4\hat{a}_t} \geq \gamma_t \geq \frac{\zeta}{\hat{a}_t}$ on $\gamma_t$, we have

$$\frac{\gamma_t}{4}\|\nabla f(x_t)\|^2 \geq \frac{\zeta\|\nabla f(x_t)\|^2}{4\hat{a}_t}.$$

Then, we get

$$\frac{\gamma_t}{4}\|\nabla f(x_t)\|^2 \geq \begin{cases} \frac{\zeta\|\nabla f(x_t)\|^2}{8L_0}, & \|\nabla f(x_t)\| \leq \frac{L_0}{L_1}, \\ \frac{\zeta\|\nabla f(x_t)\|}{8L_1}, & \|\nabla f(x_t)\| > \frac{L_0}{L_1} \end{cases}$$
$$= \frac{\zeta}{8}\min\left\{\frac{\|\nabla f(x_t)\|^2}{L_0}, \frac{\|\nabla f(x_t)\|}{L_1}\right\} \tag{14}$$

Denote $\delta_t \equiv f(x_t) - f^\star$, then from (16) and (17) we get

$$f(x_t) - f(x_{t+1}) + \frac{3\alpha_t^2 \tilde{a}_t^3}{8\hat{a}_t}\left(((N-1)(2N-1) + 2(N+1))\,\delta_t + 2(N+1)\overline{\Delta}^\star\right)$$
$$= \delta_t - \delta_{t+1} + \frac{3\alpha_t^2 \tilde{a}_t^3}{8\hat{a}_t}\left(((N-1)(2N-1) + 2(N+1))\,\delta_t + 2(N+1)\overline{\Delta}^\star\right)$$
$$\geq \frac{\zeta}{8}\min\left\{\frac{\|\nabla f(x_t)\|^2}{L_0}, \frac{\|\nabla f(x_t)\|}{L_1}\right\}$$

Let $\alpha_t \leq \frac{1}{c\tilde{a}_t}\cdot\sqrt{\frac{\hat{a}_t}{\tilde{a}_t}}$, where $c$ is a constant such that $\sqrt{((N-1)(2N-1) + 2(N+1))T} \leq c$. Now we take full expectation and use from Mishchenko et al. (2020, Lemma 6):

$$\mathbb{E}\left[\min_{t=0,\dots,T-1}\left\{\frac{\zeta}{8}\min\left\{\frac{\|\nabla f(x_t)\|^2}{L_0}, \frac{\|\nabla f(x_t)\|}{L_1}\right\}\right\}\right]$$
$$\leq \frac{\left(1 + \frac{3\alpha_t^2 \tilde{a}_t^3}{8\hat{a}_t}((N-1)(2N-1) + 2(N+1))\right)^T}{T}\delta_0 + \frac{3\alpha_t^2 \tilde{a}_t^3}{4\hat{a}_t}(N+1)\overline{\Delta}^\star.$$

$\square$

**Corollary 3.** *Fix $\varepsilon > 0$. Choose $c = \sqrt{((N-1)(2N-1) + 2(N+1))T}$. Let $\alpha_t \leq 8\sqrt{\frac{\hat{a}_t}{3\tilde{a}_t^3 T(N+1)\overline{\Delta}^\star}}$. Then, if $T \geq \frac{256\delta_0}{\zeta\varepsilon}$, we have*

$$\mathbb{E}\left[\min_{t=0,\dots,T-1}\left\{\min\left\{\frac{\|\nabla f(x_t)\|^2}{L_0}, \frac{\|\nabla f(x_t)\|}{L_1}\right\}\right\}\right] \leq \varepsilon.$$

*Proof of Corollary 3.* Since $c = \sqrt{((N-1)(2N-1) + 2(N+1))T}$ and $\alpha_t \leq \frac{1}{c a_t}\sqrt{\frac{\hat{a}_t}{\tilde{a}_t}}$, $\alpha_t \leq 8\sqrt{\frac{\hat{a}_t}{3\tilde{a}_t^3 T(N+1)\overline{\Delta}^\star}}$, due to the choice of $T \geq \frac{256\delta_0}{\zeta\varepsilon}$, we obtain that

$$\frac{\left(1 + \frac{3\alpha_t^2\tilde{a}_t^3}{8\hat{a}_t}((N-1)(2N-1) + 2(N+1))\right)^T}{T}\delta_0 \leq \frac{e^{\frac{3}{8}}\delta_0}{T} \leq \frac{2\delta_0}{T} \leq \frac{\zeta\varepsilon}{16},$$

and that

$$\frac{3\alpha_t^2\tilde{a}_t^3}{4\hat{a}_t}(N+1)\overline{\Delta}^\star \leq \frac{\zeta\varepsilon}{16}.$$

Therefore, $\mathbb{E}\left[\min_{t=0,\dots,T-1}\left\{\min\left\{\frac{\|\nabla f(x_t)\|^2}{L_0}, \frac{\|\nabla f(x_t)\|}{L_1}\right\}\right\}\right] \leq \varepsilon.$ $\qquad\square$

## C.2 Asymmetric generalized-smooth functions under PŁ-condition

**Theorem 7.** *Let Assumptions 1 and 2 hold for functions $f$, $\{f_m\}_{m=1}^M$ and $\{f_{mj}\}_{m=1,j=0}^{M,N-1}$. Let Assumption 4 hold. Choose $0 < \zeta \leq \frac{1}{4}$. Let $\delta_0 \overset{def}{=} f(x_0) - f^\star$. Choose any integer $T > \frac{64\delta_0 L_1^2}{\mu\zeta}$. For all $0 \leq t \leq T-1$, denote*

$$\hat{a}_t = L_0 + L_1\|\nabla f(x_t)\|, \quad a_t = L_0 + L_1\max_m\|\nabla f_m(x_t)\|.$$

*Put $\overline{\Delta}^\star = f^\star - \frac{1}{MN}\sum_{m=1}^M\sum_{j=1}^N f_{mj}^\star$. Impose the following conditions on the client stepsizes $\alpha_t$ and global stepsizes $\gamma_t$:*

$$\alpha_t \leq \min\Bigg\{\frac{\sqrt{2}}{\sqrt{3M(M-1)}\tilde{a}_t}, \frac{\sqrt{\hat{a}_t}}{c\tilde{a}_t^{3/2}},$$

$$\sqrt{\frac{\hat{a}_t\mu\zeta}{12L_1^2\tilde{a}_t^3\left(\delta_t((N-1)(2N-1) + 2(N+1)) + 2(N+1)\overline{\Delta}^\star\right)}}$$

$$\sqrt{\frac{8\hat{a}_t\delta_0}{3T\tilde{a}_t^3\left(\delta_t((N-1)(2N-1) + 2(N+1)) + 2(N+1)\overline{\Delta}^\star\right)}}\Bigg\},$$

$$\frac{\zeta}{\hat{a}_t} \leq \gamma_t \leq \frac{1}{4\hat{a}_t}, \quad 0 \leq t \leq T-1,$$

*where $c \geq \sqrt{((N-1)(2N-1) + 2(N+1))T}$. Let $\delta_0 \overset{def}{=} f(x_0) - f^\star$. Let $\tilde{T}$ be an integer such that $0 \leq \tilde{T} \leq \frac{64\delta_0 L_1^2}{\mu\zeta}$, $A > 0$ be a constant, $\alpha \leq \sqrt{\frac{\delta_0}{AT}}$. Then, the iterates $\{x_t\}_{t=0}^{T-1}$ of Algorithm 2 satisfy*

$$\delta_T \leq \left(1 - \frac{\mu\zeta}{4L_0}\right)^{T-\tilde{T}}\delta_0 + \frac{4L_0 A\alpha^2}{\mu\zeta},$$

*where $\delta_T \overset{def}{=} f(x_T) - f^\star$.*

*Proof of Theorem 7.* Let us follow the first steps of the proof of Theorem 3. Consider (16):

$$\frac{\gamma_t}{4}\|\nabla f(x_t)\|^2 \leq f(x_t) - f(x_{t+1})$$

$$+ \frac{3\alpha_t^2\tilde{a}_t^3}{8\hat{a}_t}\left(((N-1)(2N-1) + 2(N+1))\delta_t + 2(N+1)\overline{\Delta}^\star\right).$$

Since $\gamma_p \geq \frac{\zeta}{\hat{a}_p}$, and $f$ satisfies Polyak–Łojasiewicz Assumption 4, we obtain that

$$\frac{\mu\zeta\left(f(\hat{x}_{t_p}) - f^\star\right)}{2\hat{a}_p} \leq f(x_t) - f(x_{t+1})$$
$$+ \frac{3\alpha_t^2 \tilde{a}_t^3}{8\hat{a}_t}\left(\delta_t((N-1)(2N-1) + 2(N+1)) + 2(N+1)\overline{\Delta}^\star\right).$$

**1.** Let $\tilde{T}$ be the number of steps $t$, so that $\|\nabla f(\hat{x}_t)\| \geq \frac{L_0}{L_1}$. For such $t$, we have $L_0 + L_1\|\nabla f(x_t)\| = \hat{a}_t \leq 2L_1\|\nabla f(x_t)\|$. Therefore, we get

$$\frac{\mu\zeta\left(f(x_t) - f^\star\right)}{4L_1\|\nabla f(x_t)\|} \leq f(x_t) - f(x_{t+1})$$
$$+ \frac{3\alpha_t^2 \tilde{a}_t^3}{8\hat{a}_t}\left(\delta_t((N-1)(2N-1) + 2(N+1)) + 2(N+1)\overline{\Delta}^\star\right).$$

Notice that the relation $\hat{a}_t \leq 2L_1\|\nabla f(x_t)\|$ and Lemma 1 together imply

$$\frac{\|\nabla f(x_t)\|}{4L_1} \leq \frac{\|\nabla f(x_t)\|^2}{2\hat{a}_t} \leq f(x_t) - f^\star.$$

Hence, we have

$$\frac{\mu\zeta}{16L_1^2} \leq f(x_t) - f(x_{t+1})$$
$$+ \frac{3\alpha_t^2 \tilde{a}_t^3}{8\hat{a}_t}\left(\delta_t((N-1)(2N-1) + 2(N+1)) + 2(N+1)\overline{\Delta}^\star\right).$$

Subtracting $f^\star$ on both sides and introducing $\delta_t \stackrel{\text{def}}{=} f(x_t) - f^\star$, we obtain

$$\delta_{t+1} \leq \delta_t - \frac{\mu\zeta}{16L_1^2}$$
$$+ \frac{3\alpha_t^2 \tilde{a}_t^3}{8\hat{a}_t}\left(\delta_t((N-1)(2N-1) + 2(N+1)) + 2(N+1)\overline{\Delta}^\star\right).$$

As $\alpha_t \leq \sqrt{\frac{\hat{a}_t \mu\zeta}{12L_1^2 \tilde{a}_t^3\left(\delta_t((N-1)(2N-1)+2(N+1))+2(N+1)\overline{\Delta}^\star\right)}}$, it follows that

$$\frac{3\alpha_t^2 \tilde{a}_t^3}{8\hat{a}_t}\left(\delta_t((N-1)(2N-1) + 2(N+1)) + 2(N+1)\overline{\Delta}^\star\right) \leq \frac{\mu\zeta}{32L_1^2}.$$

Therefore, we get

$$\delta_{t+1} \leq \delta_t - \frac{\mu\zeta}{32L_1^2}.$$

**2.** Suppose now that $\|\nabla f(x_t)\| \leq \frac{L_0}{L_1}$. For such $t$, we have $L_0 + L_1\|\nabla f(x_t)\| = \hat{a}_p \leq 2L_0$. Hence,

$$\frac{\mu\zeta\left(f(x_t) - f^\star\right)}{4L_0} \leq f(x_t) - f(x_{t+1})$$
$$+ \frac{3\alpha_t^2 \tilde{a}_t^3}{8\hat{a}_t}\left(\delta_t((N-1)(2N-1) + 2(N+1)) + 2(N+1)\overline{\Delta}^\star\right).$$

Subtracting $f^\star$ on both sides and introducing $\delta_t \stackrel{\text{def}}{=} f(x_t) - f^\star$, we obtain

$$\delta_{t+1} \leq \delta_t \rho + \frac{3\alpha_t^2 \tilde{a}_t^3}{8\hat{a}_t}(\delta_t((N-1)(2N-1) + 2(N+1)) + 2(N+1)\overline{\Delta}^\star).$$

where $\rho \stackrel{\text{def}}{=} 1 - \frac{\mu\zeta}{4L_0}$. Let $\alpha_t \stackrel{\text{def}}{=} \alpha\hat{\alpha}_t$ with $\hat{\alpha}_t \leq \sqrt{\frac{8\hat{a}_t A}{3\tilde{a}_t^3\left(\delta_t((N-1)(2N-1)+2(N+1))+2(N+1)\overline{\Delta}^\star\right)}}$ for some constant $A > 0$. Then,

$$\delta_{t+1} \leq \rho\delta_t + A\alpha^2.$$

Unrolling the recursion, we derive

$$\delta_T \leq \rho^{T-\tilde{T}}\delta_0 + A\alpha^2 \sum_{i=0}^{\infty} \rho^i - \frac{\mu\zeta}{32L_1^2} \sum_{i=0}^{N-1} \rho^i$$

$$\leq \rho^{T-\tilde{T}}\delta_0 + \frac{A\alpha^2}{1-\rho} - \frac{1-\rho^{\tilde{T}}}{1-\rho} \frac{\mu\zeta}{32L_1^2}.$$

Notice that $\delta_{t+1} \leq \delta_t + A\alpha^2$, which implies

$$\delta_T \leq \delta_0 + \left(T - \tilde{T}\right) A\alpha^2 - \tilde{T} \frac{\mu\zeta}{32L_1^2}.$$

Since $\alpha \leq \sqrt{\frac{\delta_0}{AT}}$, we conclude that

$$0 \leq \delta_T \leq 2\delta_0 - \tilde{T}\frac{\mu\zeta}{32L_1^2}, \quad \Rightarrow \tilde{T} \leq \frac{64\delta_0 L_1^2}{\mu\zeta}.$$

Therefore, for $T > \frac{64\delta_0 L_1^2}{\mu\zeta}$ we can guarantee that $T - \tilde{T} > 0$ and

$$\delta_T \leq \rho^{T-\tilde{T}}\delta_0 + \frac{A\alpha^2}{1-\rho} - \tilde{T}\rho^{\tilde{T}}\frac{\mu\zeta}{32L_1^2}$$

$$\leq \rho^{T-\tilde{T}}\delta_0 + \frac{A\alpha^2}{1-\rho}.$$

$$\square$$

**Corollary 7.** *Fix $\varepsilon > 0$. Choose $\alpha \leq \min\left\{\sqrt{\frac{\delta_0}{AT}}, L_1\sqrt{\frac{8\delta_0\varepsilon}{L_0 AT}}\right\}$. Then, if $T \geq \frac{64\delta_0 L_1^2}{\mu\zeta} + \frac{4L_0}{\mu\zeta}\ln\frac{2\delta_0}{\varepsilon}$, we have $\delta_T \leq \varepsilon$.*

*Proof of Corollary 9.* Since $0 \leq \tilde{T} \leq \frac{64\delta_0 L_1^2}{\mu\zeta}$, $A > 0$, $\alpha \leq \sqrt{\frac{\delta_0}{AT}}$, $\alpha \leq L_1\sqrt{\frac{8\delta_0\varepsilon}{L_0 AT}}$, due to the choice of $T \geq \frac{64\delta_0 L_1^2}{\mu\zeta} + \frac{4L_0}{\mu\zeta}\ln\frac{2\delta_0}{\varepsilon}$, we obtain that

$$\left(1 - \frac{\mu\zeta}{4L_0}\right)^{T-\tilde{T}}\delta_0 \leq e^{-\frac{\mu\zeta}{4L_0}\left(T-\tilde{T}\right)}\delta_0 \leq \frac{\varepsilon}{2},$$

and that

$$\frac{4L_0 A}{\mu\zeta} \cdot \frac{\delta_0}{AT} \leq \frac{\varepsilon}{2}.$$

Therefore, $\delta_T \leq \varepsilon$. $\square$

### C.3 Symmetric generalized-smooth non-convex functions

**Theorem 8.** *Let Assumptions 1 and 3 hold for functions $f$ and $\{f_m\}_{m=1}^{M}$. Choose any $T \geq 1$. For all $0 \leq t \leq T - 1$, denote*

$$\hat{a}_t = L_0 + L_1\|\nabla f(x_t)\|, \quad \tilde{a}_t = L_0 + L_1 \max_{mj}\|\nabla f_{mj}(x_t)\|.$$

*Put $\overline{\Delta}^\star = f^\star - \frac{1}{MN}\sum_{j=0}^{N-1}\sum_{m=1}^{M} f_m^\star$. Impose the following conditions on the client stepsizes $\alpha_t$ and global stepsizes $\gamma_t$ :*

$$\alpha_t \leq \min\left\{\frac{\sqrt{2}}{\sqrt{9N(N-1)}\tilde{a}_t}, \frac{\sqrt{\hat{a}_t}}{c\tilde{a}_t^{3/2}}\right\}, \quad \alpha_t \leq \max\left\{\frac{1}{4G_t L_1(N-1)}, \frac{1}{6\left(L_0 + L_1 G_t\right)(N-1)}\right\},$$

$$\frac{\zeta}{\hat{a}_t} \leq \gamma_t \leq \frac{1}{8\hat{a}_t}, \quad 0 \leq t \leq T - 1,$$

*where* $0 < \zeta \leq \frac{1}{4}$, $c \geq \sqrt{((N-1)(2N-1)+2(N+1))T}$ *and* $G_t = \max_{\substack{m=1,\dots,M \\ j=1,\dots,N}} \{\|\nabla f_{mj}(x_t)\|\}$. *Let* $\delta_0 \overset{def}{=} f(x_0) - f^\star$. *Then, the iterates* $\{x_t\}_{t=0}^{T-1}$ *of Algorithm 2 satisfy*

$$\mathbb{E}\left[\min_{t=0,\dots,T-1}\left\{\frac{\zeta}{8}\min\left\{\frac{\|\nabla f(x_t)\|^2}{L_0}, \frac{\|\nabla f(x_t)\|}{L_1}\right\}\right\}\right]$$

$$\leq \frac{8\left(1 + \frac{3\alpha_t^2 \tilde{a}_t^3}{8\hat{a}_t}((N-1)(2N-1)+2(N+1))\right)^T}{T}\delta_0 + \frac{6\alpha_t^2 \tilde{a}_t^3}{\hat{a}_t}(N+1)\Delta^\star.$$

**Lemma 8.** *Recall that* $\tilde{a}_t = L_0 + L_1 \max_{m,j}\|\nabla f_{mj}(x_t)\|$. *Then*

$$\frac{\gamma_t^2\|g_t\|^2}{2} \leq 3\tilde{a}_t\gamma_t^2 \frac{1}{MN}\sum_{j=1}^{N}\sum_{m=1}^{M}\|x_{t,j-1}^m - x_t\|^2 + \gamma_t^2\|\nabla f(x_t)\|^2.$$

*Proof.*

$$\frac{\|g_t\|^2}{2} = \frac{1}{2}\left\|\frac{1}{MN}\sum_{j=1}^{N}\sum_{m=1}^{M}\nabla f_{m,\pi_t(j)}(x_{t,j-1}^m)\right\|^2$$

$$= \left\|\frac{1}{MN}\sum_{j=1}^{N}\sum_{m=1}^{M}\left(\nabla f_{m,\pi_t(j)}(x_{t,j-1}^m) - \nabla f_{m,\pi_t(j)}(x_t)\right)\right\|^2 + \|\nabla f(x_t)\|^2$$

$$\leq \frac{1}{MN}\sum_{j=1}^{N}\sum_{m=1}^{M}(L_0 + L_1\|\nabla f_{m,\pi_t(j)}(x_t)\|)^2\|x_{t,j-1}^m - x_t\|^2\exp\left\{2L_1\|x_{t,j-1}^m - x_t\|\right\} + \|\nabla f(x_t)\|^2$$

$$\leq (\tilde{a}_t)^2\frac{1}{MN}\sum_{j=1}^{N}\sum_{m=1}^{M}\|x_{t,j-1}^m - x_t\|^2\exp\left\{2L_1\|x_{t,j-1}^m - x_t\|\right\} + \|\nabla f(x_t)\|^2.$$

Let $G_t = \max_{\substack{m=1,\dots,M \\ j=1,\dots,N}}\{\|\nabla f_{mj}(x_t)\|\}$. By the induction with respect to $j = 1,\dots,N$, we prove that, for any $m \in \{1,\dots,M\}$, we have $\|\nabla f_{m,\pi_t(j)}(x_{t,j-1}^m)\| \leq 2G_t$. Indeed, notice that

$$\|\nabla f_{m,\pi_t(j)}(x_{t,j-1}^m)\| \leq \|\nabla f_{m,\pi_t(j)}(x_t)\| + \|\nabla f_{m,\pi_t(j)}(x_{t,j-1}^m) - \nabla f_{m,\pi_t(j)}(x_t)\|$$

$$\leq G_t + (L_0 + L_1\|\nabla f_{m,\pi_t(j)}(x_t)\|)\exp\left\{L_1\|x_{t,j-1}^m - x_t\|\right\}\|x_{t,j-1}^m - x_t\|$$

$$\leq G_t + (L_0 + L_1\|\nabla f_{m,\pi_t(j)}(x_t)\|)\exp\left\{L_1\|x_{t,j-1}^m - x_t\|\right\}\|x_{t,j-1}^m - x_t\|$$

$$\leq G_t + (L_0 + L_1 G_t)\exp\left\{L_1\|x_{t,j-1}^m - x_t\|\right\}\|x_{t,j-1}^m - x_t\|.$$

By the induction assumption, we obtain that

$$\|x_{t,j-1}^m - x_t\| \leq \alpha_t\left\|\sum_{i=1}^{j-1}\nabla f_{m,\pi_t(i)}(x_{t,i-1}^m)\right\| \leq \alpha_t\sum_{i=1}^{j-1}\|\nabla f_{m,\pi_t(i)}(x_{t,i-1}^m)\|$$

$$\leq \alpha_t(j-1)\cdot 2G_t \leq \alpha_t(N-1)\cdot 2G_t.$$

Hence, we have that

$$\|\nabla f_{m,\pi_t(j)}(x_{t,j-1}^m)\| \leq G_t + (L_0 + L_1 G_t)\exp\left\{\alpha_t L_1(N-1)\cdot 2G_t\right\}\alpha_t(N-1)\cdot 2G_t.$$

Let $\alpha_t \leq \max\left\{\frac{1}{4G_t L_1(N-1)}, \frac{1}{6(L_0 + L_1 G_t)(N-1)}\right\}$. Then, $\|\nabla f_{m,\pi_t(j)}(x_{t,j-1}^m)\| \leq 2G_t$.

Therefore, we conclude that

$$\frac{\|g_t\|^2}{2} \leq 3(\tilde{a}_t)^2\frac{1}{MN}\sum_{j=1}^{N}\sum_{m=1}^{M}\|x_{t,j-1}^m - x_t\|^2 + \|\nabla f(x_t)\|^2.$$

$\square$

**Lemma 9.** *Let Assumptions 1 and 2 hold for functions $f$ and $\{f_m\}_{m=1}^M$ . Then, if we choose $\alpha_t \leq \frac{\sqrt{2}}{\sqrt{3n(n-1)(\tilde{a}_t)}}$, we get*

$$\mathbb{E}\left[\frac{1}{N}\sum_{j=1}^N \left\|x_{t,j}^m - x_t\right\|^2 \Big| x_t\right] \leq 4\alpha_t^2 \tilde{a}_t \left((N-1)(2N-1) + 2(N+1)(f(x_t) - f^\star)\right)$$

$$+ 8\alpha_t^2 \tilde{a}_t (N+1)\overline{\Delta}^\star. \tag{15}$$

*Proof.* From (9) we have

$$x_{t,j}^m = x_{t,j-1}^m - \alpha_t \nabla f_{m,\pi_t(j)}(x_{t,j-1}^m) = x_t - \sum_{k=1}^j \alpha_t \nabla f_{m,\pi_t(k)}(x_{t,k-1}^m).$$

Thus,

$$\left\|x_{t,j}^m - x_t\right\|^2 = \left\|\sum_{k=1}^j \alpha_t \nabla f_{m,\pi_t(k)}(x_{t,k-1}^m)\right\|^2$$

$$\leq 2\left\|\sum_{k=1}^j \alpha_t \left(\nabla f_{m,\pi_t(k)}(x_{t,k-1}^m) - \nabla f_{m,\pi_t(k)}(x_t)\right)\right\|^2$$

$$+ 2\left\|\sum_{k=1}^j \alpha_t \nabla f_{m,\pi_t(k)}(x_t)\right\|^2$$

$$\leq 2j \sum_{k=1}^j (\alpha_t)^2 \left(L_0 + L_1 \left\|\nabla f_{m,\pi_t(k)}(x_t)\right\|\right)^2 \exp\left\{2L_1 \left\|x_{t,k-1}^m - x_t\right\|\right\} \left\|x_{t,k-1}^m - x_t\right\|^2$$

$$+ 2\left\|\sum_{k=1}^j \alpha_t \nabla f_{m,\pi_t(k)}(x_t)\right\|^2.$$

Using last inequality, we get

$$\frac{1}{N}\sum_{j=1}^N \left\|x_{t,j}^m - x_t\right\|^2 \leq \sum_{j=1}^N \frac{6j}{N} \sum_{k=1}^j (\alpha_t)^2 \left(L_0 + L_1 \left\|\nabla f_{m,\pi_t(k)}(x_t)\right\|\right)^2 \left\|x_{t,k-1}^m - x_t\right\|^2$$

$$+ \frac{2}{N}\sum_{j=1}^N \left\|\sum_{k=1}^j \alpha_t \nabla f_{m,\pi_t(k)}(x_t)\right\|^2$$

$$\leq (\alpha_t)^2 (\tilde{a}_t)^2 \sum_{j=1}^N \frac{6j}{N} \sum_{k=1}^j \left\|x_{t,k-1}^m - x_t\right\|^2 + \frac{2\alpha_t^2}{N}\sum_{j=1}^N \left\|\sum_{k=1}^j \nabla f_{m,\pi_t(k)}(x_t)\right\|^2.$$

Let $\alpha_t \leq \frac{\beta}{\tilde{a}_t}$, where $\beta$ is constant. Then, we take a conditional expectation of the last inequality and get the following

$$\mathbb{E}\left[\frac{1}{N}\sum_{j=1}^N \left\|x_{t,j}^m - x_t\right\|^2 \Big| x_t\right] \leq \mathbb{E}\left[\frac{6\beta^2}{N}\sum_{j=1}^N j \sum_{k=1}^j \left\|x_{t,k-1}^m - x_t\right\|^2 \Big| x_t\right]$$

$$+ \frac{2\alpha_t^2}{N}\sum_{j=1}^N \mathbb{E}\left[\left\|\sum_{k=1}^j \nabla f_{m,\pi_t(k)}(x_t)\right\|^2 \Big| x_t\right].$$

Denote $\sigma_t^2 = \frac{1}{N}\sum_{j=0}^{N-1}\left\|\nabla f_{m,\pi_t(j)}(x_t) - f(x_t)\right\|^2$, and consider $\mathbb{E}\left[\left\|\sum_{k=1}^{j}\nabla f_{m,\pi_t(k)}(x_t)\right\|^2\middle| x_t\right]$. From Malinovsky et al. (2022, Lemma 1) we get

$$\mathbb{E}\left[\left\|\sum_{k=1}^{j}\nabla f_{m,\pi_t(k)}(x_t)\right\|^2\middle| x_t\right] \le j^2\left\|\nabla f(x_t)\right\| + j^2\mathbb{E}\left[\left\|\frac{1}{j}\sum_{k=1}^{j}\left(\nabla f_{m,\pi_t(k)}(x_t) - f(x_t)\right)\right\|^2\middle| x_t\right]$$

$$\le j^2\left\|\nabla f(x_t)\right\| + \frac{j(N-j)}{N-1}\sigma_t^2.$$

Thus,

$$\mathbb{E}\left[\frac{1}{N}\sum_{j=1}^{N}\left\|x_{t,j}^m - x_t\right\|^2\middle| x_t\right] \le \mathbb{E}\left[\frac{6\beta^2}{N}\sum_{j=1}^{N}j\sum_{k=1}^{j}\left\|x_{t,k}^m - x_t\right\|^2\middle| x_t\right]$$

$$+ \frac{2\alpha_t^2}{N}\sum_{j=1}^{N}\left(j^2\left\|\nabla f(x_t)\right\| + \frac{j(N-j)}{N-1}\sigma_t^2\right)$$

$$\le \mathbb{E}\left[\frac{6\beta^2}{N}\cdot\frac{N(N-1)}{2}\sum_{j=1}^{N}\left\|x_{t,j}^m - x_t\right\|^2\middle| x_t\right]$$

$$+ \frac{2\alpha_t^2}{N}\left(\frac{(N(N-1)(2N-1))}{6}\left\|\nabla f(x_t)\right\|^2\right)$$

$$+ \frac{2\alpha_t^2}{N}\frac{N(N+1)}{3}\sigma_t^2.$$

Further,

$$3\cdot\mathbb{E}\left[\frac{1}{N}\sum_{j=1}^{N}\left\|x_{t,j}^m - x_t\right\|^2\middle| x_t\right] \le 9\beta^2 N(N-1)\mathbb{E}\left[\frac{1}{N}\sum_{j=1}^{N}\left\|x_{t,j}^m - x_t\right\|^2\middle| x_t\right]$$

$$+ 2\alpha_t^2\left(\frac{(N-1)(2N-1)}{2}\left\|\nabla f(x_t)\right\|^2 + (N+1)\sigma_t^2\right).$$

Thus, if we choose $\beta \le \sqrt{\frac{2}{9N(N-1)}}$, $\eta$ such that $\eta\exp\eta = 1$, we get $1/\eta \le 2$ and

$$\mathbb{E}\left[\frac{1}{N}\sum_{j=1}^{N}\left\|x_{t,j}^m - x_t\right\|^2\middle| x_t\right] \le (3 - 9\beta^2 N(N-1))\mathbb{E}\left[\frac{1}{N}\sum_{j=1}^{N}\left\|x_{t,j}^m - x_t\right\|^2\middle| x_t\right]$$

$$\le 2\alpha_t^2\left(\frac{(N-1)(2N-1)}{2}\left\|\nabla f(x_t)\right\|^2 + (N+1)\sigma_t^2\right)$$

$$\le 2\alpha_t^2\Big(2(N-1)(2N-1)(f(x_t) - f^\star)(L_0 + L_1\left\|\nabla f(x_t)\right\|)$$

$$+ (N+1)\frac{1}{N}\sum_{j=1}^{N}\left\|\nabla f_{m,\pi_t(j)}(x_t)\right\|^2\Big)$$

$$\overset{\text{Lemma 2}}{\le} 4\alpha_t^2\Big((N-1)(2N-1)(f(x_t) - f^\star)\tilde{a}_t$$

$$+ 2(N+1)(\tilde{a}_t)\frac{1}{N}\sum_{j=1}^{N}(f_{mj}(x_t) - f_{mj}^\star)\Big)$$

$$\le 4\alpha_t^2\Big((N-1)(2N-1)(f(x_t) - f^\star)\tilde{a}_t$$

$$+ 2(N+1)(\tilde{a}_t)\frac{1}{N}\sum_{j=1}^{N}(f_{mj}(x_t) - f_{mj}^\star)\Big)$$

Now, adding and removing $f^\star$ to the sum factor on the right-hand side, we get

$$\mathbb{E}\left[\frac{1}{MN}\sum_{j=1}^{N}\sum_{m=1}^{M}\left\|x_{t,j}^{m}-x_t\right\|^2\,\bigg|\,x_t\right]\leq 4\alpha_t^2\tilde{a}_t(N-1)(2N-1)(f(x_t)-f^\star)$$

$$+8\alpha_t^2\tilde{a}_t(N+1)\frac{1}{NM}\sum_{j=1}^{N}\sum_{m=1}^{M}(f_{mj}(x_t)-f_{mj}^\star)$$

$$=4\alpha_t^2\tilde{a}_t\left((N-1)(2N-1)+2(N+1)\right)(f(x_t)-f^\star)$$

$$+8\alpha_t^2\tilde{a}_t(N+1)\overline{\Delta}^\star.$$

$\square$

*Proof of Theorem 8.* From Lemma 2 and (10) we get

$$f(x_{t+1})\leq f(x_t)-\gamma_t\left\langle\nabla f(x_t),g_t\right\rangle+(L_0+L_1\|\nabla f(x_t)\|)\exp\left\{L_1\gamma_t\|g_t\|\right\}\frac{\gamma_t^2\|g_t\|^2}{2}.$$

Additionally, from the fact that $2\left\langle a,b\right\rangle=-\|a-b\|^2+\|a\|^2+\|b\|^2$ we can get

$$f(x_{t+1})\leq f(x_t)-\gamma_t\left\langle\nabla f(x_t),g_t\right\rangle+(L_0+L_1\|\nabla f(x_t)\|)\exp\left\{L_1\gamma_t\|g_t\|\right\}\frac{\gamma_t^2\|g_t\|^2}{2}$$

$$\leq f(x_t)-\frac{\gamma_t}{2}\left(-\|\nabla f(x_t)-g_t\|^2+\|\nabla f(x_t)\|^2+\|g_t\|^2\right)$$

$$+(L_0+L_1\|\nabla f(x_t)\|)\exp\left\{L_1\gamma_t\|g_t\|\right\}\frac{\gamma_t^2\|g_t\|^2}{2}$$

$$\leq f(x_t)-\frac{\gamma_t}{2}\|\nabla f(x_t)\|^2+\frac{\gamma_t}{2}\|\nabla f(x_t)-g_t\|^2+(L_0+L_1\|\nabla f(x_t)\|)\exp\left\{L_1\gamma_t\|g_t\|\right\}\frac{\gamma_t^2\|g_t\|^2}{2}.$$

Consider $\frac{\gamma_t}{2}\|\nabla f(x_t)-g_t\|^2$ and denote $\hat{a}_t=(L_0+L_1\|\nabla f(x_t)\|)$ and $a_t=(L_0+L_1\max_m\|\nabla f_m(x_t)\|)$, then:

$$\frac{\gamma_t}{2}\|\nabla f(x_t)-g_t\|^2=\frac{\gamma_t}{2}\left\|\frac{1}{MN}\sum_{j=1}^{N}\sum_{m=1}^{M}\nabla f_{m,\pi_t(j)}(x_t)-\nabla f_{m,\pi_t(j)}(x_{t,j}^m)\right\|^2$$

$$\leq\frac{\gamma_t}{2}\frac{1}{MN}\sum_{j=1}^{N}\sum_{m=1}^{M}(L_0+L_1\|\nabla f_{m,\pi_t(j)}(x_t)\|)^2\|x_t-x_{t,j}^m\|^2\exp\left\{2L_1\left\|x_t-x_{t,j}^m\right\|\right\}$$

$$\leq\frac{3\gamma_t}{2}\tilde{a}_t^2\frac{1}{MN}\sum_{j=1}^{N}\sum_{m=1}^{M}\|x_t-x_{t,j}^m\|^2.$$

Let us consider $L_1 \gamma_t \|g_t\|$ now. Recall that $\alpha_t \leq \frac{\hat{a}_t}{4\tilde{a}_t(N-1)L_1 G_t}$.

$$
\begin{aligned}
L_1 \gamma_t \|g_t\| = L_1 \gamma_t & \left\| \frac{1}{MN} \sum_{j=1}^{N} \sum_{m=1}^{M} \nabla f_{m,\pi_t(j)}(x_{t,j-1}^m) \right\| \\
\leq L_1 \gamma_t & \left( \frac{1}{MN} \sum_{j=1}^{N} \sum_{m=1}^{M} \left\| \nabla f_{m,\pi_t(j)}(x_{t,j-1}^m) - \nabla f(x_t) \right\| + \|\nabla f(x_t)\| \right) \\
\leq L_1 \gamma_t & \left( \frac{\tilde{a}_t}{MN} \sum_{j=1}^{N} \sum_{m=1}^{M} \left\| x_{t,j-1}^m - x_t \right\| \exp\left\{ L_1 \left\| x_{t,j-1}^m - x_t \right\| \right\} + \|\nabla f(x_t)\| \right) \\
\leq & \frac{3 L_1 \gamma_t \tilde{a}_t}{MN} \sum_{j=1}^{N} \sum_{m=1}^{M} \left\| x_{t,j-1}^m - x_t \right\| + \frac{1}{4} \\
\leq & \frac{3 L_1 \tilde{a}_t}{4 \hat{a}_t MN} \sum_{j=1}^{N} \sum_{m=1}^{M} \left\| x_{t,j-1}^m - x_t \right\| + \frac{1}{4} \\
\leq & \frac{3\tilde{a}_t}{4\hat{a}_t} \cdot \alpha_t (N-1) L_1 G_t + \frac{1}{4} \\
\leq & \frac{1}{2}.
\end{aligned}
$$

From the above inequality and Lemma 6 we get

$$
\begin{aligned}
f(x_{t+1}) \leq & f(x_t) - \frac{\gamma_t}{2} \|\nabla f(x_t)\|^2 + \frac{3\gamma_t}{2} \tilde{a}_t^2 \frac{1}{MN} \sum_{j=1}^{N} \sum_{m=1}^{M} \|x_t - x_{t,j}^m\|^2 + \hat{a}_t \frac{\gamma_t^2 \|g_t\|^2}{2} \exp\left\{ L_1 \gamma_t \|g_t\| \right\} \\
\overset{(11)}{\leq} & f(x_t) - \frac{\gamma_t}{2} \|\nabla f(x_t)\|^2 + \frac{3\gamma_t}{2} \tilde{a}_t^2 \frac{1}{MN} \sum_{j=1}^{N} \sum_{m=1}^{M} \|x_t - x_{t,j}^m\|^2 \\
& + 2\hat{a}_t \tilde{a}_t^2 \gamma_t^2 \frac{1}{MN} \sum_{j=1}^{N} \sum_{m=1}^{M} \|x_{t,j}^m - x_t\|^2 + 2\hat{a}_t \gamma_t^2 \|\nabla f(x_t)\|^2 \\
\leq & f(x_t) + \left( 2\hat{a}_t \gamma_t^2 - \frac{\gamma_t}{2} \right) \|\nabla f(x_t)\|^2 + \left( 2\hat{a}_t \tilde{a}_t^2 \gamma_t^2 + \frac{3\gamma_t}{2} \tilde{a}_t^2 \right) \frac{1}{MN} \sum_{j=1}^{N} \sum_{m=1}^{M} \|x_{t,j}^m - x_t\|^2.
\end{aligned}
$$

Sine $\gamma_t \leq \frac{1}{8\hat{a}_t}$, we obtain

$$
f(x_{t+1}) \leq f(x_t) - \frac{\gamma_t}{4} \|\nabla f(x_t)\|^2 + \left( 2\hat{a}_t \tilde{a}_t^2 \gamma_t^2 + \frac{3\gamma_t}{2} \tilde{a}_t^2 \right) \frac{1}{MN} \sum_{j=1}^{N} \sum_{m=1}^{M} \|x_{t,j}^m - x_t\|^2.
$$

Now, if we take conditional expectation of this and use Lemma 9, we get

$$
\begin{aligned}
\mathbb{E}\left[ f(x_{t+1}) | x_t \right] \leq & f(x_t) - \frac{\gamma_t}{4} \|\nabla f(x_t)\|^2 \\
& + \left( 2\hat{a}_t \tilde{a}_t^2 \gamma_t^2 + \frac{3\gamma_t}{2} \tilde{a}_t^2 \right) \mathbb{E}\left[ \frac{1}{MN} \sum_{j=1}^{N} \sum_{m=1}^{M} \|x_{t,j}^m - x_t\|^2 \bigg| x_t \right] \\
\leq & f(x_t) - \frac{\gamma_t}{4} \|\nabla f(x_t)\|^2 \\
& + 2\alpha_t^2 \tilde{a}_t \left( 2\hat{a}_t \tilde{a}_t^2 \gamma_t^2 + \frac{3\gamma_t}{2} \tilde{a}_t^2 \right) \\
& \times \left( ((N-1)(2N-1) + 2(N+1)) (f(x_t) - f^\star) + 2(N+1)\overline{\Delta}^\star \right).
\end{aligned}
$$

Since $\gamma_t \leq \frac{1}{8\hat{a}_t}$, then

$$\frac{\gamma_t}{4} \|\nabla f(x_t)\|^2 \leq f(x_t) - \mathbb{E}\left[f(x_{t+1})|x_t\right] + \frac{\alpha_t^2 \tilde{a}_t^3}{2\hat{a}_t}\left(\left((N-1)(2N-1) + 2(N+1)\right)\delta_t + 2(N+1)\overline{\Delta}^\star\right).$$
(16)

Consider the left-hand side of (16). Due to the bounds $\frac{1}{4\hat{a}_t} \geq \gamma_t \geq \frac{\zeta}{\hat{a}_t}$ on $\gamma_t$, we have

$$\frac{\gamma_t}{4}\|\nabla f(x_t)\|^2 \geq \frac{\zeta \|\nabla f(x_t)\|^2}{4\hat{a}_t}.$$

Then, we get

$$\frac{\gamma_t}{4}\|\nabla f(x_t)\|^2 \geq \begin{cases} \frac{\zeta\|\nabla f(x_t)\|^2}{8L_0}, & \|\nabla f(x_t)\| \leq \frac{L_0}{L_1}, \\ \frac{\zeta\|\nabla f(x_t)\|}{8L_1}, & \|\nabla f(x_t)\| > \frac{L_0}{L_1} \end{cases}$$
$$= \frac{\zeta}{8}\min\left\{\frac{\|\nabla f(x_t)\|^2}{L_0}, \frac{\|\nabla f(x_t)\|}{L_1}\right\}$$
(17)

Denote $\delta_t \equiv f(x_t) - f^\star$, then from (16) and (17) we get

$$f(x_t) - f(x_{t+1}) + \frac{\alpha_t^2 \tilde{a}_t^3}{2\hat{a}_t}\left(\left((N-1)(2N-1) + 2(N+1)\right)\delta_t + 2(N+1)\overline{\Delta}^\star\right)$$
$$= \delta_t - \delta_{t+1} + \frac{\alpha_t^2 \tilde{a}_t^3}{2\hat{a}_t}\left(\left((N-1)(2N-1) + 2(N+1)\right)\delta_t + 2(N+1)\overline{\Delta}^\star\right)$$
$$\geq \frac{\zeta}{8}\min\left\{\frac{\|\nabla f(x_t)\|^2}{L_0}, \frac{\|\nabla f(x_t)\|}{L_1}\right\}$$

Let $\alpha_t \leq \frac{1}{c\tilde{a}_t}\cdot\sqrt{\frac{\hat{a}_t}{\tilde{a}_t}}$, where $c$ is a constant such that $\sqrt{((N-1)(2N-1) + 2(N+1))T} \leq c$. Now we take full expectation and use from Mishchenko et al. (2020, Lemma 6):

$$\mathbb{E}\left[\min_{t=0,\ldots,T-1}\left\{\frac{\zeta}{8}\min\left\{\frac{\|\nabla f(x_t)\|^2}{L_0}, \frac{\|\nabla f(x_t)\|}{L_1}\right\}\right\}\right]$$
$$\leq \frac{\left(1 + \frac{\alpha_t^2 \tilde{a}_t^3}{2\hat{a}_t}((N-1)(2N-1) + 2(N+1))\right)^T}{T}\delta_0 + \frac{\alpha_t^2 \tilde{a}_t^3}{\hat{a}_t}(N+1)\overline{\Delta}^\star.$$
$$\square$$

**Corollary 8.** *Fix* $\varepsilon > 0$. *Choose* $c = \sqrt{((N-1)(2N-1) + 2(N+1))T}$. *Let* $\alpha_t \leq 4\sqrt{\frac{\hat{a}_t}{\tilde{a}_t^3 T(N+1)\overline{\Delta}^\star}}$. *Then, if* $T \geq \frac{256\delta_0}{\zeta\varepsilon}$, *we have*

$$\mathbb{E}\left[\min_{t=0,\ldots,T-1}\left\{\min\left\{\frac{\|\nabla f(x_t)\|^2}{L_0}, \frac{\|\nabla f(x_t)\|}{L_1}\right\}\right\}\right] \leq \varepsilon.$$

*Proof of Corollary 8.* Since $c = \sqrt{((N-1)(2N-1) + 2(N+1))T}$ and $\alpha_t \leq \frac{1}{c\tilde{a}_t}\sqrt{\frac{\hat{a}_t}{\tilde{a}_t}}$, $\alpha_t \leq 4\sqrt{\frac{\hat{a}_t}{\tilde{a}_t^3 T(N+1)\overline{\Delta}^\star}}$, due to the choice of $T \geq \frac{256\delta_0}{\zeta\varepsilon}$, we obtain that

$$\frac{\left(1 + \frac{\alpha_t^2 \tilde{a}_t^3}{2\hat{a}_t}((N-1)(2N-1) + 2(N+1))\right)^T}{T}\delta_0 \leq \frac{e^{\frac{1}{2}}\delta_0}{T} \leq \frac{2\delta_0}{T} \leq \frac{\zeta\varepsilon}{16},$$

and that

$$\frac{\alpha_t^2 \tilde{a}_t^3}{\hat{a}_t}(N+1)\overline{\Delta}^\star \leq \frac{\zeta\varepsilon}{16}.$$

Therefore, $\mathbb{E}\left[\min_{t=0,\ldots,T-1}\left\{\min\left\{\frac{\|\nabla f(x_t)\|^2}{L_0}, \frac{\|\nabla f(x_t)\|}{L_1}\right\}\right\}\right] \leq \varepsilon$. Notice that the computation of $G_t$, $0 \leq t \leq T-1$, requires additional $T$ epochs. Therefore, the total number of epochs is at least $2T \geq \frac{512\delta_0}{\zeta\varepsilon}$. $\square$

## C.4 Symmetric generalized-smooth functions under PŁ-condition

**Theorem 9.** *Let Assumptions 1 and 3 hold for functions $f$, $\{f_m\}_{m=1}^{M}$ and $\{f_{mj}\}_{m=1,j=0}^{M,N-1}$. Let Assumption 4 hold. Choose $0 < \zeta \le \frac{1}{4}$. Let $\delta_0 \overset{def}{=} f(x_0) - f^\star$. Choose any integer $T > \frac{64\delta_0 L_1^2}{\mu\zeta}$. For all $0 \le t \le T - 1$, denote*

$$\hat{a}_t = L_0 + L_1\|\nabla f(x_t)\|, \quad a_t = L_0 + L_1 \max_m \|\nabla f_m(x_t)\|.$$

*Put $\overline{\Delta}^\star = f^\star - \frac{1}{MN}\sum_{m=1}^{M}\sum_{j=1}^{N}f_{mj}^\star$. Impose the following conditions on the client stepsizes $\alpha_t$ and global stepsizes $\gamma_t$ :*

$$\alpha_t \le \min\left\{\frac{\sqrt{2}}{\sqrt{9N(N-1)}\tilde{a}_t}, \frac{\sqrt{\hat{a}_t}}{c\tilde{a}_t^{3/2}},\right.$$

$$\sqrt{\frac{\hat{a}_t\mu\zeta}{16L_1^2\tilde{a}_t^3\left(\delta_t((N-1)(2N-1)+2(N+1))+2(N+1)\overline{\Delta}^\star\right)}}$$

$$\left.\sqrt{\frac{2\hat{a}_t\delta_0}{T\tilde{a}_t^3\left(\delta_t((N-1)(2N-1)+2(N+1))+2(N+1)\overline{\Delta}^\star\right)}}\right\},$$

$$\frac{\zeta}{\hat{a}_t} \le \gamma_t \le \frac{1}{8\hat{a}_t}, \quad 0 \le t \le T - 1,$$

*where $c \ge \sqrt{((N-1)(2N-1)+2(N+1))T}$. Let $\delta_0 \overset{def}{=} f(x_0) - f^\star$. Let $\tilde{T}$ be an integer such that $0 \le \tilde{T} \le \frac{64\delta_0 L_1^2}{\mu\zeta}$, $A > 0$ be a constant, $\alpha \le \sqrt{\frac{\delta_0}{AT}}$. Then, the iterates $\{x_t\}_{t=0}^{T-1}$ of Algorithm 2 satisfy*

$$\delta_T \le \left(1 - \frac{\mu\zeta}{4L_0}\right)^{T-\tilde{T}}\delta_0 + \frac{4L_0 A\alpha^2}{\mu\zeta},$$

*where $\delta_T \overset{def}{=} f(x_T) - f^\star$.*

*Proof of Theorem 9.* Let us follow the first steps of the proof of Theorem 3. Consider (16):

$$\frac{\gamma_t}{4}\|\nabla f(x_t)\|^2 \le f(x_t) - f(x_{t+1}) + \frac{\alpha_t^2\tilde{a}_t^3}{2\hat{a}_t}$$

$$\times\left(((N-1)(2N-1)+2(N+1))\delta_t + 2(N+1)\overline{\Delta}^\star\right).$$

Since $\gamma_t \ge \frac{\zeta}{\hat{a}_t}$, and $f$ satisfies Polyak–Łojasiewicz Assumption 4, we obtain that

$$\frac{\mu\zeta(f(x_t) - f^\star)}{2\hat{a}_t} \le f(x_t) - f(x_{t+1})$$

$$+ \frac{\alpha_t^2\tilde{a}_t^3}{2\hat{a}_t}\left(\delta_t((N-1)(2N-1)+2(N+1))+2(N+1)\overline{\Delta}^\star\right).$$

**1.** Let $\tilde{T}$ be the number of steps $t$, so that $\|\nabla f(\hat{x}_t)\| \ge \frac{L_0}{L_1}$. For such $t$, we have $L_0 + L_1\|\nabla f(x_t)\| = \hat{a}_t \le 2L_1\|\nabla f(x_t)\|$. Therefore, we get

$$\frac{\mu\zeta(f(x_t) - f^\star)}{4L_1\|\nabla f(x_t)\|} \le f(x_t) - f(x_{t+1})$$

$$+ \frac{\alpha_t^2\tilde{a}_t^3}{2\hat{a}_t}\left(\delta_t((N-1)(2N-1)+2(N+1))+2(N+1)\overline{\Delta}^\star\right).$$

Notice that the relation $\hat{a}_t \le 2L_1\|\nabla f(x_t)\|$ and Lemma 1 together imply

$$\frac{\|\nabla f(x_t)\|}{4L_1} \le \frac{\|\nabla f(x_t)\|^2}{2\hat{a}_t} \le f(x_t) - f^\star.$$

Hence, we have

$$\frac{\mu\zeta}{16L_1^2} \leq f(x_t) - f(x_{t+1}) + \frac{\alpha_t^2 \tilde{a}_t^3}{2\hat{a}_t}\left(\delta_t((N-1)(2N-1) + 2(N+1)) + 2(N+1)\overline{\Delta}^\star\right).$$

Subtracting $f^\star$ on both sides and introducing $\delta_t \stackrel{\text{def}}{=} f(x_t) - f^\star$, we obtain

$$\delta_{t+1} \leq \delta_t - \frac{\mu\zeta}{16L_1^2} + \frac{\alpha_t^2 \tilde{a}_t^3}{2\hat{a}_t}\left(\delta_t((N-1)(2N-1) + 2(N+1)) + 2(N+1)\overline{\Delta}^\star\right).$$

As $\alpha_t \leq \sqrt{\frac{\hat{a}_t \mu\zeta}{16L_1^2 \tilde{a}_t^3\left(\delta_t((N-1)(2N-1)+2(N+1))+2(N+1)\overline{\Delta}^\star\right)}}$, it follows that

$$\frac{\alpha_t^2 \tilde{a}_t^3}{2\hat{a}_t}\left(\delta_t((N-1)(2N-1) + 2(N+1)) + 2(N+1)\overline{\Delta}^\star\right) \leq \frac{\mu\zeta}{32L_1^2}.$$

Therefore, we get

$$\delta_{t+1} \leq \delta_t - \frac{\mu\zeta}{32L_1^2}.$$

**2.** Suppose now that $\|\nabla f(x_t)\| \leq \frac{L_0}{L_1}$. For such $t$, we have $L_0 + L_1\|\nabla f(x_t)\| = \hat{a}_p \leq 2L_0$. Hence,

$$\frac{\mu\zeta(f(x_t) - f^\star)}{4L_0} \leq f(x_t) - f(x_{t+1})$$

$$+ \frac{\alpha_t^2 \tilde{a}_t^3}{2\hat{a}_t}\left(\delta_t((N-1)(2N-1) + 2(N+1)) + 2(N+1)\overline{\Delta}^\star\right).$$

Subtracting $f^\star$ on both sides and introducing $\delta_t \stackrel{\text{def}}{=} f(x_t) - f^\star$, we obtain

$$\delta_{t+1} \leq \delta_t \rho + \frac{\alpha_t^2 \tilde{a}_t^3}{2\hat{a}_t}(\delta_t((N-1)(2N-1) + 2(N+1)) + 2(N+1)\overline{\Delta}^\star).$$

where $\rho \stackrel{\text{def}}{=} 1 - \frac{\mu\zeta}{4L_0}$. Let $\alpha_t \stackrel{\text{def}}{=} \alpha\hat{a}_t$ with $\hat{a}_t \leq \sqrt{\frac{2\hat{a}_t A}{\tilde{a}_t^3\left(\delta_t((N-1)(2N-1)+2(N+1))+2(N+1)\overline{\Delta}^\star\right)}}$ for some constant $A > 0$. Then,

$$\delta_{t+1} \leq \rho\delta_t + A\alpha^2.$$

Unrolling the recursion, we derive

$$\delta_T \leq \rho^{T-\tilde{T}}\delta_0 + A\alpha^2 \sum_{i=0}^{\infty} \rho^i - \frac{\mu\zeta}{32L_1^2}\sum_{i=0}^{N-1}\rho^i$$

$$\leq \rho^{T-\tilde{T}}\delta_0 + \frac{A\alpha^2}{1-\rho} - \frac{1-\rho^{\tilde{T}}}{1-\rho}\frac{\mu\zeta}{32L_1^2}.$$

Notice that $\delta_{t+1} \leq \delta_t + A\alpha^2$, which implies

$$\delta_T \leq \delta_0 + \left(T - \tilde{T}\right)A\alpha^2 - \tilde{T}\frac{\mu\zeta}{32L_1^2}.$$

Since $\alpha \leq \sqrt{\frac{\delta_0}{AT}}$, we conclude that

$$0 \leq \delta_T \leq 2\delta_0 - \tilde{T}\frac{\mu\zeta}{32L_1^2}, \quad \Rightarrow \tilde{T} \leq \frac{64\delta_0 L_1^2}{\mu\zeta}.$$

Therefore, for $T > \frac{64\delta_0 L_1^2}{\mu\zeta}$ we can guarantee that $T - \tilde{T} > 0$ and

$$\delta_T \leq \rho^{T-\tilde{T}}\delta_0 + \frac{A\alpha^2}{1-\rho} - \tilde{T}\rho^{\tilde{T}}\frac{\mu\zeta}{32L_1^2}$$

$$\leq \rho^{T-\tilde{T}}\delta_0 + \frac{A\alpha^2}{1-\rho}.$$

$\square$

**Corollary 9.** *Fix $\varepsilon > 0$. Choose $\alpha \leq \min\left\{\sqrt{\frac{\delta_0}{AT}}, L_1\sqrt{\frac{8\delta_0\varepsilon}{L_0 AT}}\right\}$. Then, if $T \geq \frac{64\delta_0 L_1^2}{\mu\zeta} + \frac{4L_0}{\mu\zeta}\ln\frac{2\delta_0}{\varepsilon}$, we have $\delta_T \leq \varepsilon$.*

*Proof of Corollary 9.* Since $0 \leq \tilde{T} \leq \frac{64\delta_0 L_1^2}{\mu\zeta}$, $A > 0$, $\alpha \leq \sqrt{\frac{\delta_0}{AT}}$, $\alpha \leq L_1\sqrt{\frac{8\delta_0\varepsilon}{L_0 AT}}$, due to the choice of $T \geq \frac{64\delta_0 L_1^2}{\mu\zeta} + \frac{4L_0}{\mu\zeta}\ln\frac{2\delta_0}{\varepsilon}$, we obtain that

$$\left(1 - \frac{\mu\zeta}{4L_0}\right)^{T-\tilde{T}}\delta_0 \leq e^{-\frac{\mu\zeta}{4L_0}(T-\tilde{T})}\delta_0 \leq \frac{\varepsilon}{2},$$

and that

$$\frac{4L_0 A}{\mu\zeta} \cdot \frac{\delta_0}{AT} \leq \frac{\varepsilon}{2}.$$

Therefore, $\delta_T \leq \varepsilon$. Notice that the computation of $G_t$, $0 \leq t \leq T-1$, requires additional $T$ epochs. Therefore, the total number of epochs is at least $2T \geq \frac{128\delta_0 L_1^2}{\mu\zeta} + \frac{8L_0}{\mu\zeta}\ln\frac{2\delta_0}{\varepsilon}$. $\qquad\square$

## D  PARTIAL PARTICIPATION

### D.1  ASYMMETRIC GENERALIZED-SMOOTH NON-CONVEX FUNCTIONS

**Theorem 4** *Let Assumptions 1 and 2 hold for functions $f$, $\{f_m\}_{m=1}^M$ and $\{f_{mj}\}_{m=1,j=1}^{M,N}$. Choose any $T \geq 1$. For all $0 \leq t \leq T-1$, denote*

$$\hat{a}_t = L_0 + L_1\|\nabla f(x_t)\|, \quad a_t = L_0 + L_1\max_m\|\nabla f_m(x_t)\|, \quad \tilde{a}_t = L_0 + L_1\max_{m,j}\|\nabla f_m^{\pi_j}(x_t)\|.$$

*Put $\Delta^\star = f^\star - \frac{1}{M}\sum_{m=1}^M f_m^\star$ and $\overline{\Delta}^\star = f^\star - \frac{1}{M}\sum_{m=1}^M \frac{1}{N}\sum_{j=1}^N f_{mj}^\star$. Impose the following conditions on the local stepsizes $\gamma_t$, server stepsizes $\eta_t$, global stepsizes $\theta_t$:*

$$\gamma_t N R \leq \eta_t R \leq \min\left\{\frac{1}{16\hat{a}_t}, \frac{2\hat{a}_t}{c}\sqrt{\frac{1}{a_t(2\hat{a}_t\tilde{a}_t^2 + \hat{a}_t^3)}}\right\}, \quad \gamma_t \leq \frac{2\hat{a}_t}{cRN}\sqrt{\frac{1}{\tilde{a}_t(2\hat{a}_t\tilde{a}_t^2 + \hat{a}_t^3)}},$$

$$\frac{\zeta}{\hat{a}_t} \leq \theta_t \leq \frac{1}{4\hat{a}_t}, \quad 0 \leq t \leq T-1,$$

*where $c \geq \sqrt{T}$, $0 < \zeta \leq \frac{1}{4}$. Let $\delta_0 \stackrel{def}{=} f(x_0) - f^\star$. Then, the iterates $\{x_t\}_{t=0}^{T-1}$ of Algorithm 3 satisfy*

$$\mathbb{E}\left[\min_{0\leq t\leq T-1}\left\{\frac{\zeta}{8}\min\left\{\frac{\|\nabla f(x_t)\|^2}{L_0}, \frac{\|\nabla f(x_t)\|}{L_1}\right\}\right\}\right]$$

$$\leq \frac{\left(1 + \frac{2\hat{a}_t\tilde{a}_t^2 + \hat{a}_t^3}{4\hat{a}_t^2}\left(\eta_t^2 a_t + \eta_t^2 R^2\hat{a}_t + \gamma_t^2 N\tilde{a}_t + \eta_t^2 R a_t\right)\right)^T}{T}\delta_0$$

$$+ \frac{2\hat{a}_t\tilde{a}_t^2 + \hat{a}_t^3}{4\hat{a}_t^2}\left(\eta_t^2 a_t\Delta^\star + \gamma_t^2 N\tilde{a}_t\overline{\Delta}^\star + \eta_t^2 R a_t\Delta^\star\right).$$

We need to use the following relations to establish convergence guarantees:

$$x_t^R = x_t - \eta_t\sum_{r=0}^{R-1}\frac{1}{C}\sum_{m\in S_t^{\lambda r}}\frac{1}{N}\sum_{j=0}^{N-1}\nabla f_m^{\pi_j}\left(x_{m,t}^{r,j}\right),$$

$$x_{m,t}^{r,j} = x_t - \eta_t\sum_{k=0}^{r-1}\frac{1}{C}\sum_{m\in S_t^{\lambda k}}\frac{1}{N}\sum_{j=0}^{N-1}\nabla f_m^{\pi_j}\left(x_{m,t}^{k,j}\right) - \gamma_t\sum_{l=0}^{j-1}\nabla f_m^{\pi_l}\left(x_{m,t}^{r,l}\right),$$

$$x_{t+1} = x_t - \frac{\theta_t}{\eta_t R}\left(x_t - x_t^R\right).$$

We assume that the whole sum is zero when the upper summation index is smaller than the lower index. We can derive the following recursion from the above relations:

$$x_t - x_{t+1} = \frac{\theta_t}{\eta_t R} \left( x_t - x_t^R \right)$$

$$= \frac{\theta_t}{R} \sum_{r=0}^{R-1} \frac{1}{C} \sum_{m \in S_t^{\lambda r}} \frac{1}{N} \sum_{j=0}^{N-1} \nabla f_m^{\pi^j} \left( x_{m,t}^{r,j} \right).$$

Further, the first statement of Lemma 1 yields the following inequality:

$$f(x_{t+1}) \le f(x_t) - \langle \nabla f(x_t), x_t - x_{t+1} \rangle + (L_0 + L_1 \|\nabla f(x_t)\|) \frac{\|x_t - x_{t+1}\|^2}{2}.$$

We deal with the last term, using the second statement of Lemma 1:

$$\|x_t - x_{t+1}\|^2 = \theta_t^2 \left\| \frac{1}{R} \sum_{r=0}^{R-1} \frac{1}{C} \sum_{m \in S_t^{\lambda r}} \frac{1}{N} \sum_{j=0}^{N-1} \nabla f_m^{\pi^j} \left( x_{m,t}^{r,j} \right) \right\|^2$$

$$\le 2\theta_t^2 \left\| \frac{1}{R} \sum_{r=0}^{R-1} \frac{1}{C} \sum_{m \in S_t^{\lambda r}} \frac{1}{N} \sum_{j=0}^{N-1} \left( \nabla f_m^{\pi^j} \left( x_{m,t}^{r,j} \right) - \nabla f(x_t) \right) \right\|^2 + 2\theta_t^2 \|\nabla f(x_t)\|^2$$

$$\le \frac{2\theta_t^2}{RCN} (L_0 + L_1 \|\nabla f(x_t)\|)^2 \sum_{r=0}^{R-1} \sum_{m \in S_t^{\lambda r}} \sum_{j=0}^{N-1} \left\| x_t - x_{m,t}^{r,j} \right\|^2$$

$$+ 4\theta_t^2 (L_0 + L_1 \|\nabla f(x_t)\|) (f(x_t) - f^\star).$$

We use the following notation: $\hat{a}_t = L_0 + L_1 \|\nabla f(x_t)\|$, $a_t = L_0 + L_1 \max_m \|\nabla f_m(x_t)\|$, $\tilde{a}_t = L_0 + L_1 \max_{m,j} \left\| \nabla f_m^{\pi^j} (x_t) \right\|$. Next, we have that

$$\left\| x_t - x_{m,t}^{r,j} \right\|^2 = \left\| \eta_t \sum_{k=0}^{r-1} \frac{1}{C} \sum_{m \in S_t^{\lambda k}} \frac{1}{N} \sum_{j=0}^{N-1} \nabla f_m^{\pi^j} \left( x_{m,t}^{k,j} \right) + \gamma_t \sum_{l=0}^{j-1} \nabla f_m^{\pi^l} \left( x_{m,t}^{r,l} \right) \right\|^2$$

$$\le 2 \left\| \eta_t \sum_{k=0}^{r-1} \frac{1}{C} \sum_{m \in S_t^{\lambda k}} \frac{1}{N} \sum_{j=0}^{N-1} \nabla f_m^{\pi^j} \left( x_{m,t}^{k,j} \right) \right\|^2 + 2 \left\| \gamma_t \sum_{l=0}^{j-1} \nabla f_m^{\pi^l} \left( x_{m,t}^{r,l} \right) \right\|^2.$$

Using Young's inequality, we obtain

$$\left\| x_t - x_{m,t}^{r,j} \right\|^2 \le 4\eta_t^2 \left\| \sum_{k=0}^{r-1} \frac{1}{C} \sum_{m \in S_t^{\lambda k}} \frac{1}{N} \sum_{j=0}^{N-1} \left( \nabla f_m^{\pi^j} \left( x_{m,t}^{k,j} \right) - \nabla f_m^{\pi^j} (x_t) \right) \right\|^2$$

$$+ 4\eta_t^2 \left\| \sum_{k=0}^{r-1} \frac{1}{C} \sum_{m \in S_t^{\lambda k}} \frac{1}{N} \sum_{j=0}^{N-1} \nabla f_m^{\pi^j} (x_t) \right\|^2$$

$$+ 4\gamma_t^2 \left\| \sum_{l=0}^{j-1} \left( \nabla f_m^{\pi^l} \left( x_{m,t}^{r,l} \right) - f_m^{\pi^l} (x_t) \right) \right\|^2$$

$$+ 4\gamma_t^2 \left\| \sum_{l=0}^{j-1} \nabla f_m^{\pi^l} (x_t) \right\|^2.$$

Using Malinovsky et al. (2022, Lemma 1), we derive the following upper bound on $\left\| x_t - x_{m,t}^{r,j} \right\|^2$:

$$
\begin{aligned}
\left\| x_t - x_{m,t}^{r,j} \right\|^2 \leq\ & 4\eta_t^2 r^2 \left(\hat{a}_t\right)^2 \frac{1}{rCN} \sum_{k=0}^{r-1} \sum_{m \in S_t^{\lambda k}} \sum_{j=0}^{N-1} \left\| x_t - x_{m,t}^{k,j} \right\|^2 \\
& + 4\eta_t^2 \frac{1}{N^2 C^2} \left( N^2 C^2 r^2 \|\nabla f(x_t)\|^2 + \frac{Cr\left(M - Cr\right)}{M-1} \sigma_t^2 \right) \\
& + 4\gamma_t^2 j \left(\hat{a}_t\right)^2 \sum_{l=0}^{j-1} \left\| x_t - x_{m,t}^{r,l} \right\|^2 \\
& + 4\gamma_t^2 \left( j^2 \|\nabla f_m(x_t)\|^2 + \frac{j(N-j)}{N-1} \sigma_{m,t}^2 \right),
\end{aligned}
$$

where

$$
\sigma_t^2 = \frac{1}{MN} \sum_{m=1}^{M} \sum_{j=0}^{N-1} \left\| \nabla f_m^{\pi_j}(x_t) - \nabla f_m(x_t) \right\|^2,
$$

$$
\sigma_{m,t}^2 = \frac{1}{N} \sum_{j=0}^{N-1} \left\| \nabla f_m^{\pi_j}(x_t) - \nabla f_m(x_t) \right\|^2.
$$

Using this bound on $\left\| x_t - x_{m,t}^{r,j} \right\|^2$, for $V_t \overset{\text{def}}{=} \frac{1}{CRN} \sum_{r=0}^{R-1} \sum_{m \in S_t^{\lambda r}} \sum_{j=0}^{N-1} \left\| x_{m,t}^{r,j} - x_t \right\|^2$, we obtain

$$
\begin{aligned}
\mathbb{E}\left[V_t\right] = &\ \frac{1}{CRN} \sum_{r=0}^{R-1} \sum_{m \in S_t^{\lambda r}} \sum_{j=0}^{N-1} \mathbb{E}\left\| x_{m,t}^{r,j} - x_t \right\|^2 \\
\leq &\ \frac{(\hat{a}_t)^2}{CRN} \\
& \times \sum_{r=0}^{R-1} \sum_{m \in S_t^{\lambda r}} \sum_{j=0}^{N-1} \left( 4\eta_t^2 r^2 \frac{1}{rCN} \sum_{k=0}^{r-1} \sum_{m \in S_t^{\lambda r}} \sum_{j=0}^{N-1} \left\| x_t - x_{m,t}^{k,j} \right\|^2 + 4\gamma_t^2 j \sum_{l=0}^{j-1} \left\| x_{m,t}^{r,l} - x_t \right\|^2 \right) \\
& + \frac{1}{CRN} \sum_{r=0}^{R-1} \sum_{m \in S_t^{\lambda r}} \sum_{j=0}^{N-1} \left( 4\gamma_t^2 \left( j^2 \|\nabla f_m(x_t)\|^2 + \frac{j(N-j)}{N-1} \sigma_{m,t}^2 \right) \right) \\
& + \frac{1}{CRN} \sum_{r=0}^{R-1} \sum_{m \in S_t^{\lambda r}} \sum_{j=0}^{N-1} \left( 4\eta_t^2 \frac{1}{N^2 C^2} \left( N^2 C^2 r^2 \|\nabla f(x_t)\|^2 + \frac{Cr(M - Cr)}{M-1} \sigma_t^2 \right) \right).
\end{aligned}
$$

Recall that $\gamma_t N R \leq \eta_t R \leq \frac{1}{16\hat{a}_t}$. Summing over indices, we arrive at

$$
\begin{aligned}
\mathbb{E}\left[V_t\right] \leq &\ \frac{R(R-1)}{2} 4\eta_t^2 \left(\hat{a}_t\right)^2 \mathbb{E}\left[V_t\right] + \frac{M(M-1)}{2} 4\gamma_t^2 \left(\hat{a}_t\right)^2 \mathbb{E}\left[V_t\right] \\
& + \frac{2}{3} \gamma_t^2 \frac{1}{M} \sum_{m=1}^{M} \|\nabla f_m(x_t)\|^2 (N-1)(2M-1) + \frac{2}{3} \gamma_t^2 (N+1) \frac{1}{M} \sum_{m=1}^{M} \sigma_{m,t}^2 \\
& + \frac{2}{3} \eta_t^2 \|\nabla f(x_t)\|^2 (R-1)(2R-1) + \frac{2}{3} \frac{M-C}{(M-1)C} \eta_t^2 \frac{R+1}{N^2} \sigma_t^2 \\
\leq &\ 2\eta_t^2 \left(\hat{a}_t\right)^2 \left(1 + R^2\right) \mathbb{E}\left[V_t\right] + \frac{2}{3} \gamma_t^2 \frac{1}{M} \sum_{m=1}^{M} \|\nabla f_m(x_t)\|^2 (N-1)(2M-1) \\
& + \frac{2}{3} \eta_t^2 \|\nabla f(x_t)\|^2 (R-1)(2R-1) + \frac{2}{3} \gamma_t^2 (N+1) \frac{1}{M} \sum_{m=1}^{M} \sigma_{m,t}^2 \\
& + \frac{2}{3} \eta_t^2 \frac{R+1}{N^2} \frac{M-C}{(M-1)C} \sigma_t^2.
\end{aligned}
$$

To derive the bound on $\mathbb{E}\left[V_t\right]$ we need to require that $\gamma_t NR \leq \eta_t R \leq \frac{1}{16\hat{a}_t}$ to have $1 - 2\eta_t^2\left(\hat{a}_t\right)^2\left(1 + R^2\right) > 0$. Using Lemma 1, we have

$$
\begin{aligned}
\mathbb{E}\left[V_t\right] &\leq 2\gamma_t^2 N^2 \frac{1}{M}\sum_{m=1}^{M}\|\nabla f_m(x_t)\|^2 + 2\eta_t^2 R^2\|\nabla f(x_t)\|^2 \\
&+ 2\gamma_t^2 N \frac{1}{M}\sum_{m=1}^{M}\sigma_{m,t}^2 + 2\eta_t^2 \frac{R}{N^2}\frac{M-C}{(M-1)C}\sigma_t^2 \\
&\leq 4\gamma_t^2 N^2 \frac{1}{M}\sum_{m=1}^{M}\left(L_0 + L_1\|\nabla f_m(x_t)\|\right)\left(f_m(x_t) - f_m^\star\right) \\
&+ 4\eta_t^2 R^2\hat{a}_t\left(f(x_t) - f(x_\star)\right) + 2\gamma_t^2 N \frac{1}{M}\sum_{m=1}^{M}\frac{1}{N}\sum_{j=0}^{N-1}\|\nabla f_m^{\pi_j}(x_t)\|^2 \\
&+ 2\eta_t^2 R \frac{M-C}{(M-1)C}\frac{1}{M}\sum_{m=1}^{M}\|\nabla f_m(x_t)\|^2 \\
&\leq 4\eta_t^2 \frac{1}{M}\sum_{m=1}^{M}\left(L_0 + L_1\|\nabla f_m(x_t)\|\right)\left(f_m(x_t) - f_m^\star\right) \\
&+ 4\eta_t^2 R^2\hat{a}_t\left(f(x_t) - f(x_\star)\right) \\
&+ 4\gamma_t^2 N \frac{1}{M}\sum_{m=1}^{M}\frac{1}{N}\sum_{j=0}^{N-1}\left(L_0 + L_1\|\nabla f_m^{\pi_j}(x_t)\|\right)\left(f_m^{\pi_j}(x_t) - f_m^{\pi_j,\star}\right) \\
&+ 4\eta_t^2 R \frac{M-C}{(M-1)C}\frac{1}{M}\sum_{m=1}^{M}\left(L_0 + L_1\|\nabla f_m(x_t)\|\right)\left(f_m(x_t) - f_m^\star\right).
\end{aligned}
$$

The bound for $\mathbb{E}\left[V_t\right]$ is given by the following:

$$
\begin{aligned}
\mathbb{E}\left[V_t\right] &\leq 4\eta_t^2 a_t\left(f(x_t) - f^\star + \left(f^\star - \frac{1}{M}\sum_{m=1}^{M}f_m^\star\right)\right) + 4\eta_t^2 R^2\hat{a}_t\left(f(x_t) - f^\star\right) \\
&+ 4\gamma_t^2 N\tilde{a}_t\left(f(x_t) - f^\star + \left(f^\star - \frac{1}{M}\sum_{m=1}^{M}\frac{1}{N}\sum_{j=0}^{N-1}f_m^{\pi_j,\star}\right)\right) \\
&+ 4\eta_t^2 Ra_t\frac{M-C}{(M-1)C}\left(f(x_t) - f^\star + \left(f^\star - \frac{1}{M}\sum_{m=1}^{M}f_m^\star\right)\right).
\end{aligned}
$$

Recall that $\Delta^\star = f^\star - \frac{1}{M}\sum_{m=1}^{M}f_m^\star$, $\overline{\Delta}^\star = f^\star - \frac{1}{M}\sum_{m=1}^{M}\frac{1}{N}\sum_{j=0}^{N-1}f_m^{\pi_j,\star}$. Therefore,

$$
\begin{aligned}
\mathbb{E}\left[V_t\right] &\leq 4\eta_t^2 a_t\left(f(x_t) - f^\star + \Delta^\star\right) + 4\eta_t^2 R^2\hat{a}_t\left(f(x_t) - f^\star\right) \\
&+ 4\gamma_t^2 N\tilde{a}_t\left(f(x_t) - f^\star + \overline{\Delta}^\star\right) + 4\eta_t^2 Ra_t\frac{M-C}{(M-1)C}\left(f(x_t) - f^\star + \Delta^\star\right).
\end{aligned}
$$

Rewriting, we obtain

$$
\begin{aligned}
\mathbb{E}\left[V_t\right] &\leq 4\left(f(x_t) - f^\star\right)\left(\eta_t^2 a_t + \eta_t^2 R^2\hat{a}_t + \gamma_t^2 N\tilde{a}_t + \eta_t^2 Ra_t\frac{M-C}{(M-1)C}\right) \\
&+ 4\eta_t^2 a_t\Delta^\star + 4\gamma_t^2 N\tilde{a}_t\overline{\Delta}^\star + 4\eta_t^2 Ra_t\frac{M-C}{(M-1)C}\Delta^\star \\
&\leq 4\left(f(x_t) - f^\star\right)\left(\eta_t^2 a_t + \eta_t^2 R^2\hat{a}_t + \gamma_t^2 N\tilde{a}_t + \eta_t^2 Ra_t\right) \\
&+ 4\eta_t^2 a_t\Delta^\star + 4\gamma_t^2 N\tilde{a}_t\overline{\Delta}^\star + 4\eta_t^2 Ra_t\Delta^\star.
\end{aligned}
$$

Following this, we need to establish a bound for the scalar product

$$-\langle \nabla f\left(x_t\right), x_t - x_{t+1} \rangle = \theta_t \Big\langle \nabla f\left(x_t\right), -\frac{1}{R} \sum_{r=0}^{R-1} \frac{1}{C} \sum_{m \in S_t^{\lambda r}} \frac{1}{N} \sum_{j=0}^{N-1} \nabla f_m^{\pi^j}\left(x_{m,t}^{r,j}\right) \Big\rangle.$$

Using the identity $2\langle a, b \rangle = \|a + b\|^2 - \|a\|^2 - \|b\|^2$, we obtain

$$-\langle \nabla f\left(x_t\right), x_t - x_{t+1} \rangle = - \left( \frac{\theta_t}{2} \|\nabla f\left(x_t\right)\|^2 + \frac{\theta_t}{2} \left\| \frac{1}{R} \sum_{r=0}^{R-1} \frac{1}{C} \sum_{m \in S_t^{\lambda r}} \frac{1}{N} \sum_{j=0}^{N-1} \nabla f_m^{\pi^j}\left(x_{m,t}^{r,j}\right) \right\|^2 \right)$$

$$+ \frac{\theta_t}{2} \left\| \nabla f(x_t) - \frac{1}{R} \sum_{r=0}^{R-1} \frac{1}{C} \sum_{m \in S_t^{\lambda r}} \frac{1}{N} \sum_{j=0}^{N-1} \nabla f_m^{\pi^j}\left(x_{m,t}^{r,j}\right) \right\|^2$$

$$= - \left( \frac{\theta_t}{2} \|\nabla f\left(x_t\right)\|^2 + \frac{\theta_t}{2} \left\| \frac{1}{R} \sum_{r=0}^{R-1} \frac{1}{C} \sum_{m \in S_t^{\lambda r}} \frac{1}{N} \sum_{j=0}^{N-1} \nabla f_m^{\pi^j}\left(x_{m,t}^{r,j}\right) \right\|^2 \right)$$

$$+ \frac{\theta_t}{2} \left\| \frac{1}{R} \sum_{r=0}^{R-1} \frac{1}{C} \sum_{m \in S_t^{\lambda r}} \frac{1}{N} \sum_{j=0}^{N-1} \left( \nabla f_m^{\pi^j}\left(x_{m,t}^{r,j}\right) - \nabla f_m^{\pi^j}\left(x_t\right) \right) \right\|^2.$$

Using Lemma 1 and omitting one of the terms, we get

$$-\langle \nabla f\left(x_t\right), x_t - x_{t+1} \rangle \leq -\frac{\theta_t}{2} \|\nabla f\left(x_t\right)\|^2 + \frac{\theta_t}{2} \left(\tilde{a}_t\right)^2 \frac{1}{R} \sum_{r=0}^{R-1} \frac{1}{C} \sum_{m \in S_t^{\lambda r}} \frac{1}{N} \sum_{j=0}^{N-1} \left\| x_{m,t}^{r,j} - x_t \right\|^2.$$

Taking the expectation with respect to the randomness of the algorithm, we have

$$\mathbb{E}\left[f(x_{t+1})\right] \leq f(x_t) - \frac{\theta_t}{2} \|\nabla f\left(x_t\right)\|^2$$

$$+ \frac{\theta_t \tilde{a}_t^2}{2R} \sum_{r=0}^{R-1} \frac{1}{C} \sum_{m \in S_t^{\lambda r}} \frac{1}{N} \sum_{j=0}^{N-1} \left\| x_{m,t}^{r,j} - x_t \right\|^2$$

$$+ \frac{\hat{a}_t}{2} \|x_t - x_{t+1}\|^2.$$

Recalling the definition of $\mathrm{E}\left[V_t\right]$ and taking the conditional expectation, we obtain

$$\mathrm{E}\left[f(x_{t+1}) \mid x_t\right] \leq f(x_t) - \frac{\theta_t}{2} \|\nabla f\left(x_t\right)\|^2 + \frac{\theta_t \tilde{a}_t^2}{2} \mathrm{E}\left[V_t\right] + \theta_t^2 \hat{a}_t \|\nabla f(x_t)\|^2 + \theta_t^2 \hat{a}_t^3 \mathrm{E}\left[V_t\right]$$

$$= f(x_t) - \frac{\theta_t}{2} \left(1 - 2\theta_t \hat{a}_t\right) \|\nabla f\left(x_t\right)\|^2 + \frac{\theta_t \tilde{a}_t^2}{2} \mathrm{E}\left[V_t\right] + \theta_t^2 \hat{a}_t^3 \mathrm{E}\left[V_t\right].$$

Using the fact that $\theta_t \leq \frac{1}{4\hat{a}_t}$, we arrive at

$$\mathrm{E}\left[f(x_{t+1}) \mid x_t\right] \leq f(x_t) - \frac{\theta_t}{4} \|\nabla f\left(x_t\right)\|^2 + \frac{\tilde{a}_t^2}{8\hat{a}_t} \mathrm{E}\left[V_t\right] + \frac{\hat{a}_t^3}{16\hat{a}_t^2} \mathrm{E}\left[V_t\right].$$

Recalling the bound on $\mathrm{E}\left[V_t\right]$, we obtain

$$\mathrm{E}\left[f(x_{t+1}) \mid x_t\right] \leq f(x_t) - \frac{\theta_t}{4} \|\nabla f\left(x_t\right)\|^2 + \frac{\tilde{a}_t^2}{8\hat{a}_t} \mathrm{E}\left[V_t\right] + \frac{\hat{a}_t^3}{16\hat{a}_t^2} \mathrm{E}\left[V_t\right]$$

$$\leq f(x_t) - \frac{\theta_t}{4} \|\nabla f\left(x_t\right)\|^2$$

$$+ \left( \frac{\tilde{a}_t^2}{2\hat{a}_t} + \frac{\hat{a}_t^3}{4\hat{a}_t^2} \right) \left(f(x_t) - f^\star\right) \left(\eta_t^2 a_t + \eta_t^2 R^2 \hat{a}_t + \gamma_t^2 N \tilde{a}_t + \eta_t^2 R a_t\right)$$

$$+ \left( \frac{\tilde{a}_t^2}{2\hat{a}_t} + \frac{\hat{a}_t^3}{4\hat{a}_t^2} \right) \left(\eta_t^2 a_t \Delta^\star + \gamma_t^2 N \tilde{a}_t \overline{\Delta}^\star + \eta_t^2 R a_t \Delta^\star\right). \tag{18}$$

Using the fact that $\theta_t \geq \frac{\zeta}{\hat{a}_t}$, we get that

$$\frac{\theta_t \|\nabla f(x_t)\|^2}{4} \geq \frac{\zeta \|\nabla f(x_t)\|^2}{4\hat{a}_t}.$$

Therefore,

$$\frac{\theta_t \|\nabla f(x_t)\|^2}{4} \geq \begin{cases} \frac{\zeta \|\nabla f(x_t)\|^2}{8L_0}, & \|\nabla f(x_t)\| \leq \frac{L_0}{L_1}, \\ \frac{\zeta \|\nabla f(x_t)\|}{8L_1}, & \|\nabla f(x_t)\| > \frac{L_0}{L_1}, \end{cases} = \frac{\zeta}{8} \min \left\{ \frac{\|\nabla f(x_t)\|^2}{L_0}, \frac{\|\nabla f(x_t)\|}{L_1} \right\}.$$

Denote $\delta_t \stackrel{\text{def}}{=} f(x_t) - f^\star$. Then we have

$$\frac{\zeta}{8} \min \left\{ \frac{\|\nabla f(x_t)\|^2}{L_0}, \frac{\|\nabla f(x_t)\|}{L_1} \right\} \leq -\delta_{t+1} + \delta_t$$

$$+ \frac{2\hat{a}_t \tilde{a}_t^2 + \hat{a}_t^3}{4\hat{a}_t^2} \left( \eta_t^2 a_t + \eta_t^2 R^2 \hat{a}_t + \gamma_t^2 N \tilde{a}_t + \eta_t^2 R a_t \right) \delta_t$$

$$+ \frac{2\hat{a}_t \tilde{a}_t^2 + \hat{a}_t^3}{4\hat{a}_t^2} \left( \eta_t^2 a_t \Delta^\star + \gamma_t^2 N \tilde{a}_t \overline{\Delta}^\star + \eta_t^2 R a_t \Delta^\star \right).$$

Recall that $\eta_t \leq \frac{2\hat{a}_t}{cR} \sqrt{\frac{1}{a_t(2\hat{a}_t \tilde{a}_t^2 + \hat{a}_t^3)}}$, $\gamma_t \leq \frac{2\hat{a}_t}{cRN} \sqrt{\frac{1}{\tilde{a}_t(2\hat{a}_t \tilde{a}_t^2 + \hat{a}_t^3)}}$, $c \geq \sqrt{T}$. Using Mishchenko et al. (2020, Lemma 6), we appear at

$$\min_{t=0,1,\ldots T-1} \left\{ \frac{\zeta}{8} \min \left\{ \frac{\|\nabla f(x_t)\|^2}{L_0}, \frac{\|\nabla f(x_t)\|}{L_1} \right\} \right\}$$

$$\leq \frac{\left(1 + \frac{2\hat{a}_t \tilde{a}_t^2 + \hat{a}_t^3}{4\hat{a}_t^2} \left( \eta_t^2 a_t + \eta_t^2 R^2 \hat{a}_t + \gamma_t^2 N \tilde{a}_t + \eta_t^2 R a_t \right) \right)^T}{T} \delta_0$$

$$+ \frac{2\hat{a}_t \tilde{a}_t^2 + \hat{a}_t^3}{4\hat{a}_t^2} \left( \eta_t^2 a_t \Delta^\star + \gamma_t^2 N \tilde{a}_t \overline{\Delta}^\star + \eta_t^2 R a_t \Delta^\star \right).$$

**Corollary 4.** *Fix $\varepsilon > 0$. Choose $c = 2\sqrt{T}$. Let $\eta_t \leq 2\hat{a}_t \sqrt{\frac{3\delta_0}{2a_t(2\hat{a}_t \tilde{a}_t^2 + \hat{a}_t^3) \Delta^\star RT}}$, $\gamma_t \leq \frac{2\hat{a}_t}{N} \sqrt{\frac{3\delta_0}{\tilde{a}_t(2\hat{a}_t \tilde{a}_t^2 + \hat{a}_t^3) \overline{\Delta}^\star RT}}$. Then, if $T \geq \frac{72\delta_0}{\zeta \varepsilon}$, we have*

$$\mathbb{E} \left[ \min_{t=0,\ldots,T-1} \left\{ \min \left\{ \frac{\|\nabla f(x_t)\|^2}{L_0}, \frac{\|\nabla f(x_t)\|}{L_1} \right\} \right\} \right] \leq \varepsilon.$$

*Proof of Corollary 4.* Since $c = 2\sqrt{T}$ and $\eta_t \leq \frac{2\hat{a}_t}{cR} \sqrt{\frac{1}{a_t(2\hat{a}_t \tilde{a}_t^2 + \hat{a}_t^3)}}$, $\gamma_t \leq \frac{2\hat{a}_t}{cRN} \sqrt{\frac{1}{\tilde{a}_t(2\hat{a}_t \tilde{a}_t^2 + \hat{a}_t^3)}}$, and $\eta_t \leq 2\hat{a}_t \sqrt{\frac{3\delta_0}{2a_t(2\hat{a}_t \tilde{a}_t^2 + \hat{a}_t^3) \Delta^\star RT}}$, $\gamma_t \leq \frac{2\hat{a}_t}{N} \sqrt{\frac{3\delta_0}{\tilde{a}_t(2\hat{a}_t \tilde{a}_t^2 + \hat{a}_t^3) \overline{\Delta}^\star RT}}$ due to the choice of $T \geq \max \left\{ \frac{72\delta_0}{\zeta \varepsilon}, \frac{12\Delta^\star}{\zeta \varepsilon}, \frac{6\overline{\Delta}^\star}{\zeta \varepsilon} \right\}$, we obtain that

$$\frac{\left(1 + \frac{2\hat{a}_t \tilde{a}_t^2 + \hat{a}_t^3}{4\hat{a}_t^2} \left( \eta_t^2 a_t + \eta_t^2 R^2 \hat{a}_t + \gamma_t^2 N \tilde{a}_t + \eta_t^2 R a_t \right) \right)^T}{T} \delta_0 \leq \frac{e\delta_0}{T} \leq \frac{3\delta_0}{T} \leq \frac{\zeta \varepsilon}{24},$$

$$\frac{2\hat{a}_t \tilde{a}_t^2 + \hat{a}_t^3}{4\hat{a}_t^2} \left( \eta_t^2 a_t \Delta^\star + \eta_t^2 R a_t \Delta^\star \right) \leq \frac{\zeta \varepsilon}{24},$$

and that

$$\frac{2\hat{a}_t \tilde{a}_t^2 + \hat{a}_t^3}{4\hat{a}_t^2} \gamma_t^2 N \tilde{a}_t \overline{\Delta}^\star \leq \frac{\zeta \varepsilon}{24}.$$

Therefore, $\mathbb{E} \left[ \min_{t=0,\ldots,T-1} \left\{ \min \left\{ \frac{\|\nabla f(x_t)\|^2}{L_0}, \frac{\|\nabla f(x_t)\|}{L_1} \right\} \right\} \right] \leq \varepsilon.$ $\qquad \square$

### D.2 ASYMMETRIC GENERALIZED-SMOOTH FUNCTIONS UNDER PŁ-CONDITION

**Theorem 10.** *Let Assumptions 1 and 2 hold for functions* $f$, $\{f_m\}_{m=1}^{M}$ *and* $\{f_{mj}\}_{m=1,j=1}^{M,N}$. *Let Assumption 4 hold. Choose* $0 < \zeta \leq \frac{1}{4}$. *Let* $\delta_0 \stackrel{def}{=} f(x_0) - f^{\star}$. *Choose any integer* $T > \frac{64\delta_0 L_1^2}{\mu\zeta}$. *For all* $0 \leq t \leq T - 1$, *denote*

$$\hat{a}_t = L_0 + L_1 \|\nabla f(x_t)\|, \quad a_t = L_0 + L_1 \max_m \|\nabla f_m(x_t)\|, \quad \tilde{a}_t = L_0 + L_1 \max_{m,j} \|\nabla f_m^{\pi_j}(x_t)\|.$$

*Put* $\Delta^{\star} = f^{\star} - \frac{1}{M}\sum_{m=1}^{M} f_m^{\star}$ *and* $\overline{\Delta}^{\star} = f^{\star} - \frac{1}{M}\sum_{m=1}^{M}\frac{1}{N}\sum_{j=0}^{N-1} f_{mj}^{\star}$. *Impose the following conditions on the local stepsizes* $\gamma_t$, *server stepsizes* $\eta_t$, *global stepsizes* $\theta_t$ :

$$\gamma_t N R \leq \eta_t R \leq \min\left\{\frac{1}{16\hat{a}_t}, \frac{2\hat{a}_t}{c}\sqrt{\frac{1}{a_t(2\hat{a}_t\tilde{a}_t^2 + \hat{a}_t^3)}}, \sqrt{\frac{\hat{a}_t^2\mu\zeta}{32L_1^2(\delta_t + \Delta^{\star})a_t(2\hat{a}_t\tilde{a}_t^2 + \hat{a}_t^3)}},\right.$$
$$\left.\sqrt{\frac{\hat{a}_t^2\delta_0}{TL_1^2(\delta_t + \Delta^{\star})a_t(2\hat{a}_t\tilde{a}_t^2 + \hat{a}_t^3)}}\right\},$$

$$\gamma_t N R \leq \min\left\{\frac{2\hat{a}_t}{c}\sqrt{\frac{1}{\tilde{a}_t(2\hat{a}_t\tilde{a}_t^2 + \hat{a}_t^3)}}, \sqrt{\frac{\hat{a}_t^2\delta_0}{TL_1^2\left(\delta_t + \overline{\Delta}^{\star}\right)\tilde{a}_t(2\hat{a}_t\tilde{a}_t^2 + \hat{a}_t^3)}},\right.$$
$$\left.\sqrt{\frac{\hat{a}_t^2\mu\zeta}{32L_1^2\left(\delta_t + \overline{\Delta}^{\star}\right)\tilde{a}_t(2\hat{a}_t\tilde{a}_t^2 + \hat{a}_t^3)}}\right\}.$$

$$\frac{\zeta}{\hat{a}_t} \leq \theta_t \leq \frac{1}{4\hat{a}_t}, \quad 0 \leq t \leq T - 1,$$

*where* $c \geq \sqrt{T}$. *Let* $\tilde{T}$ *be an integer such that* $0 \leq \tilde{T} \leq \frac{64\delta_0 L_1^2}{\mu\zeta}$, $A > 0$ *be a constant,* $\alpha \leq \sqrt{\frac{\delta_0}{AT}}$. *Then, the iterates* $\{x_t\}_{t=0}^{T-1}$ *of Algorithm 3 satisfy*

$$\delta_T \leq \left(1 - \frac{\mu\zeta}{4L_0}\right)^{T-\tilde{T}}\delta_0 + \frac{4L_0 A\alpha^2}{\mu\zeta},$$

*where* $\delta_T \stackrel{def}{=} f(x_T) - f^{\star}$.

*Proof of Theorem 10.* Let us follow the first steps of the proof of Theorem 4. Consider (18):

$$\frac{\theta_t}{4}\|\nabla f(x_t)\|^2 \leq f(x_t) - f(x_{t+1})$$
$$+ \frac{2\hat{a}_t\tilde{a}_t^2 + \hat{a}_t^3}{4\hat{a}_t^2}(f(x_t) - f^{\star})\left(\eta_t^2 a_t + \eta_t^2 R^2\hat{a}_t + \gamma_t^2 N\tilde{a}_t + \eta_t^2 Ra_t\right)$$
$$+ \frac{2\hat{a}_t\tilde{a}_t^2 + \hat{a}_t^3}{4\hat{a}_t^2}\left(\eta_t^2 a_t\Delta^{\star} + \gamma_t^2 N\tilde{a}_t\overline{\Delta}^{\star} + \eta_t^2 Ra_t\Delta^{\star}\right).$$

Since $\theta_t \geq \frac{\zeta}{\hat{a}_t}$, and $f$ satisfies Polyak–Łojasiewicz Assumption 4, we obtain that

$$\frac{\mu\zeta(f(x_t) - f^{\star})}{2\hat{a}_t} \leq f(x_t) - f(x_{t+1})$$
$$+ \frac{2\hat{a}_t\tilde{a}_t^2 + \hat{a}_t^3}{4\hat{a}_t^2}(f(x_t) - f^{\star})\left(\eta_t^2 a_t + \eta_t^2 R^2\hat{a}_t + \gamma_t^2 N\tilde{a}_t + \eta_t^2 Ra_t\right)$$
$$+ \frac{2\hat{a}_t\tilde{a}_t^2 + \hat{a}_t^3}{4\hat{a}_t^2}\left(\eta_t^2 a_t\Delta^{\star} + \gamma_t^2 N\tilde{a}_t\overline{\Delta}^{\star} + \eta_t^2 Ra_t\Delta^{\star}\right).$$

**1.** Let $\tilde{T}$ be the number of steps $t$, so that $\|\nabla f(\hat{x}_t)\| \geq \frac{L_0}{L_1}$. For such $t$, we have $L_0 + L_1 \|\nabla f(x_t)\| = \hat{a}_t \leq 2L_1 \|\nabla f(x_t)\|$. Therefore, we get

$$\frac{\mu\zeta\left(f(x_t) - f^\star\right)}{4L_1 \|\nabla f(x_t)\|} \leq f(x_t) - f(x_{t+1})$$
$$+ \frac{2\hat{a}_t\tilde{a}_t^2 + \hat{a}_t^3}{4\hat{a}_t^2}\left(f(x_t) - f^\star\right)\left(\eta_t^2 a_t + \eta_t^2 R^2 \hat{a}_t + \gamma_t^2 N\tilde{a}_t + \eta_t^2 Ra_t\right)$$
$$+ \frac{2\hat{a}_t\tilde{a}_t^2 + \hat{a}_t^3}{4\hat{a}_t^2}\left(\eta_t^2 a_t\Delta^\star + \gamma_t^2 N\tilde{a}_t\overline{\Delta}^\star + \eta_t^2 Ra_t\Delta^\star\right).$$

Notice that the relation $\hat{a}_t \leq 2L_1 \|\nabla f(x_t)\|$ and Lemma 1 together imply

$$\frac{\|\nabla f(x_t)\|}{4L_1} \leq \frac{\|\nabla f(x_t)\|^2}{2\hat{a}_t} \leq f(x_t) - f^\star.$$

Hence, we have

$$\frac{\mu\zeta}{16L_1^2} \leq f(x_t) - f(x_{t+1})$$
$$+ \frac{2\hat{a}_t\tilde{a}_t^2 + \hat{a}_t^3}{4\hat{a}_t^2}\left(f(x_t) - f^\star\right)\left(\eta_t^2 a_t + \eta_t^2 R^2 \hat{a}_t + \gamma_t^2 N\tilde{a}_t + \eta_t^2 Ra_t\right)$$
$$+ \frac{2\hat{a}_t\tilde{a}_t^2 + \hat{a}_t^3}{4\hat{a}_t^2}\left(\eta_t^2 a_t\Delta^\star + \gamma_t^2 N\tilde{a}_t\overline{\Delta}^\star + \eta_t^2 Ra_t\Delta^\star\right).$$

Subtracting $f^\star$ on both sides and introducing $\delta_t \overset{\text{def}}{=} f(x_t) - f^\star$, we obtain

$$\delta_{t+1} \leq \delta_t - \frac{\mu\zeta}{16L_1^2}$$
$$+ \frac{2\hat{a}_t\tilde{a}_t^2 + \hat{a}_t^3}{4\hat{a}_t^2}\delta_t\left(\eta_t^2 a_t + \eta_t^2 R^2 \hat{a}_t + \gamma_t^2 N\tilde{a}_t + \eta_t^2 Ra_t\right)$$
$$+ \frac{2\hat{a}_t\tilde{a}_t^2 + \hat{a}_t^3}{4\hat{a}_t^2}\left(\eta_t^2 a_t\Delta^\star + \gamma_t^2 N\tilde{a}_t\overline{\Delta}^\star + \eta_t^2 Ra_t\Delta^\star\right).$$

As $\gamma_t \leq \sqrt{\frac{4\hat{a}_t^2\mu\zeta}{128L_1^2\left(\delta_t + \overline{\Delta}^\star\right)\tilde{a}_t R^2 N^2\left(2\hat{a}_t\tilde{a}_t^2 + \hat{a}_t^3\right)}}$ and $\eta_t \leq \sqrt{\frac{4\hat{a}_t^2\mu\zeta}{128L_1^2\left(\delta_t + \Delta^\star\right)a_t R^2\left(2\hat{a}_t\tilde{a}_t^2 + \hat{a}_t^3\right)}}$, it follows that

$$\frac{2\hat{a}_t\tilde{a}_t^2 + \hat{a}_t^3}{4\hat{a}_t^2}\delta_t\left(\eta_t^2 a_t + \eta_t^2 R^2 \hat{a}_t + \gamma_t^2 N\tilde{a}_t + \eta_t^2 Ra_t\right) +$$
$$+ \frac{2\hat{a}_t\tilde{a}_t^2 + \hat{a}_t^3}{4\hat{a}_t^2}\left(\eta_t^2 a_t\Delta^\star + \gamma_t^2 N\tilde{a}_t\overline{\Delta}^\star + \eta_t^2 Ra_t\Delta^\star\right) \leq \frac{\mu\zeta}{32L_1^2}.$$

Therefore, we get

$$\delta_{t+1} \leq \delta_t - \frac{\mu\zeta}{32L_1^2}.$$

**2.** Suppose now that $\|\nabla f(x_t)\| \leq \frac{L_0}{L_1}$. For such $t$, we have $L_0 + L_1 \|\nabla f(x_t)\| = \hat{a}_p \leq 2L_0$. Hence,

$$\frac{\mu\zeta\left(f(x_t) - f^\star\right)}{4L_0} \leq f(x_t) - f(x_{t+1})$$
$$+ \frac{2\hat{a}_t\tilde{a}_t^2 + \hat{a}_t^3}{4\hat{a}_t^2}\left(f(x_t) - f^\star\right)\left(\eta_t^2 a_t + \eta_t^2 R^2 \hat{a}_t + \gamma_t^2 N\tilde{a}_t + \eta_t^2 Ra_t\right)$$
$$+ \frac{2\hat{a}_t\tilde{a}_t^2 + \hat{a}_t^3}{4\hat{a}_t^2}\left(\eta_t^2 a_t\Delta^\star + \gamma_t^2 N\tilde{a}_t\overline{\Delta}^\star + \eta_t^2 Ra_t\Delta^\star\right).$$

Subtracting $f^\star$ on both sides and introducing $\delta_t \overset{\text{def}}{=} f(x_t) - f^\star$, we obtain

$$\delta_{t+1} \leq \delta_t \rho + \frac{2\hat{a}_t \tilde{a}_t^2 + \hat{a}_t^3}{4\hat{a}_t^2} \delta_t \left( \eta_t^2 a_t + \eta_t^2 R^2 \hat{a}_t + \gamma_t^2 N \tilde{a}_t + \eta_t^2 R a_t \right)$$
$$+ \frac{2\hat{a}_t \tilde{a}_t^2 + \hat{a}_t^3}{4\hat{a}_t^2} \left( \eta_t^2 a_t \Delta^\star + \gamma_t^2 N \tilde{a}_t \overline{\Delta}^\star + \eta_t^2 R a_t \Delta^\star \right).$$

where $\rho \overset{\text{def}}{=} 1 - \frac{\mu\zeta}{4L_0}$. Let $\gamma_t \overset{\text{def}}{=} \alpha \hat{\gamma}_t$ and $\eta_t \overset{\text{def}}{=} \alpha \hat{\eta}_t$ with $\hat{\gamma}_t \leq \sqrt{\frac{4\hat{a}_t^2 A}{4L_1^2 (\delta_t + \overline{\Delta}^\star) \tilde{a}_t R^2 N^2 (2\hat{a}_t \tilde{a}_t^2 + \hat{a}_t^3)}}$ and $\hat{\eta}_t \leq \sqrt{\frac{4\hat{a}_t^2 A}{4L_1^2 (\delta_t + \Delta^\star) a_t R^2 (2\hat{a}_t \tilde{a}_t^2 + \hat{a}_t^3)}}$, for some constant $A > 0$. Then,

$$\delta_{t+1} \leq \rho \delta_t + A\alpha^2.$$

Unrolling the recursion, we derive

$$\delta_T \leq \rho^{T-\tilde{T}} \delta_0 + A\alpha^2 \sum_{i=0}^{\infty} \rho^i - \frac{\mu\zeta}{32L_1^2} \sum_{i=0}^{N-1} \rho^i$$
$$\leq \rho^{P-\tilde{P}} \delta_0 + \frac{A\alpha^2}{1-\rho} - \frac{1-\rho^{\tilde{P}}}{1-\rho} \frac{\mu\zeta}{32L_1^2}.$$

Notice that $\delta_{t+1} \leq \delta_t + A\alpha^2$, which implies

$$\delta_T \leq \delta_0 + \left( T - \tilde{T} \right) A\alpha^2 - \tilde{T} \frac{\mu\zeta}{32L_1^2}.$$

Since $\alpha \leq \sqrt{\frac{\delta_0}{AT}}$, we conclude that

$$0 \leq \delta_T \leq 2\delta_0 - \tilde{T} \frac{\mu\zeta}{32L_1^2}, \quad \Rightarrow \tilde{T} \leq \frac{64\delta_0 L_1^2}{\mu\zeta}.$$

Therefore, for $T > \frac{64\delta_0 L_1^2}{\mu\zeta}$ we can guarantee that $T - \tilde{T} > 0$ and

$$\delta_T \leq \rho^{T-\tilde{T}} \delta_0 + \frac{A\alpha^2}{1-\rho} - \tilde{T} \rho^{\tilde{T}} \frac{\mu\zeta}{32L_1^2}$$
$$\leq \rho^{T-\tilde{T}} \delta_0 + \frac{A\alpha^2}{1-\rho}.$$

$\square$

**Corollary 10.** *Fix $\varepsilon > 0$. Choose $\alpha \leq \min\left\{ \sqrt{\frac{\delta_0}{AT}}, L_1 \sqrt{\frac{8\delta_0 \varepsilon}{L_0 AT}} \right\}$. Then, if $T \geq \frac{64\delta_0 L_1^2}{\mu\zeta} + \frac{4L_0}{\mu\zeta} \ln \frac{2\delta_0}{\varepsilon}$, we have $\delta_T \leq \varepsilon$.*

*Proof of Corollary 10.* Since $0 \leq \tilde{T} \leq \frac{64\delta_0 L_1^2}{\mu\zeta}$, $A > 0$, $\alpha \leq \sqrt{\frac{\delta_0}{AT}}$, $\alpha \leq L_1 \sqrt{\frac{8\delta_0 \varepsilon}{L_0 AT}}$, due to the choice of $T \geq \frac{64\delta_0 L_1^2}{\mu\zeta} + \frac{4L_0}{\mu\zeta} \ln \frac{2\delta_0}{\varepsilon}$, we obtain that

$$\left( 1 - \frac{\mu\zeta}{4L_0} \right)^{T-\tilde{T}} \delta_0 \leq e^{-\frac{\mu\zeta}{4L_0}(T-\tilde{T})} \delta_0 \leq \frac{\varepsilon}{2},$$

and that

$$\frac{4L_0 A}{\mu\zeta} \cdot \frac{\delta_0}{AT} \leq \frac{\varepsilon}{2}.$$

Therefore, $\delta_T \leq \varepsilon$. $\square$

# E   EXTENSION TO GLOBAL STEPSIZES WITH PSEUDOGRADIENTS

Let us consider Algorithm 1. For the other two algorithms same results can be obtained in a similar manner. We replace $\gamma_p = \frac{1}{c_0 + c_1 \|\nabla f(\hat{x}_{t_p})\|}$ with $\gamma_p = \frac{1}{c_0' + c_1' \|g_p\|}$ after the computation of $g_p$ in the pseudocode. Recall that

$$\|g_p\| = \left\| \frac{1}{M(v - t_p)} \sum_{m=1}^{M} \sum_{j=t_p+1}^{v} \nabla f_m(x_j^m) \right\|.$$

By the triangle inequality, we obtain that

$$\|g_p\| \leq \frac{1}{(v - t_p)} \left\| \frac{1}{M} \sum_{m=1}^{M} \sum_{j=t_p+1}^{v} \left( \nabla f_m(x_j^m) - \nabla f_m(\hat{x}_{t_p}) \right) \right\| + \left\| \nabla f(\hat{x}_{t_p}) \right\|$$

$$\leq \frac{1}{(v - t_p)M} \sum_{m=1}^{M} \sum_{j=t_p+1}^{v} \left\| \nabla f_m(x_j^m) - \nabla f_m(\hat{x}_{t_p}) \right\| + \left\| \nabla f(\hat{x}_{t_p}) \right\|$$

Since every $f_m$ is $(L_0, L_1)$-smooth, we have that

$$\|g_p\| \leq \frac{a_p}{(v - t_p)M} \sum_{m=1}^{M} \sum_{j=t_p+1}^{v} \left\| \hat{x}_{t_p} - x_j^m \right\| + \left\| \nabla f(\hat{x}_{t_p}) \right\|.$$

By Jensen's inequality we have that

$$\frac{a_p}{(v - t_p)M} \sum_{m=1}^{M} \sum_{j=t_p+1}^{v} \left\| \hat{x}_{t_p} - x_j^m \right\| \leq \frac{a_p}{(v - t_p)M} \sqrt{(v - t_p)M \sum_{m=1}^{M} \sum_{j=t_p+1}^{v} \left\| \hat{x}_{t_p} - x_j^m \right\|^2}$$

$$\overset{\text{Lemma 4}}{\leq} \sqrt{8 (v_p - t_p) a_p^3 \alpha_p^2 \left( f(\hat{x}_{t_p}) - f^\star + \Delta^\star \right)}.$$

For any sufficiently small $\delta > 0$, let us choose $\alpha_p \leq \frac{\delta}{\sqrt{8(v_p - t_p) a_p^3 \left( f(\hat{x}_{t_p}) - f^\star + \Delta^\star \right)}}$. Then, $\|g_p\| \leq \|\nabla f(\hat{x}_{t_p})\| + \delta$.

The lower bound on $\|g_p\|$ is obtained similarly: we just need to write the triangle inequality for the $\|\nabla f(\hat{x}_{t_p})\|$. We have

$$\left\| \nabla f(\hat{x}_{t_p}) \right\| \leq \frac{1}{(v - t_p)M} \sum_{m=1}^{M} \sum_{j=t_p+1}^{v} \left\| \nabla f_m(x_j^m) - \nabla f_m(\hat{x}_{t_p}) \right\| + \|g_p\|$$

$$\leq \delta + \|g_p\|.$$

Finally, we obtain that $\|\nabla f(\hat{x}_{t_p})\| - \delta \leq \|g_p\| \leq \|\nabla f(\hat{x}_{t_p})\| + \delta$. Hence

$$\frac{1}{(c_0' + c_1'\delta) + c_1' \left\| \nabla f(\hat{x}_{t_p}) \right\|} \leq \frac{1}{c_0' + c_1' \|g_p\|} \leq \frac{1}{(c_0' - c_1'\delta) + c_1' \left\| \nabla f(\hat{x}_{t_p}) \right\|}.$$

It means that the practical choice of the stepsize only slightly differs in the constants in the denominator. So, all our theory works for it as well.

# F   ADDITIONAL EXPERIMENTAL DETAILS FOR MAIN PART

In this section, we provide additional experimental details: parameters search grids and some technical details that did not fit in the main text. For all the plots we provide in the legend all the best parameters found by the grid search. The parameter grids are provided as table for every method. The code is available at `https://github.com/postrou/local_steps_rr`.

It can be seen from pseudocode of Algorithms 1, 2, 3, that global stepsize depends on the full gradient. However, our numerical tests showed that use of gradient approximations $g_p$ for Algorithm 1 and $g_t$ for Algorithms 2, 3 gives better numerical results while being less computationally expensive. Thus, in our practical experiments we decided to use this approximation in calculation of global stepsize. We want to point out, that the theoretical analysis for this "practical" version of the algorithm can be done considering very small inner stepsizes. Although, we decided not to include it in the current version to keep the presentation more concise and avoid additional complexities.

### F.1 METHODS WITH RANDOM RESHUFFLING

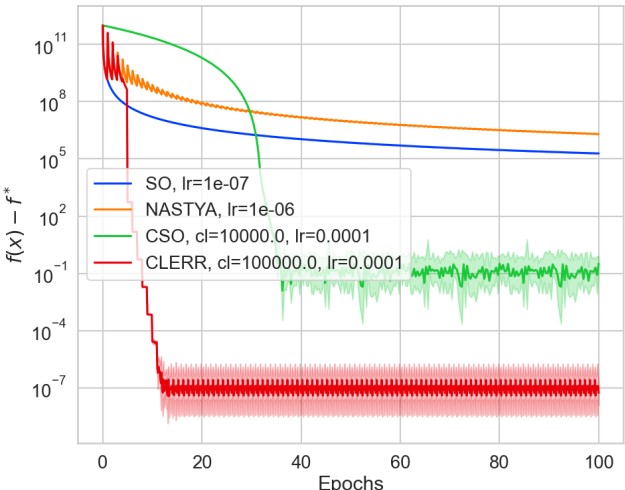

Figure 5: Function residual for (4), $\alpha_t = 10^{-7}$. The best parameters are provided in the legend.

In these experiments we compare methods with random reshuffling, that shuffle data once at the start of training process. The main idea is to show the positive impact of random reshuffling and clipping on algorithm performance. We incorporate these two techniques inside our CLERR method (Algorithm 2).

Firstly, consider (4). For these experiments we take $d = 1$ and randomly sample 1000 shifts $x_i \in [-10, 10]$. We run all the methods for 10 different seeds on a logarithmic hyperparameter grid. Then we choose the best hyperparameters according to the best mean loss values on the second half of epochs. The parameter grid is provided in Table 1. To find $f^*$, we run the Newton method for couple iterations until convergence.

Since both Nastya and Algorithm 2 have jumping at the end of every epoch, if we tuned the inner stepsize along with other parameters, the inner stepsize would go to zero and the outer stepsize would be selected such as these methods solve the problem in 1 step. This would be unfair because other baselines do not use a jumping technique, so they would not be able to achieve such performance. Thus, we decided to fix the inner stepsize for Algorithm 2 and Nastya equal to the best stepsize, chosen for SO, and tune the clipping level and outer stepsize with the outer stepsize not exceeding the values supported in theory. Here and later, for simplicity, we speak about Algorithm 2 in terms of stepsize and clipping level, that we can obtain from $c_0$ and $c_1$ from (3). The best stepsize for SO is $10^{-7}$, so we choose inner stepsize for Nastya and Algorithm 2 the same. Nastya chooses outer stepsize equal $10^{-7}$, while CSO and CLERR (Algorithm 2) choose it equal to $10^{-4}$. CSO clips gradients at the level $10^4$, while CLERR – at the level $10^5$.

| Method | Stepsize | Clipping Level | Inner Stepsize |
|---|---|---|---|
| SO | $[10^{-8}, 10^{-2}]$ | - | - |
| NASTYA | $[10^{-8}, 10^{-2}]$ | - | $10^{-7}$ |
| CSO | $[10^{-8}, 10^{-2}]$ | $[10^0, 10^5]$ | - |
| Algorithm 2 | $[10^{-8}, 10^{-2}]$ | $[10^0, 10^5]$ | $10^{-7}$ |

Table 1: Parameter grids for experiments on methods with random reshuffling on (4).

### F.1.1 RESNET-18 ON CIFAR-10

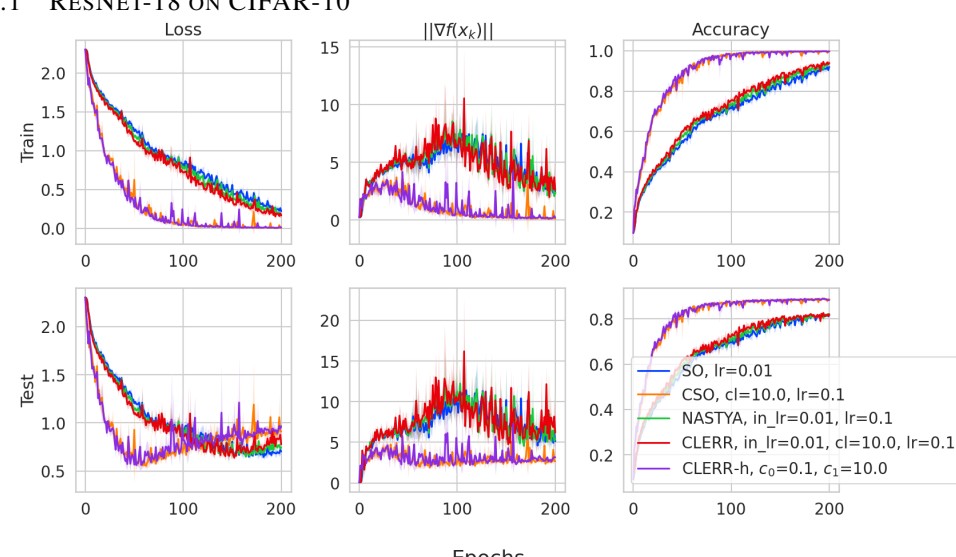

Figure 6: Loss, gradient norm and accuracy on train and test dataset for ResNet-18 on CIFAR-10. The best parameters are provided in the legend.

In this experiment we consider image classification task. We train ResNet-18 He et al. (2016) on the CIFAR-10 Krizhevsky et al. (2009) dataset. The implementation of ResNet-18 was taken from `https://github.com/kuangliu/pytorch-cifar`. All the methods are run on 3 different random seeds on logarithmic hyperparameter grid. Then we choose the best hyperparameters according to the best mean test accuracy on the last 25% of epochs.

In this experiment, we do not fix the inner stepsize for Nastya and CLERR, since methods do not try to make it as small as possible, as it was in the previous experiment. However, both SO, Nastya, and CLERR choose the same inner stepsize $10^{-2}$ as the best. Then, both Nastya and CLERR choose bigger outer step size $10^{-1}$, and CLERR also chooses clipping level on outer step size as 10. Despite the fact that both Nastya and CLERR choose bigger outer stepsizes compared to inner stepsize, jumping does not have any impact on this problem. CLERR clips outer gradients at the level of 10, so this also does not help method to converge to a better area.

Moreover, we provide results of heuristically modified Algorithm 2, where we fix clipping level and inner stepsize of Algorithm 2 equal to the best clipping level and the best stepsize from CSO correspondingly. The tunable parameters are only $c_0$ and $c_1$ for outer stepsize. We call this method CLERR-h. CLERR-h chooses an outer stepsize equal to 5, while the clipping level is very tiny and equal to $10^{-2}$. All the parameter grids are provided in Table 2.

| Method | Stepsize | Clipping Level | Inner Stepsize | $c_0$ | $c_1$ |
|---|---|---|---|---|---|
| RR | $[10^{-3}, 10^{-1}]$ | - | - | - | - |
| NASTYA | $[10^{-3}, 10^{-1}]$ | - | $[10^{-4}, 10^{0}]$ | - | - |
| CRR | $[10^{-3}, 10^{-1}]$ | $[10^{0}, 10^{3}]$ | - | - | - |
| CLERR | $[10^{-3}, 10^{-1}]$ | $[10^{0}, 10^{3}]$ | $[10^{-4}, 10^{0}]$ | - | - |
| CLERR-h | - | $10^{1}$ | $10^{-1}$ | $[10^{-2}, 10^{1}]$ | $[10^{-2}, 10^{1}]$ |

Table 2: Parameter grids for experiments on methods with random reshuffling on ResNet-18 on CIFAR-10.

## F.2 METHODS WITH LOCAL STEPS

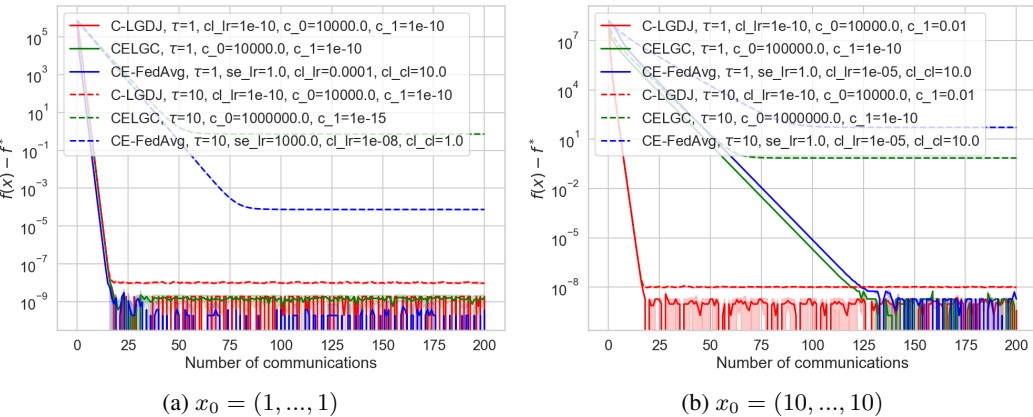

(a) $x_0 = (1, ..., 1)$          (b) $x_0 = (10, ..., 10)$

Figure 7: Function residual for (4), starting from different $x_0$ for different number of local steps on the client device $\tau$. The best parameters are provided in the legend.

In these experiments we compare methods with local steps: Algorithm 1 (C-LGDJ) with Communication Efficient Local Gradient Clipping (CELGC) (Liu et al., 2022) and Clipping-Enabled-FedAvg (CE-FedAvg) Zhang et al. (2022). For comparison we take problem (4) for $d = 100$, where we randomly sample 1000 shifts $x_i \in [-10, 10]^d$. To make the distributions of data on each client more distinct between each other, we sort the whole dataset at the beginning of the experiment by $\|x_i\|$. Each method has 10 clients, where each client has equal number of data. We provide results for two starting points: $x_0 = (1, ..., 1)$ and $x_0 = (10, ..., 10)$. All the methods are run for 10 different random seeds on logarithmic hyperparameter grid. The best hyperparameters are chosen according to the best mean loss on the last 25% of epochs.

Each client performs $\tau = 1$ or $\tau = 10$ local steps, and each local step is performed on the whole local data. For ease of implementation and due to computational limitations we iterate over all the clients sequentially.

We reformulate constants $c_0$ and $c_1$ as server stepsize and clipping level from (3) to better interpret the experimental results. We start by paying attention to results with a single local step. Firstly, consider C-LGDJ (Algorithm 1). It chooses tiny client stepsizes $10^{-10}$ and small server stepsizes $5 \cdot 10^{-5}$ for both starting points. For Figure 7a it also takes very big clipping level for server $10^{14}$, compared to Figure 7b, where it clips on level $10^6$, which is obvious because on the second picture methods start farther from the minimum and have bigger gradients. Secondly, consider CELGC. In both cases, it takes very small client stepsizes: $5 \cdot 10^{-5}$ and $5 \cdot 10^{-6}$ respectively, and very big clipping levels: $10^{14}$ and $10^{15}$ respectively. Finally, CE-FedAvg also takes small client stepsizes: $10^{-4}$ and $10^{-5}$, rather big server stepsizes, which are equal to 1, and average client clipping levels: 10 in both cases. For $\tau = 10$ we have the same parameters for C-LGDJ, CELGC tries to make even smaller steps with high clipping levels, while CE-FedAvg uses a much bigger server stepsize and much smaller client stepsize, for the case from Figure 7a.

The grids of hyperparameters for $x_0 = (1, ..., 1)$ are provided in Table 3, and for $x_0 = (10, ..., 10)$ – in Table 4.

| Method | Cl. Stepsize | Se. Stepsize | Cl. Clip Level | $c_0$ | $c_1$ |
|---|---|---|---|---|---|
| Clipped-L-SGD-J | $[10^{-10}, 10^0]$ | - | - | $[10^{-10}, 10^6]$ | $[10^{-10}, 10^6]$ |
| CELGC | - | - | - | $[10^{-15}, 10^{10}]$ | $[10^{-15}, 10^{10}]$ |
| CE-FedAvg | $[10^{-10}, 10^0]$ | $[10^{-10}, 10^3]$ | $[10^0, 10^4]$ | - | - |

Table 3: Parameter grids for experiments on methods with local steps on (4) for $x_0 = (1, ..., 1)$. Here "cl." means "Client", and "se." – "server".

| Method | Cl. Stepsize | Se. Stepsize | Cl. Clip Level | $c_0$ | $c_1$ |
|---|---|---|---|---|---|
| Clipped-L-SGD-J | $[10^{-10}, 10^0]$ | - | - | $[10^{-10}, 10^6]$ | $[10^{-10}, 10^6]$ |
| CELGC | - | - | - | $[10^{-10}, 10^{10}]$ | $[10^{-10}, 10^{10}]$ |
| CE-FedAvg | $[10^{-10}, 10^0]$ | $[10^{-10}, 10^0]$ | $[10^0, 10^4]$ | - | - |

Table 4: Parameter grids for experiments on methods with local steps on (4) for $x_0 = (10, ..., 10)$. Here "cl." means "client", and "se." – "server".

### F.3 METHODS WITH LOCAL STEPS, RANDOM RESHUFFLING AND PARTIAL PARTICIPATION

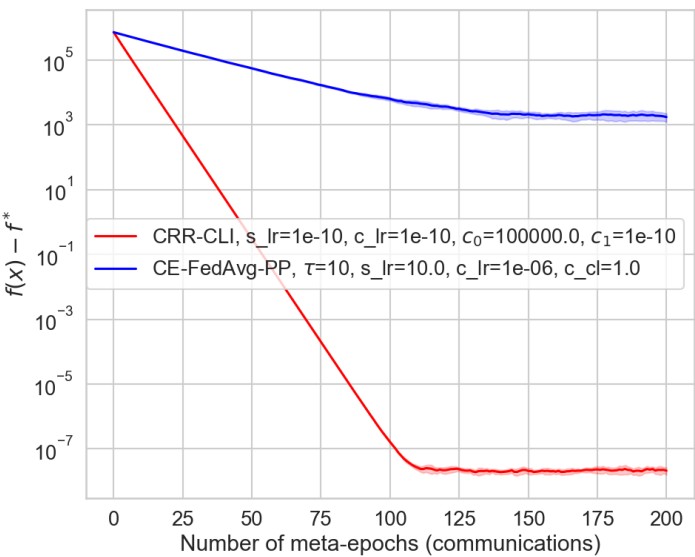

Figure 8: Function residual for (4), starting from $x_0 = (1, ..., 1)$ with batch size 16. The best parameters are provided in the legend.

In these experiments we compare methods with clipping, random reshuffling, local steps and partial participation: Algorithm 3 (CRR-CLI) and with CE-FedAvg Zhang et al. (2022) with partial participation (CE-FedAvg-PP). For comparison we take problem (4) for $d = 100$, where we randomly sample 1000 shifts $x_i \in [-10, 10]^d$. Again, to make the distributions of data on each client more distinct between each other, we sort the whole dataset at the beginning of the experiment by $\|x_i\|$. All the methods are run for 10 different random seeds on logarithmic hyperparameter grid. The best hyperparameters are chosen according to the best mean loss on the last 25% of epochs.

Each method has 10 clients, where each client has the same amount of data. The size of the cohort is chosen to be 2. The method performs local steps on each client from the cohort, after which it performs communication and goes to the next cohort. In the Algorithm 3 the clients to the cohort are chosen sequentially with sliding window after Client-Reshuffling. In CE-FedAvg-PP clients to the cohort are always chosen randomly. The starting point is chosen $x_0 = (1, ..., 1)$. All the methods are run for 10 different random seeds. The best hyperparameters are chosen according to the best mean loss on the last 25% of epochs.

For local steps we chose batch size equal to 16. In Algorithm 3 every client goes sequentially over the whole shuffled local dataset with batch size window. In CE-FedAvg-PP we fix number of local steps to 10, and each client samples batch on every local step.

Just like in previous experiment in Section 5.2, all the methods try to reduce the influence of local steps by making inner stepsizes very small. Algorithm 3 chooses both client and server stepsizes equal $10^{-10}$, and CE-FedAvg-PP chooses client stepsize equal $10^{-6}$ and client clipping level equals 1. Speaking of outer steps, Algorithm 3 chooses global stepsize equal to $5 \cdot 10^{-7}$ with clipping level $10^{16}$. And CE-FedAvg-PP has server stepsize equal to 10. The grids of hyperparameters are provided in Table 5.

| Method | Cl. Stepsize | Se. Stepsize | Cl. Clip Level | $c_0$ | $c_1$ |
|---|---|---|---|---|---|
| CRR-CLI | $[10^{-10}, 10^6]$ | $[10^{-10}, 10^6]$ | - | $[10^{-10}, 10^5]$ | $[10^{-10}, 10^5]$ |
| CE-FedAvg-PP | $[10^{-10}, 10^3]$ | $[10^{-10}, 10^3]$ | $[10^0, 10^4]$ | - | - |

Table 5: Parameter grids for experiments on methods with clipping, random reshuffling, local steps and partial participation. Here "cl." means "client", and "se." – "server".

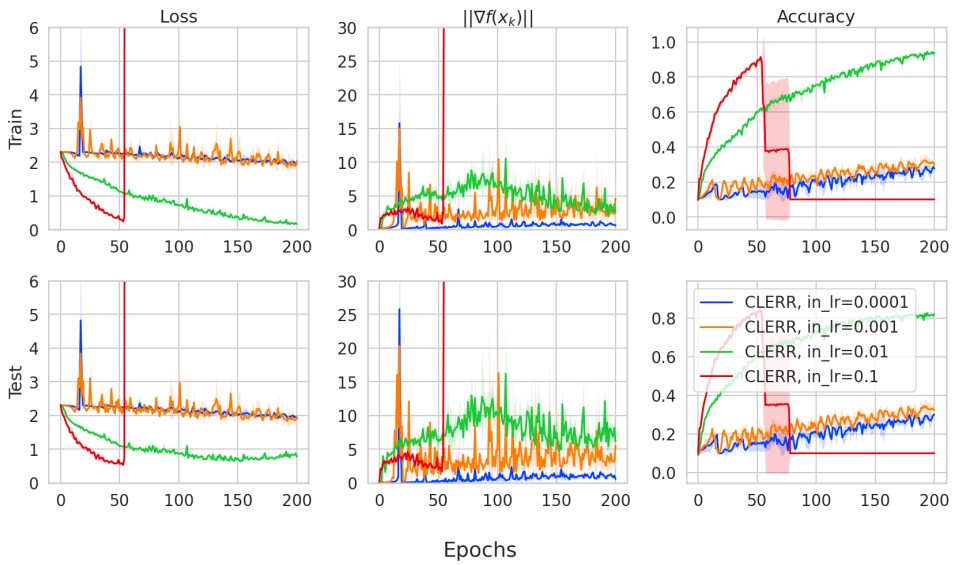

Figure 9: Algorithm 2 with different step sizes on ResNet-18 on CIFAR-10.

## G  ADDITIONAL EXPERIMENTS

In this section, we provide additional numerical experiments, that did not fit in the main paper: in Section G.1 we investigate the influence of that inner step size on the behavior of Algorithm 2, and in Section G.2 we provide additional experiments on logistic regression, where we compare Algorithm 2 with clipped SGD.

### G.1  HOW THE INNER STEP SIZE AFFECTS CONVERGENCE OF THE METHOD

In this experiment, we investigate the influence of the inner step size on the behavior of Algorithm 2 on ResNet-18 on CIFAR-10. To do this, we take the same hyperparameters for Algorithm 2 as in Sections 5.1.1, F.1.1 and only change the inner step size. The results are provided in Figure 9.

On the one hand, if we take the inner step size too small (blue and orange lines), it converges very slowly. This is obvious since Algorithm 2 becomes regular Clipped-GD, which can be seen from pseudocode. Because Clipped-GD performs a single step per epoch, it has slow convergence. On the other hand, if we take the inner step size too big (red line), the method diverges. It does not have clipping on the inner step, so such behavior is expected. To summarize, it is important to take the inner step size small, but not too small, because it may slow down the convergence.

### G.2  LOGISTIC REGRESSION EXPERIMENTS

Since in the experiments on neural networks (Sections 5.1.1, F.1.1) regular CSO (SGD with clipping) showed very good results, we decided to conduct additional experiments on logistic regression, where we compare CSO with our Algorithm 2. We consider gisette and realsim datasets from libsvm library Chang & Lin (2011). All the methods are run for 3 different random seeds on logarithmic hyperparameter grid. The best hyperparameters are chosen according to the best mean loss on the last 25% of epochs. The results are presented in Figure 9.

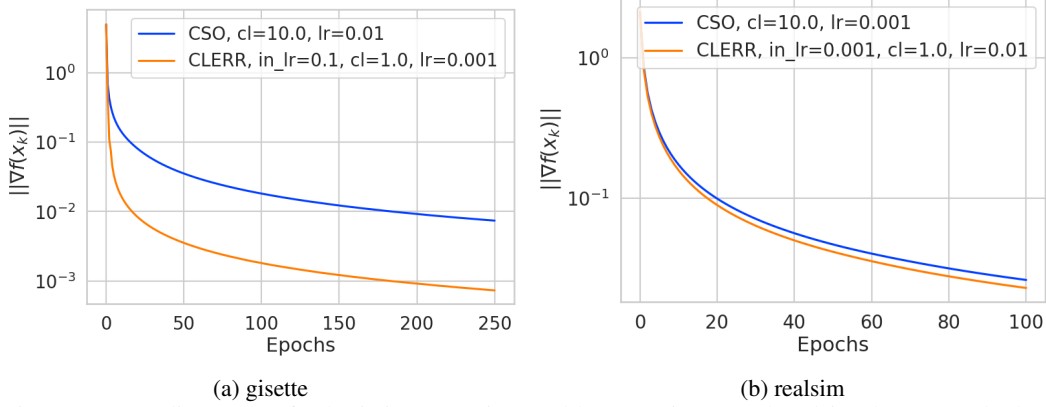

(a) gisette                (b) realsim

Figure 10: Gradient norm for logistic regression problem on gisette and realsim datasets. The best parameters are provided in the legend.

Since the inner stepsize for CLERR has the same meaning as stepsize for CSO, we decided to take the same parameter grids for these two parameters. The same goes for clipping levels in spite of the fact that CLERR clips the gradient approximation only in the end of the epoch. This experiment shows that CLERR either has the same performance as CSO or better. Since logistic regression is $(L_0, L_1)$-smooth, such result is expected, as Algorithm 2 is designed for such type of functions. Figure 10a shows us that CLERR chooses very small outer stepsize $10^{-3}$, while inner step size is bigger than the one in CSO: $10^{-1}$ vs $10^{-2}$. In the Figure 10b CLERR chooses parameters in the opposite way: inner step size is very small and equal to the one from CSO, while the outer stepsize is bigger. The parameter grids for gisette dataset is presented in Table 6, and for realsim – in Table 7.

|  | Stepsize | Clipping Level | Inner Stepsize |
|---|---|---|---|
| CSO | $[10^{-3}, 10^{-1}]$ | $[10^0, 10^2]$ | - |
| CLERR | $[10^{-3}, 10^{-1}]$ | $[10^0, 10^2]$ | $[10^{-3}, 10^{-1}]$ |

Table 6: Parameter grids for logistic regression experiments on gisette dataset

|  | Stepsize | Clipping Level | Inner Stepsize |
|---|---|---|---|
| CSO | $[10^{-5}, 10^{-1}]$ | $[10^0, 10^2]$ | - |
| CLERR | $[10^{-3}, 10^{-1}]$ | $[10^0, 10^2]$ | $[10^{-5}, 10^{-1}]$ |

Table 7: Parameter grids for logistic regression experiments on realsim dataset

## H    EXTENDED RELATED WORK

The usage of distributed methods is dictated by the fact that data can be naturally distributed across multiple devices/clients and be private, which is a typical scenario in Federated Learning (FL) (Konecný et al., 2016; McMahan et al., 2016; Kairouz et al., 2019). FL systems have practical considerations and are backed by extensive experiments from recent years. These highlight important effective design rules and algorithmic features. Below is a quick overview of some key points.

**Partial Participation.** Partial Participation (PP) is a FL technique in which a server selects a subset of clients to engage in the training process during each communication round. Its application may be necessary in scenarios where server capacity or client availability is limited (Kairouz et al., 2021). The technique is useful when the number of clients is large, as the benefits of convergence do not grow proportionally with the size of the cohort (Charles et al., 2021). Clients can be selectively chosen to form a cohort, prioritizing those that deliver the most impactful information (Chen et al., 2020).

**Local training.** Local Training (LT), where clients perform multiple optimization steps on their local data before engaging in the resource-intensive process of parameter synchronization, stands

out as one of the most effective and practical techniques for training FL models. LT was proposed by Mangasarian (1995); Povey et al. (2014); Moritz et al. (2015) and later promoted by McMahan et al. (2016). While these works provided strong empirical evidence for the efficiency and potential of LT-based methods, they lacked theoretical backing. Early theoretical analyses of LT methods relied on restrictive data homogeneity assumptions, which are often unrealistic in real-world federated learning (FL) settings (Stich, 2018; Li et al., 2019; Haddadpour & Mahdavi, 2019). Later, Khaled et al. (2019a;b) removed limiting data homogeneity assumptions for LocalGD (Gradient Descent (GD) with LT). Then, Woodworth et al. (2020); Glasgow et al. (2022) derived lower bounds for GD with LT and data sampling, showing that its communication complexity is no better than minibatch Stochastic Gradient Descent (SGD) in settings with heterogeneous data. Another line of works focused on the mitigating so-called client drift phenomenon, which naturally occurs in LocalGD applied to distributed problems with heterogeneous local functions (Karimireddy et al., 2020; Tran-Dinh et al., 2021; Gorbunov et al., 2021b; Thapa et al., 2022; Mishchenko et al., 2022; Malinovsky et al., 2023b; Yi et al., 2024).

Although removing the dependence on data homogeneity was a key advancement, the theoretical result suggests LT worsens GD, which contradicts empirical evidence showing LT significantly improves it. Karimireddy et al. (2020) identified the client drift phenomenon as the main cause of the gap and proposed a solution to mitigate it, which led to the development of the Scaffold method, featuring the same communication complexity as GD. Later, another algorithm S-Local-GD was proposed by Gorbunov et al. (2021b). Finally, Mishchenko et al. (2022) demonstrated that a novel and simplified form of LT exemplified by their ProxSkip method, results in provable communication acceleration compared to GD. LocalGD is at the base of Federated Averaging (FedAvg) (McMahan et al., 2016). Essentially, FedAvg is a variant of LocalGD with participating devices and data sampled randomly. FedAvg has found applications in various ML tasks, such as, e.g., mobile keyboard prediction (Hard et al., 2018). Wide applicability of FedAvg motivates theoretical study of its backbone LocalGD algorithm.

**Random reshuffling.** Stochastic Gradient Descent (SGD) serves as the foundation for nearly all advanced methods used to train supervised machine learning models. SGD is often refined with techniques like minibatching, momentum, and adaptive stepsizes. However, beyond these enhancements, it is important to decide how to select the next data point for training. Typically, variants of SGD apply a sampling with replacement approach where each new training data point is selected from the full dataset independently of previous samples. Although standard Stochastic Gradient Descent (SGD) (Robbins & Monro, 1951) is well-understood from a theoretical perspective (Rakhlin et al., 2012; Bottou et al., 2018; Nguyen et al., 2018; Gower et al., 2019; Drori & Shamir, 2020; Khaled & Richtárik, 2020; Sokolov, 2022; Demidovich et al., 2024), most widely-used ML frameworks rely on *sampling without replacement*, as it works better in the training neural networks (Bottou, 2009; Recht & Ré, 2013; Bengio, 2012; Sun, 2020). It leverages the finite-sum structure by ensuring each function is used once per epoch. However, this introduces bias: individual steps may not reflect full gradient descent steps on average. Thus, proving convergence requires more advanced techniques. Three popular variants of sampling without replacement are commonly used. *Random Reshuffling (RR)*, where the training data is randomly reshuffled before the start of every epoch, is an extremely popular and well-studied approach. The aim of RR is to disrupt any potentially untoward default data sequencing that could hinder training efficiency. RR works very well in practice. *Shuffle Once (SO)* is analogous to RR, however, the training data is permuted randomly only once prior to the training process. The empirical performance is similar to RR. *Incremental Gradient (IG)* is identical to SO with the difference that the initial permutation is deterministic. This approach is the simplest, however, ineffective. IG has been extensively studied over a long period (Luo, 1991; Grippo, 1994; Li et al., 2022; Ying et al., 2019; Gürbüzbalaban et al., 2019; Nguyen et al., 2021). A major challenge with IG lies in selecting a particular permutation for cycling through the iterations, a task that Nedic & Bertsekas (2001) highlight as being quite difficult. (Bertsekas, 2015) provides an example that underscores the vulnerability of IG to poor orderings, especially when contrasted with RR. Meaningful theoretical analyses of the SO method have only emerged recently (Safran & Shamir, 2020; Rajput et al., 2020). RR has been shown to outperform both SGD and IG for objectives that are twice-smooth (Gürbüzbalaban et al., 2015; Haochen & Sra, 2019). Jain et al. (2019) examine the convergence of RR for smooth objectives. Safran & Shamir (2020); Rajput et al. (2020) provide lower bounds for RR. Mishchenko et al. (2020) recently conducted a thorough analysis of IG, SO and RR using innovative and simplified proof techniques, resulting in better convergence

rates. Recent advances on RR can be found in (Sadiev et al., 2022; Cha et al., 2023; Cai et al., 2023; Koloskova et al., 2023b).

**Other useful features.** Further techniques in FL include compression during the communication rounds (Alistarh et al., 2018; Gorbunov et al., 2021a; Panferov et al., 2024), clients' drift reduction (Karimireddy et al., 2020; Gorbunov et al., 2021b).

