# OpenReview forum: "Methods with Local Steps and Random Reshuffling for Generally Smooth Non-Convex Federated Optimization"
_ICLR.cc/2025/Conference — ICLR 2025 Poster_

### Official Review · Reviewer_R3wR · 2024-10-31

**Soundness:** 2
**Presentation:** 3
**Contribution:** 2
**Rating:** 6
**Confidence:** 5

**Summary:**

This paper introduces three novel federated learning (FL) algorithms—Clip-LocalGDJ, CLERR, and Clipped RR-CLI. The first method adds gradient clipping to localGD on the server side. Building on the first method, the second method adapts reshuffling to local updates. The third method extends the second method to address the client drift problem by shuffling the clients. The convergence of all methods is analyzed under $(L_0, L_1)$ generalized smoothness in non-convex settings without assuming data heterogeneity. Additionally, this paper analyzes the proposed methods under the Polyak-Łojasiewicz condition within the generalized smoothness setting.

**Strengths:**

1. This work adapts gradient clipping to local GD with convergence guarantees.

2. This work fills in several theoretical gaps:

2.1. Convergence analysis for an FL method with partial participation of clients under the $(L_0, L_1)$-smoothness assumption.

2.2. Convergence analysis for FL methods under both $(L_0, L_1)$-smoothness and Polyak-Łojasiewicz conditions.

**Weaknesses:**

1.This work claims that its analysis uses weaker assumptions by not assuming data heterogeneity. However, in the convergence upper bounds presented in Theorems 1, 3, and 4, the right-hand side contains terms such as $\Delta^$ or $\bar{\Delta}^$, which are closely related to data heterogeneity. Specifically, $\Delta^*$ is an example of data heterogeneity as defined in earlier work [1]. This undermines the claim of weaker assumptions since the convergence analysis still implicitly depends on data heterogeneity through these terms. It would be better to make it clear that what is the cost of removing the heterogeneity assumptions.

[1] Xiang Li, Kaixuan Huang, Wenhao Yang, Shusen Wang, Zhihua Zhang:
On the Convergence of FedAvg on Non-IID Data. ICLR 2020

2.  There are many works [2, 3, 4] that provide convergence guarantees without heterogeneity bounds in their complexity. It would be more complete to add these related work.

 [2] Sai Praneeth Karimireddy, Satyen Kale, Mehryar Mohri, Sashank J. Reddi, Sebastian U. Stich, Ananda Theertha Suresh:
SCAFFOLD: Stochastic Controlled Averaging for Federated Learning. ICML 2020: 5132-5143.

 [3] Quoc Tran-Dinh, Nhan H. Pham, Dzung T. Phan, Lam M. Nguyen:
FedDR - Randomized Douglas-Rachford Splitting Algorithms for Nonconvex Federated Composite Optimization. NeurIPS 2021: 30326-30338.

[4] Chandra Thapa, Mahawaga Arachchige Pathum Chamikara, Seyit Camtepe, Lichao Sun:
SplitFed: When Federated Learning Meets Split Learning. AAAI 2022: 8485-8493 2020.

3. After Corollary 1 and Corollary 2, the authors claim that when $L_1 = 0$, their iteration complexity matches that of local GD and GD. However, from my understanding, the motivation for analyzing generalized smoothness is not just to recover known results in the smooth case. In Zhang et al. (2020b), it was empirically shown that gradient clipping can achieve faster convergence. Therefore, to strengthen the theoretical contributions of this work, the authors could add more theoretical discussions on how the iteration complexity of the proposed methods accelerates beyond local GD, rather than simply stating that their complexity matches the previous results in the smooth case. A discussion similar to Remark 5 in Zhang et al. (2020b) would be helpful.

**Questions:**

please refer to the weakness.

---

> ### Author Response · Authors · 2024-11-19
> **Authors' rebuttal**
>
> Dear Reviewer R3wR,
>
> Thank you for the time and effort devoted to reviewing our paper. We appreciate your careful reading, thoughtful evaluation, and constructive feedback. Below, we address the mentioned weaknesses and respond to your questions individually.
>
> >**This work claims that its analysis uses weaker assumptions by not assuming data heterogeneity. However, in the convergence upper bounds presented in Theorems 1, 3, and 4, the right-hand side contains terms such as \Delta^ or \bar{\Delta}^ , which are closely related to data heterogeneity. Specifically, $\Delta^{*}$ is an example of data heterogeneity as defined in earlier work [1]. This undermines the claim of weaker assumptions since the convergence analysis still implicitly depends on data heterogeneity through these terms. It would be better to make it clear that what is the cost of removing the heterogeneity assumptions. [1] Xiang Li, Kaixuan Huang, Wenhao Yang, Shusen Wang, Zhihua Zhang: On the Convergence of FedAvg on Non-IID Data. ICLR 2020**
>
> We kindly disagree with the Reviewer. In fact, **we do not assume any specific bound on the data heterogeneity** – local loss functions can be arbitrarily heterogeneous. The quantity $\Delta^\ast$ is well-defined for many real-world problems. Moreover, in our analysis, $f^\ast$ and $f_i^\ast$ can be replaced with any lower bounds on $f(x)$ and $f_i(x)$. In the context of Machine Learning, where loss functions are typically non-negative, it means that one can take $f^\ast = f_i^\ast = 0$, implying $\Delta^\ast = 0$.
>
> Essentially, our analysis builds upon the paper [I], which analyzes LocalGD on heterogeneous data for the first time (see Line 115). It is the most general and the most difficult setup. Moreover, our result has a further very important advantage. In [I] the result $f(\bar{x}_T) - f^{\star} \le \frac{||x_0-x^{\star}||}{\gamma T} + 24\gamma^2\sigma^2HL$ is obtained for the convex setting. Notice that one can not choose a very small inner stepsize $\gamma$ as it affects the convergence rate (see the first term in the RHS). Our result in Theorem 1 allows us to set the inner stepsizes arbitrarily small and control the $\Delta^{\star}$-term without affecting the rate. The right-hand side there has the form:
> $\frac{\left(1 + \frac{3(H-1)\alpha_p^2a_p^3}{2\hat{a}_p}\right)^P}{P}\delta_0
>          + \frac{3(H-1)\alpha_p^2a_p^3}{2\hat{a}_p}\Delta^\star
> $
>
> The $\Delta^{\star}$-term stems from the client-drift phenomenon (clients are heterogeneous). But this is not a restriction: restrictive condition would mean either a homogeneity of clients or some uniform bound on heterogeneity. We consider the most general case without any such assumptions! Moreover, we deal with heterogeneity in the algorithm itself: in the convergence result above, since $\Delta^{\star}$-term depends on $\alpha_p^2,$ which are inner stepsizes, making $\alpha_p$ small we control this term! The convergence speed is not affected, if we choose them small.
>
> [I] Ahmed Khaled, Konstantin Mishchenko, and Peter Richtárik. First analysis of local gd on heterogeneous data. ArXiv, abs/1909.04715, 2019a.
>
> >**There are many works [2, 3, 4] that provide convergence guarantees without heterogeneity bounds in their complexity. It would be more complete to add these related work. [2] Sai Praneeth Karimireddy, Satyen Kale, Mehryar Mohri, Sashank J. Reddi, Sebastian U. Stich, Ananda Theertha Suresh: SCAFFOLD: Stochastic Controlled Averaging for Federated Learning. ICML 2020: 5132-5143. [3] Quoc Tran-Dinh, Nhan H. Pham, Dzung T. Phan, Lam M. Nguyen: FedDR - Randomized Douglas-Rachford Splitting Algorithms for Nonconvex Federated Composite Optimization. NeurIPS 2021: 30326-30338. [4] Chandra Thapa, Mahawaga Arachchige Pathum Chamikara, Seyit Camtepe, Lichao Sun: SplitFed: When Federated Learning Meets Split Learning. AAAI 2022: 8485-8493 2020.**
>
> We sincerely thank you for sharing these relevant references. Kindly notice that SCAFFOLD is already cited. We cite the other papers and diligently address them in the Related Work section in the revised version. The algorithms in these papers are designed for the standard smoothness assumption. Our methods are tailored for the generalized $(L_0,L1)$-smoothness assumption. As we mention in the main text, our work is the first (to the best of our knowledge) that presents the analysis of FL methods without limiting assumptions. We believe, $(L_0,L_1)$-generalization of the methods in [2,3,4] is a separate interesting open question. In fact, variance reduced methods are known to be impractical as they stuck in local optima [**]. By reducing the inner stepsize $\alpha_p$ we are reducing the $\Delta^{\star}$-term in Theorem 1, so our algorithm has a variance reduction mechanism. But it does not remove variance completely, inner stepsizes are tunable, which makes our algorithm practical.
>
> [**] Aaron Defazio, Léon Bottou, On the ineffectiveness of variance reduced optimization for deep learning, NeurIPS, 2019

---

> ### Author Response · Authors · 2024-11-19
> **Authors' rebuttal**
>
> >**After Corollary 1 and Corollary 2, the authors claim that when L_1=0, their iteration complexity matches that of local GD and GD. However, from my understanding, the motivation for analyzing generalized smoothness is not just to recover known results in the smooth case. In Zhang et al. (2020b), it was empirically shown that gradient clipping can achieve faster convergence. Therefore, to strengthen the theoretical contributions of this work, the authors could add more theoretical discussions on how the iteration complexity of the proposed methods accelerates beyond local GD, rather than simply stating that their complexity matches the previous results in the smooth case. A discussion similar to Remark 5 in Zhang et al. (2020b) would be helpful.**
>
> Thank you for this good suggestion! We added a clarification that we do not just recover the GD result. We recover the Zhang et al. (2020b) result as well.
>
> By setting $H=1$ in Algorithm 1, we essentially obtain clipped Gradient Descent in the (L_0,L_1)-case. Theorem 1 and Corollary 1 in our paper provide convergence guarantees for this algorithm as well. In Theorem 3 from Zhang et al. (2020b) an upper bound is obtained in the form $O\left(\frac{L_0\delta_0}{\varepsilon}\right).$ We guarantee the same rate of $O\left(\frac{L_0\delta_0}{\varepsilon}\right).$
>
> If we resolved all the concerns you have raised, would you be open to **increasing** the score? Should any additional issues remain, we would be more than willing to **discuss** them in detail.

---

> ### Author Response · Authors · 2024-11-25
>
> Dear Reviewer R3wR,
>
> We sincerely appreciate the time you invested in providing your thorough initial review. Your thoughtful comments have been important in refining the revised version of our paper. We have carefully addressed all of your suggestions, and with the discussion period deadline approaching, we would greatly value your feedback. It is essential to further enhancing our work.
>
> We look forward to your feedback and any additional suggestions you may have.

---

> > ### Comment · Reviewer_R3wR · 2024-11-25
> >
> > Thank you for your explanations. I understand that this work does not rely on assumptions regarding heterogeneity. While the $\Delta^*$ term can be controlled by the step size, it is possible that the convergence rate with respect to $P$ is divergent. The denominator grows linearly, while the numerator grows exponentially with respect to $P$. Consequently, after a certain number of steps, this fraction will increase with $P$. The only case the bound converge is when $H=1$, but this mean the clients should communicate every 1 local step. This is inefficient. Therefore, I think the contribution of Theorem 1 is limited. I will keep my rating.

---

> ### Author Response · Authors · 2024-11-26
>
> **Dear Reviewer R3wR,**
>
> Thank you very much for your thoughtful comment! We sincerely appreciate the opportunity to clarify this point further.
>
> The reason both terms are controlled by the inner stepsizes $\alpha_p$ stems from a fundamental inequality in Calculus and Combinatorics: $\left(1 + \frac{1}{n}\right)^n \leq e < 3$ for all $n > 0$. This key result underpins our analysis and helps explain how the terms behave.
>
> Let us now formally address why the first term is effectively controlled by $\alpha_p$:
>
> Consider the term $\frac{\left(1 + \frac{3(H-1)\alpha_p^2a_p^3}{2\hat{a}_p}\right)^P}{P}\delta_0$.
>
> - In the trivial case where $H = 1$, this term decreases **sublinearly $(1/P)$** as it simplifies to $\frac{\delta_0}{P}$.
> - For $H > 1$, let us assume $\alpha_p \leq \sqrt{\frac{2\hat{a}_p}{3(H-1)a_p^3P}}$. This condition ensures that $\alpha_p \lesssim \frac{1}{\sqrt{P}}$, keeping the inner stepsizes sufficiently small.
>
> Under this assumption:
> $$
> \frac{\left(1 + \frac{3(H-1)\alpha_p^2a_p^3}{2\hat{a}_p}\right)^P}{P}\delta_0 \leq \frac{\left(1 + \frac{1}{P}\right)^P}{P}\delta_0 \leq \frac{e}{P}\delta_0 < \frac{3}{P}\delta_0.
> $$
>
> This demonstrates that, with a careful choice of $\alpha_p$, the term decreases **sublinearly $(1/P)$** even when $H > 1$.
>
> For further details, please refer to Appendix B.1, Corollary 1, where we provide the full formal derivations.
>
> To enhance clarity, we will incorporate a verbal explanation of this reasoning into the main text of the paper. We kindly ask you to review our proof and **reconsider your score**, as we rigorously show that our convergence bound is valid and does not diverge under an appropriate choice of the stepsize.
>
> If you have any additional questions or require further clarification, please do not hesitate to reach out.

---

> > ### Comment · Reviewer_R3wR · 2024-11-27
> >
> > Thanks for your clarification. I agree a verbal explanation of this reasoning into the main text of the paper would be more clear. My concerns have been addressed and I would like to adjust the score.

---

> > > ### Author Response · Authors · 2024-11-30
> > > **Authors' response**
> > >
> > > Dear Reviewer R3wR,
> > >
> > > Thank you for your thoughtful discussion, valuable comments, and for increasing the score.

---

### Official Review · Reviewer_mTke · 2024-11-01

**Soundness:** 3
**Presentation:** 4
**Contribution:** 4
**Rating:** 8
**Confidence:** 4

**Summary:**

The paper proposes three federated learning algorithms for solving minimization problems under a generalized smoothness assumption. More specifically, the authors propose a new method, called Clip-LocalGDJ, that uses local steps, a second one that combines local steps with Random Reshuffling, while the third method allows also for partial participation of the clients in the federated learning regime. For all the above methods, convergence guarantees are established under a generalized smoothness assumption, matching the existing results under the classical smoothness assumption. In addition, the authors provide theoretical guarantees for the convergence of each method in the  Polyak-Łojasiewicz (PL) setting, proving a linear rate of convergence. Extensive experimental results are provided, validating the theory and testing the effectiveness of each method in practice.

**Strengths:**

- The analyses of all algorithms are made under the relaxed smoothness assumption and the results are tight in the sense that they coincide with the already established ones in the case when the classical smoothness assumption is true.
- The Clipped RR-CLI algorithm combines local steps, random reshuffling and partial client participation and is analyzed for distributed non-convex and ($L_0, L_1$) smooth problems. This is the first algorithm that combines all the above three properties and is being analyzed under such relaxed conditions.
- The paper is well-written and the presentation is easy to follow for the reader.

**Weaknesses:**

- Since the analysis is provided under the most relaxed assumptions that exist in the literature, do you believe that an analogous proof could be extended in cases where the underlying problem is a minimax optimization problem or in scenarios where extrapolation is also used in the process?
- How does the stepsizes used in Theorems 1, 2 compare with the ones in previous works? Do the best stepsizes found in the Experimental Section coincide with the ones predicted in the theory?
- How might the large-scale nature of data in practical federated learning settings pose challenges for implementing the proposed algorithms? What potential obstacles do you see in adopting these methods widely in practice?

Minor Typos:
- In Section 1.3 (line 186): " clipped IGWang et al. (2024)" the reference should be added correctly.
- In Appendix E: it would be better to incorporate beyond the plots some further details on the additional experiments.

**Questions:**

- Which algorithm of the three should be preferred in practice? In particular, given that Algorithm 3 allows also the partial participation of the clients, it seems to be the best choice in practical scenarios where all clients are not available during the time that the algorithm is run. However, in cases where all clients are available, are there any potential disadvantages of using Algorithm 3 in comparison to Algorithm 2? Additionally, in full participation cases should one prefer in practice Algorithm 2 that utilizes also Random Reshuffling over Algorithm 1?
- Algorithm 3 allows for partial participation by performing a reshuffling of the clients that participate to solve the underlying problem. Is there any practical application, where performing random reshuffling in the clients turns out to be more beneficial than the classical sampling with-replacement technique for partial participation? Are there any theoretical insights or empirical evidence on the trade-off between these two approaches?

**Details Of Ethics Concerns:**

No ethics concerns.

---

> ### Author Response · Authors · 2024-11-19
> **Authors' rebuttal**
>
> Dear Reviewer mTke,
>
> Thanks for your time and positive evaluation of our work! Let us comment on the mentioned weaknesses and respond to questions separately.
>
> >**Since the analysis is provided under the most relaxed assumptions that exist in the literature, do you believe that an analogous proof could be extended in cases where the underlying problem is a minimax optimization problem or in scenarios where extrapolation is also used in the process?**
>
> Thank you for your insightful question! Recently, several notable papers have rigorously analyzed optimization methods for $(L_0, L_1)$-smooth functions, contributing valuable perspectives to the field. In particular:
>
> Daniil Vankov, Angelia Nedich, Lalitha Sankar, *"Generalized Smooth Stochastic Variational Inequalities: Convergence Analysis"*
>
> Daniil Vankov, Angelia Nedich, Lalitha Sankar, *"Generalized Smooth Variational Inequalities: Methods with Adaptive Stepsizes"*, ICML, 2024
>
> One can try to incorporate random reshuffling into these papers.
>
>
> >**How does the stepsizes used in Theorems 1, 2 compare with the ones in previous works? Do the best stepsizes found in the Experimental Section coincide with the ones predicted in the theory?**
>
> Thank you for the question! Previous works include [1, 2]. In the algorithm Nastya [1] the inner stepsize $\gamma$ is upper bounded by $\frac{1}{2NL_0\sqrt{T}}.$ Our algorithms recover this upper bounds up to the constant factor when $L_1=0.$ E.g., in **Appendix.C.1**, Theorem 3, when $L_1=0,$ the inner stepsize $\alpha_t$ is upper bounded by $\frac{1}{L_0\sqrt{((N − 1)(2N − 1) + 2(N + 1))}T}\approx \frac{1}{NL_0\sqrt{2}}.$ In **Appendix.B.1**, Theorem 1 yields $\alpha_p\le \frac{1}{L_0\sqrt{P}}.$ The server stepsizes in Nastya are bounded by $\frac{1}{L_0}.$ Our Theorems guarantee an upper bound of $\frac{1}{4L_0}$ when $L_1=0.$
> In the algorithm RR-CLI [2] the stepsizes are connected by the relation $\gamma N R \le \eta R \le \theta \le \frac{1}{16L_0}$ which is recovered by Clipped RR-CLI in **Appendix.D.1**, Theorem 4, when $L_1=0.$
>
> Our theory provides upper bounds on all stepsizes. In general, the inner stepsizes should be small to control the neighborhood $\Delta^{\star}$-term. So the upper bound on the inner stepsizes is a worst-case guarantee. Practitioners usually tune stepsizes. Regarding the connection between theoretical and experimental stepsizes in **Section 5** and **Appendix E**, our theoretical results provide worst-case convergence guarantees. As a result, the theoretical estimation of the stepsize can be overly pessimistic. For this reason, we treat stepsize parameters as hyperparameters and tune them using grid search.
>
> [1] Grigory Malinovsky, Konstantin Mishchenko, and Peter Richtárik. *"Server-side stepsizes and sampling without replacement provably help in federated optimization"*.
>
> [2] Grigory Malinovsky, Samuel Horváth, Konstantin Burlachenko, and Peter Richtárik. *"Federated learning with regularized client participation"*.
>
> >**How might the large-scale nature of data in practical federated learning settings pose challenges for implementing the proposed algorithms? What potential obstacles do you see in adopting these methods widely in practice?**
>
> Thank you for the thoughtful question. Let us highlight several challenges that might arise in practical implementations of the proposed method:
>
> 1. **Incomplete Local Epochs:** A significant issue is that some clients might not complete their local epochs before synchronization. Similar challenges have been studied under standard smoothness assumptions, as discussed in:
>    - Cho, Yae Jee, et al. *"On the Convergence of Federated Averaging with Cyclic Client Participation."* International Conference on Machine Learning. PMLR, 2023.
>    - Yang, Haibo, et al. *"Understanding Server-Assisted Federated Learning in the Presence of Incomplete Client Participation."* arXiv preprint arXiv:2405.02745 (2024).
>    Analyzing such scenarios under the general smoothness assumption remains an open problem, and we plan to address it in future work.
>
> 2. **Partial Participation Challenges:** Implementing a cyclic pattern of client participation, as assumed in some theoretical models, is not straightforward in real-world applications. Factors such as varying client availability and system constraints make this organization difficult to achieve.
>
> 3. **Asynchronous Aggregation:** Many real-world federated learning systems rely on asynchronous aggregation mechanisms to handle client heterogeneity and communication delays. While this approach is practical, incorporating it into our framework requires further analysis, which we leave as a direction for future work.
>
> >**In Section 1.3 (line 186): " clipped IGWang et al. (2024)" the reference should be added correctly.**
>
> Thank you for your comment! We believe that there is no mistake, but we will double-check the references to make sure everything is correct.

---

> ### Author Response · Authors · 2024-11-19
> **Authors' rebuttal**
>
> >**In Appendix E: it would be better to incorporate beyond the plots some further details on the additional experiments.**
>
> We thank the Reviewer for a valuable remark. We complement **Appendix E** with clarifications regarding the experiments in the Rebuttal Revision of the paper. We provide all the technical details, along with parameter grids and plots with the best-chosen parameters. Moreover, we add some additional experiments on Logistic Regression couple of datasets in **Appendix F**. The experiments on two more datasets are still in progress.
>
> >**Which algorithm of the three should be preferred in practice? In particular, given that Algorithm 3 allows also the partial participation of the clients, it seems to be the best choice in practical scenarios where all clients are not available during the time that the algorithm is run. However, in cases where all clients are available, are there any potential disadvantages of using Algorithm 3 in comparison to Algorithm 2? Additionally, in full participation cases should one prefer in practice Algorithm 2 that utilizes also Random Reshuffling over Algorithm 1?**
>
> Thank you for the question! When the number of workers is large, partial participation is preferable. In this case, Algorithm 3 is the best option as it utilizes partial participation. Otherwise, if we have access to full gradients on the workers, then Algorithm 1 is preferable. In case when the workers can compute only a stochastic gradient, then Algorithm 2 is recommended. We added a clarification in **Section 2**!
>
> >**Algorithm 3 allows for partial participation by performing a reshuffling of the clients that participate to solve the underlying problem. Is there any practical application, where performing random reshuffling in the clients turns out to be more beneficial than the classical sampling with-replacement technique for partial participation? Are there any theoretical insights or empirical evidence on the trade-off between these two approaches?**
>
> In practice, it is common to use a cycling pattern for client participation, which can involve either random reshuffling or deterministic shuffling of clients. This approach is often employed in scenarios where clients are split into groups based on specific factors, such as time zones, hierarchical structures, or other organizational constraints. For instance, in time zone-based grouping, clients in similar time zones may be synchronized to optimize participation, while in hierarchical structures, participation might follow a defined order to align with organizational priorities.
>
> Random reshuffling of clients can also be interpreted from the perspective of sequential federated learning, where clients participate in a structured sequence rather than randomly. This method ensures that all clients are eventually included in the learning process, addressing fairness and data diversity concerns.
>
> Li, Yipeng, and Xinchen Lyu. "Convergence analysis of sequential federated learning on heterogeneous data." Advances in Neural Information Processing Systems 36 (2024).
>
> These applications hold significant importance in the federated learning community, as they provide practical strategies to manage client participation effectively in real-world settings, balancing efficiency, fairness, and performance. Incorporating such patterns into federated learning algorithms can lead to more robust and scalable solutions.

---

> > ### Comment · Reviewer_mTke · 2024-11-19
> >
> > Thank you very much for your detailed response!
> > I will keep my score.

---

> > > ### Author Response · Authors · 2024-11-25
> > >
> > > Dear Reviewer mTke,
> > >
> > > Thank you for your thoughtful questions and interest in our work. If you have any remaining concerns, we would welcome the opportunity to address them in further detail.
> > >
> > > Best regards, Authors

---

> > > > ### Comment · Reviewer_mTke · 2024-12-02
> > > >
> > > > Thank you very much for the fruitful discussion. All of my questions and concerns have been addressed thoroughly.

---

### Official Review · Reviewer_pNnc · 2024-11-03

**Soundness:** 3
**Presentation:** 2
**Contribution:** 3
**Rating:** 6
**Confidence:** 3

**Summary:**

This paper studies and analyzes the federated learning algorithms under the nonconvex setting with the generalized smoothness condition. In the deterministic setting where each client can compute its full gradient, the authors propose a new method with local steps with jumping. In the stochastic (finite-sum) setting, the authors propose two algorithms with partial participation and shuffling on both clients and data sides. All the algorithms proposed are new, and rigorous convergence analysis under generalized smoothness is provided in this paper.

**Strengths:**

- The authors provide a comprehensive analysis for federated learning under generalized smoothness conditions, which I believe is novel and new.
- It is interesting to see random reshuffling is analyzed under generalized smoothness for federated learning problems.
- The spectrum of the federated learning algorithms with their convergence results in this paper is quite substantial.

**Weaknesses:**

- In the section of related work, the authors missed recent advances since 2020 in random reshuffling (e.g., [1-3]), which I believe should be included for a complete context.
- The global step size $\theta_t$ requires evaluating the full gradient, i.e., $\|\nabla f(x_t)\|$, which requires further computation and communication costs.
- Algorithm 2 returns $x_T$ in the statement, but the convergence is only analyzed for the best iterate. The statement of Algorithm 2 possibly needs rephrased.
- The experiments are mostly conducted on a synthetic function (4).

[1] Cha, Jaeyoung, Jaewook Lee, and Chulhee Yun. "Tighter lower bounds for shuffling SGD: Random permutations and beyond." International Conference on Machine Learning. PMLR, 2023.
[2] Cai, Xufeng, Cheuk Yin Lin, and Jelena Diakonikolas. "Empirical risk minimization with shuffled SGD: a primal-dual perspective and improved bounds." arXiv preprint arXiv:2306.12498 (2023).
[3] Koloskova, Anastasia, et al. "On Convergence of Incremental Gradient for Non-Convex Smooth Functions." arXiv preprint arXiv:2305.19259 (2023).

**Questions:**

Besides the weaknesses mentioned, I have the following questions.
- Is it possible to analyze the local steps for Algorithms 2 and 3? Would the local steps bring benefits with random reshuffling, or are there any technical difficulties that the authors foresee?
- Is it possible to analyze the case of different inner stepsizes for different clients?
- In general how does one choose the constants $c_0, c_1$ in the step sizes?
- Is Appendix E on additional experiment details unfinished? There are only a few figures with captions that extend beyond the box.

---

> ### Author Response · Authors · 2024-11-19
> **Authors' rebuttal**
>
> >Strengths:
>
> >The authors provide a comprehensive analysis for federated learning under generalized smoothness conditions, which I believe is novel and new.
>
> >It is interesting to see random reshuffling is analyzed under generalized smoothness for federated learning problems.
>
> >The spectrum of the federated learning algorithms with their convergence results in this paper is quite substantial.
>
> Thank you for your time and the positive feedback!
>
> >Weaknesses:
>
> >In the section of related work, the authors missed recent advances since 2020 in random reshuffling (e.g., [1-3]), which I believe should be included for a complete context.
>
> Thank you for providing the relevant references. We greatly appreciate your input and carefully address these in the related work section.
>
> >The global step size requires evaluating the full gradient, which requires further computation and communication costs.
>
> Thank you for the insightful comment! We would like to clarify that the global stepsize in our method requires the norm of the full gradient, but not the gradient itself. In practice, instead of using the exact norm of the full gradient, we actually already employ the norm of the gradient estimator $g_t$, as demonstrated in our numerical experiments. We have added clarifications regarding this point to the experimental section and Appendix E to improve the paper's clarity.
> Furthermore, while it is indeed **possible** to provide a theoretical analysis for this practical version of the algorithm, we chose not to include it in the current version to maintain a concise presentation and avoid additional complexities. However, we agree that incorporating the practical version using the norm of the gradient estimator $g_t$ would be valuable, and we plan to include it, along with the corresponding analysis, in the camera-ready version. In fact, the analysis with $g_t$ instead of $\nabla f(x_t)$ can be carried out using the **current lemmas** if the inner stepsize is sufficiently small. Specifically, we will use an analogue of Lemma 4 to account for the possibility of using $g_t$ in our analysis. We sincerely appreciate your suggestion and will ensure it is addressed thoroughly.
>
> >Algorithm 2 returns in the statement, but the convergence is only analyzed for the best iterate. The statement of Algorithm 2 possibly needs rephrased.
>
> Thank you for your feedback. We will address the **typo** in the output of Algorithm 2 to ensure it is consistent and coherent with the outputs presented in Algorithm 1 and Algorithm 3. This correction will be reflected in the revised version of the paper.
>
> >The experiments are mostly conducted on a synthetic function (4).
>
> We respectfully disagree with this statement. In the paper, we provided detailed results for the **ResNet-18** model on the **CIFAR-10** dataset. Please refer to Section 5.1.1, where we clearly describe our experimental results, setup, and other important details. This section explains our methods, configurations, and metrics to ensure clarity and transparency. If there are any parts that seem unclear or need more explanation, we would be happy to address them.

---

> ### Author Response · Authors · 2024-11-19
> **Authors' rebuttal**
>
> >Questions:
>
> >Besides the weaknesses mentioned, I have the following questions.
>
> >Is it possible to analyze the local steps for Algorithms 2 and 3? Would the local steps bring benefits with random reshuffling, or are there any technical difficulties that the authors foresee?
>
> It seems there may be some **confusion** on this point, and we would like to clarify it. Please note that all the algorithms in our paper **utilize local steps**. Specifically, in Algorithm 1, we use Gradient Descent updates for the local steps. In Algorithms 2 and 3, we employ Random Reshuffling (SGD without replacement) as the local solver for these steps.
>
> Our analysis highlights the benefits of using the Random Reshuffling method for local steps, as it ensures that each data point in the dataset is used exactly once per epoch. Therefore, we have already provided the **results** that align with your request. If there are any additional clarifications needed, we would be happy to address them. Thank you for your attention to this matter.
>
> >Is it possible to analyze the case of different inner stepsizes for different clients?
>
> Thank you for your suggestion! While the approach of using different stepsizes for different clients could offer some flexibility, it would significantly increase the complexity of hyperparameter tuning, making the parameter search process more challenging in practice.
> That said, we acknowledge that our **current** analysis supports the use of **different** stepsizes across clients, provided these stepsizes respect a common upper bound for convergence guarantees. However, exploring and analyzing adaptive versions of the algorithm, which might automatically adjust stepsizes for each client based on specific criteria, is an interesting direction that we leave for future work.
>
> >In general how does one choose the constants $c_0, c_1$ in the step sizes?
>
> We provide **detailed** specifications of the constants $c_0$ and $c_1$​ in Appendices B, C, and D, where the complete statements of Theorems 1, 3, and 4 are presented. Specifically, $c_0$ is defined as $const * L_0$​, and $c_1$ is defined as $const * L_1,$ where $const \ge 4.$ We will include a clarifying remark regarding these constants in the main part of the camera-ready version to enhance clarity. Thank you for pointing this out!
> It is also worth noting that $c_0$​ and $c_1$​ can be treated as **tunable** parameters in practical implementations. Adjusting these constants can lead to improved convergence performance in numerical experiments, offering an additional degree of flexibility for practitioners.
>
> >Is Appendix E on additional experiment details unfinished? There are only a few figures with captions that extend beyond the box.
>
> Thank you for your response. We have supplemented Appendix E with clarifications regarding the experiments. Specifically, we provide all the technical **details**, including parameter grids and plots with the best-chosen parameters. Moreover, we have added additional **experiments** on Logistic Regression with a couple of datasets in Appendix F. Experiments on two more datasets are still in progress.
>
> If we **address** all the issues you highlighted, would you kindly consider **increasing** the score? Should you have any remaining concerns, we would be more than happy to discuss them further.

---

> ### Author Response · Authors · 2024-11-25
> **Authors' rebuttal**
>
> Dear Reviewer pNnc,
>
> We are very grateful for the time you dedicated to providing your thorough initial review. Your thoughtful comments have been important in helping us prepare a revised version of the paper. We have addressed each point in detail, and with the discussion period deadline approaching, we would highly appreciate your valuable feedback. It is essential in improving our work further.
>
> We look forward to your feedback and any further suggestions you may offer!

---

> ### Comment · Reviewer_pNnc · 2024-11-25
>
> I thank the reviewers' detailed response. For W2, it will be beneficial to include the analysis with the gradient estimator $\\|g_t\\|$. But I know it is hard to make such a change in this short discussion period, so it is ok for the authors to commit their promised changes later. My other questions have been addressed, so I slightly increased my score.

---

> > ### Author Response · Authors · 2024-11-26
> >
> > Thank you for the interest in our work and useful suggestions for its improvement! In fact, it is not too difficult to explain the idea. We provide our derivations below.
> >
> > Let us consider **Algorithm 1**. For the version of it with a practical stepsize, we remove the **Line 4** with $\gamma_p = \frac{1}{c_0+c_1||\nabla f (\hat{x}_{t_p})||},$ and add on the **Line 13** after the computation of $g_p$ in **Line 12** the other choice of the server stepsize $\gamma_p = \frac{1}{c_0+c_1||g_p||}.$ Next, we have
> >
> > $$\lVert g_p \rVert = \lVert \frac{1}{M(v - t_p)}\sum_{m=1}^{M}\sum_{j = t_p+1}^{v}\nabla f_m(x_j^m) \rVert.$$
> > By the triangle inequality, it is no greater than the sum of
> > $\frac{1}{(v - t_p)}\lVert\frac{1}{M}\sum_{m=1}^{M}\sum_{j = t_p+1}^{v}\left(\nabla f_m(x_j^m) - \nabla f_m(\hat{x}_{t_p})\right)\rVert$
> >
> > and $\lVert \nabla f(\hat{x}_{t_p})\rVert.$
> >
> > Further, $\frac{1}{(v - t_p)}\lVert\frac{1}{M}\sum_{m=1}^{M}\sum_{j = t_p+1}^{v}\left(\nabla f_m(x_j^m) - \nabla f_m(\hat{x}_{t_p})\right)\rVert$ is no greater than
> >
> > $A:=\frac{1}{(v - t_p)M}\sum_{m=1}^{M}\sum_{j = t_p+1}^{v}\lVert\nabla f_m(x_j^m) - \nabla f_m(\hat{x}_{t_p})\rVert.$ Since $f_m$ is $(L_0,L_1)$-smooth, we have that
> >
> > $A\le \frac{a_p}{(v - t_p)M}\sum_{m=1}^{M}\sum_{j = t_p+1}^{v}||\hat{x}_{t_p} - x_j^m||.$
> >
> > By **Jensen's inequality** we have that $\frac{1}{(v - t_p)M}\sum_{m=1}^{M}\sum_{j = t_p+1}^{v}||\hat{x}_{t_p} - x_j^m||$ is no greater than
> >
> > $\frac{1}{(v - t_p)M}\sqrt{(v-t_p)M\sum_{m=1}^{M}\sum_{j = t_p+1}^{v}||\hat{x}_{t_p} - x_j^m||^2}$
> >
> > The result of **Lemma 4** implies that it can be bounded by $\sqrt{8\left(v_p -t_p\right)a_p\alpha_p^2\left(f(\hat{x}_{t_p}) - f^{\star} + \Delta^\star\right)}.$
> >
> > Set $B_p:=a_p\sqrt{8\left(v_p -t_p\right)a_p\alpha_p^2\left(f(\hat{x}_{t_p}) - f^{\star} + \Delta^\star\right)}.$
> >
> > Therefore, $\lVert g_p \rVert \le \lVert \nabla f(\hat{x}_{t_p})\rVert + B_p.$
> >
> > The lower bound on the $\lVert g_p \rVert$ is obtained similarly: we just need to write the triangle inequality for the $\lVert \nabla f(\hat{x}_{t_p})\rVert.$ We have
> >
> > $\lVert \nabla f(\hat{x}_{t_p})\rVert \le A + ||g_p||.$
> >
> > We have an upper bound on $A$ already. Hence
> > $\lVert \nabla f(\hat{x}_{t_p})\rVert \le$
> >
> > $ a_p \sqrt{8\left(v_p -t_p\right)a_p\alpha_p^2\left(f(\hat{x}_{t_p}) - f^{\star} + \Delta^\star\right)} + ||g_p|| = B_p + ||g_p||$
> >
> > Finally, we obtain that $\lVert \nabla f(\hat{x}_{t_p})\rVert - B_p \le\lVert g_p \rVert$
> >
> > $\le\lVert \nabla f(\hat{x}_{t_p})\rVert + B_p.$
> >
> > Notice that $B_p$ is controlled by the inner stepsizes $\alpha_p.$ If we choose every $\alpha_p$ small enough, we can bound every $B_p$ by a very small constant $\delta.$
> >
> > Then the stepsizes will be only slightly different. In the beginning we had $\gamma_p = \frac{1}{c_0+c_1||\nabla f (\hat{x}_{t_p})||}.$
> >
> > With the more practical choice we would have
> >
> > $\gamma_p = \frac{1}{c_0+c_1||g_p||} \le \frac{1}{c_0 + c_1\cdot(||\nabla f (\hat{x}_{t_p})|| - \delta)}$
> >
> > $= \frac{1}{(c_0 - c_1\delta) + c_1||\nabla f (\hat{x}_{t_p})||}$
> >
> > $= \frac{1}{c_0^{new1} + c_1||\nabla f (\hat{x}_{t_p})||}.$
> >
> > From below:
> >
> > $\gamma_p = \frac{1}{c_0+c_1||g_p||} \ge \frac{1}{c_0 + c_1\cdot(||\nabla f (\hat{x}_{t_p})|| + \delta)}$
> >
> > $= \frac{1}{(c_0 + c_1\delta) + c_1||\nabla f (\hat{x}_{t_p})||}$
> >
> > $= \frac{1}{c_0^{new2} + c_1||\nabla f (\hat{x}_{t_p})||}.$
> >
> > It means that **the practical choice of the stepsize only differs in the constants in the denominator**. So, all our theory works for it as well.

---

### Official Review · Reviewer_tEec · 2024-11-04

**Soundness:** 3
**Presentation:** 3
**Contribution:** 3
**Rating:** 8
**Confidence:** 3

**Summary:**

This paper proposed and analyzed new methods with local steps, partial participation of clients, and Random Reshuffling for Federated Learning with generalized $(L_0, L_1)$ smoothness assumption. Moreover, they also analyzed these methods with the additional Polyak-Łojasiewicz condition. Their results are consistent with the known results for standard smoothness, and they also conducted experiments to support their theoretical claims.

**Strengths:**

The authors proposed new methods with local steps, Random Reshuffling, and partial participation for Federated Learning and proved the convergence of their algorithms under $(L0, L1)$-smoothness assumption and under additional PL assumption.

**Weaknesses:**

(1): It would be better if the authors can add a conclusion/discussion section at the end of the main text.

(2): The section E Additional experimental details are missing.

**Questions:**

See Weakness.

---

> ### Author Response · Authors · 2024-11-19
> **Authors' rebuttal**
>
> >Strengths:
> The authors proposed new methods with local steps, Random Reshuffling, and partial participation for Federated Learning and proved the convergence of their algorithms under $(L_0, L_1)$-smoothness assumption and under additional PL assumption.
>
> Thank you for your time and the positive feedback!
>
> >Weaknesses:
>
> >(1): It would be better if the authors can add a conclusion/discussion section at the end of the main text.
>
> Thank you for your suggestion! We reorganized the paper a little bit. We squeezed several formulas to save space and moved some parts of the related work section to the extended related work section in the appendix. We also moved part of the Experiments section to Appendix E. It allowed us to **add** the conclusion paragraph which now has the following form:
>
> **Conclusion**
>
> In this paper, we consider a more general smoothness assumption and propose three new distributed methods for Federated Learning with local steps under this setting. Specifically, we analyze local gradient descent (GD) steps, local steps with Random Reshuffling, and a method that combines local steps with Random Reshuffling and Partial Participation. We provide a tight analysis for general non-convex and Polyak-Łojasiewicz settings, recovering previous results as special cases. Furthermore, we present numerical results to support our theoretical findings.
>
> For future work, it would be valuable to explore local methods with communication compression under the generalized smoothness assumption, as well as methods incorporating incomplete local epochs. Additionally, investigating local methods with client drift reduction mechanisms to address the effects of heterogeneity, along with potentially parameter-free approaches, represents a promising direction.
>
> >(2): The section E Additional experimental details are missing.
>
> Thank you for your valuable feedback. In the revised version, we have included all the technical details about synthetic data generation, parameter grids, and the parameters that were selected, as outlined in **Appendix E**. Additionally, we have incorporated **new** experiments on Logistic Regression, which can be found in **Appendix F**. Experiments on two additional datasets are currently in progress.
>
>
> If we address all the concerns you mentioned, would it be possible for you to consider **increasing the score**? If there are any remaining issues, we would be more than happy to **discuss** them further.

---

> ### Author Response · Authors · 2024-11-25
> **Authors' rebuttal**
>
> Dear Reviewer tEec,
>
> Thank you once again for taking the time to provide your initial review.
>
> We have carefully addressed your comments in a detailed response, added respective changes into the revised version of the paper, and with the discussion period deadline fast approaching, we would deeply value your feedback. Your insights will be essential in refining our work further.
>
> We look forward to your thoughts and any additional suggestions you may have!

---

> > ### Comment · Reviewer_tEec · 2024-11-26
> >
> > Thank you very much for your detailed response and the revised paper.
> >
> > A small concern for me is that in the author's rebuttal to reviewer R3wR, it is said:"Our result in Theorem 1 allows us to set the inner stepsizes arbitrarily small and control the $\Delta^{\star}$-term without affecting the rate." It may be true that, in theory, setting the local stepsizes arbitrarily small would not affect the converge rate. However, in reality, if we set the local stepsizes extremely small, then the local update will be meaningless. Intuitively, the different choices of local stepsizes would affect the performance of the algorithm. However, such potential affection of different choices of local stepsizes is not shown in Theorem 1. I think it will be beneficial to include the analysis of the local stepsizes from theory or from experiments.
> >
> > I will keep my score.

---

> ### Author Response · Authors · 2024-11-27
>
> We thank the reviewer for this remark. Indeed, we did not provide enough details on what we mean by “arbitrary small inner stepsizes”. We add numerical evidence for the change of behavior of the CLERR method (Algorithm 2) for ResNet-18 on CIFAR10 in Section F.1 in the Appendix. The same works for Algorithm 1, since they both assume arbitrary small inner step size in the convergence theorems. As you can see from Figure 9, taking too big inner step sizes at first leads to faster convergence, but eventually, the method diverges. On the other hand, the small inner step size slows the convergence because, in this case, the method becomes similar to Clip-GD, which performs only one step per epoch.

---

> > ### Comment · Reviewer_tEec · 2024-11-27
> >
> > Thank you for your detailed response. I increased my score.

---

> > > ### Author Response · Authors · 2024-11-30
> > > **Authors' response**
> > >
> > > Dear Reviewer tEec,
> > >
> > > Thank you for your thoughtful discussion, valuable comments, and for increasing the score.

---

### Meta-Review · Area_Chair_D8LE · 2024-12-24

**Metareview:**

This is the first paper which studies federated learning style optimization problem for generalized smooth objectives. They propose and study a spectrum of federated learning algorithms for this setting and achieve rates which match that of Lipschitz smooth functions. Paper is technically correct and provides experimental evaluation of the proposed algorithms.

**Additional Comments On Reviewer Discussion:**

During rebuttal authors added a conclusion section and more experimental details. Further they also extended the theoretical analysis, improved the explanation of the results, and added more comparison with related works.

---

### Decision · Program_Chairs · 2025-01-22

Accept (Poster)